# Transforming jet flavour tagging at ATLAS

**The ATLAS Collaboration**✉

Jet flavour tagging enables the identification of jets originating from heavy-flavour quarks in proton–proton collisions at the Large Hadron Collider, playing a critical role in its physics programmes. This paper presents GN2, a transformer-based flavour tagging algorithm deployed by the ATLAS Collaboration that represents a different methodology compared to previous approaches. Designed to classify jets based on the flavour of their constituent particles, GN2 processes low-level tracking information in an end-to-end architecture and incorporates physics-informed auxiliary training objectives to enhance both interpretability and performance. Its performance is validated in both simulation and collision data. The measured $c$-jet (light-jet) rejection in data is improved by a factor of 3.5 (1.8) for a 70% $b$-jet tagging efficiency, compared to the previous algorithm. GN2 provides substantial benefits for physics analyses involving heavy-flavour jets, such as measurements of Higgs boson pair production and the couplings of bottom and charm quarks to the Higgs boson, and demonstrates the impact of advanced machine learning methods in experimental particle physics.

The Large Hadron Collider (LHC)[1] is the world's most powerful particle collider. It is used to extend the boundaries of our understanding of fundamental particles and their interactions. It offers a unique opportunity to test the Standard Model (SM) of particle physics, as well as search for new phenomena beyond the Standard Model (BSM). The demanding experimental conditions at the LHC necessitate continuous innovation by the main experiments, pushing them to apply cutting-edge technologies to efficiently identify physics processes of interest within the largest proton–proton ($pp$) collision dataset ever recorded. Hadronic jets, collimated streams of particles initialised by quarks or gluons, are the most abundant physics objects in $pp$ collision events, and their characteristics are widely utilised in data analyses.

The flavour of a hadronic jet is determined by the types of hadrons or leptons it contains. Flavour tagging concerns the classification of hadronic jets into those containing $b$-hadrons ($b$-jets), $c$-hadrons ($c$-jets), hadronic $\tau$-lepton decays ($\tau$-jets), and none of the above (light-jets), using algorithms sensitive to the distinctive properties of the respective classes. Since the beginning of Run 1 of the LHC (2009–2013), the ATLAS experiment[2,3] has achieved continuous improvement in the performance of these algorithms. The progress has mostly been driven by the integration of machine-learning techniques, including boosted decision trees and neural networks. The state-of-the-art algorithms used thus far to analyse the data at

$\sqrt{s} = 13$ TeV from Run 2 of the LHC (2015–2018)[4,5] led to very impactful physics results such as the observations of the Higgs boson decaying to bottom quarks[6] and its production in association with a pair of top quarks[7]. Flavour tagging plays an essential role in the comprehensive research programme of ATLAS, which includes precision measurements of the Higgs boson[8], top quark[9] and other SM processes[10], as well as the searches for supersymmetry[11] and other BSM phenomena[12]. This work describes a flavour tagging algorithm developed by the ATLAS Collaboration for the analysis of data from $pp$ collisions recorded during Run 2 (2015–2018) and Run 3 (2022–2026) of the LHC at centre-of-mass energies of $\sqrt{s} = 13$ TeV and $\sqrt{s} = 13.6$ TeV, respectively.

Flavour-tagging techniques rely on the long lifetime, high mass, high decay multiplicity and characteristic decay modes of $b$- and $c$-hadrons, and the properties of heavy-quark fragmentation[13]. The typical lifetime of the order of $\tau \approx 1.5$ ps[13–15] for $b$-hadrons in jets with transverse momenta in the range from tens to hundreds of GeV results in them travelling a mean flight length $\langle l \rangle = \beta\gamma c\tau$ in the range from few millimetres to centimetres before decaying, which often leads to a secondary vertex significantly displaced from the collision point. Displaced vertices can also be produced by $c$-hadrons, which have lifetimes of $\tau \approx 0.2$–1.0 ps, depending on the species[16–18], and $\tau$-leptons, which have a lifetime of $\tau \approx 0.29$ ps but a much lower decay

multiplicity[18,19]. The majority of *b*-jets also contain a tertiary vertex from the decay of the *c*-hadron produced in the *b*-hadron decay.

The traditional flavour-tagging algorithms developed by the ATLAS Collaboration are based on a two-stage approach[4,5,20]. In the first step, specialised low-level algorithms employ complementary approaches to extract information from the trajectories of the charged-particle constituents ('tracks') associated with the jet. These specialised algorithms either rely on the properties of individual tracks or leverage their correlations with properties of other tracks to explicitly reconstruct displaced vertices. In the second step, the outputs of low-level algorithms are subsequently combined in a high-level multivariate classifier to maximise performance. The most recent algorithm employed by the ATLAS Collaboration, following this paradigm, is a deep neural network (DL1d) that leverages a low-level track-based algorithm (DIPS)[21] based on Deep Sets[22]. DL1d has already improved the performance by a factor of 1.3 relative to the most advanced algorithm used in published Run-2 physics analyses[5].

The introduction of graph neural networks for object reconstruction in particle physics experiments[23] prompted a shift in the design strategy of the ATLAS Collaboration. This led to the development of the General Network (GN) series of flavour-tagging algorithms, which directly process track and jet information and are trained using target labels extracted from Monte Carlo (MC) simulation. In parallel, the CMS Collaboration followed a similar trajectory, evolving from two-stage approaches[24,25] to unified, end-to-end network architectures[26–28].

The ATLAS GN tagger uses jet flavour prediction as its primary training target and introduces auxiliary training objectives to reconstruct the internal structure of a jet by grouping tracks originating from a common vertex and by predicting the underlying physics process from which each track originated. Such physics domain knowledge is embedded in a combined loss function that enables a simultaneous optimisation, instead of relying on manually optimised low-level algorithms. This flexible structure allows the swift re-tuning of the algorithms to suit alternative experimental conditions or physics goals. A demonstrator version, GN1, achieves the above design goals using a graph-neural-network[29], while the deployment version, GN2, applies a single transformer model[30], illustrated in Fig. 1. Details of the

algorithm architectures are summarised in the 'Methods' section, together with descriptions of the ATLAS detector, simulation samples, physics objects, and analysis strategies.

GN2 achieves a remarkable performance boost compared with the DL1d algorithm, with improvements by a factor of 1.5-4 observed in its major experimental applications. The deployment of GN2 should greatly enhance the physics reach of ATLAS in flagship analyses, such as the search for Higgs pair production and the *c*-quark Yukawa coupling measurement, for which the projected sensitivity at the High Luminosity LHC is improved by up to 30%[31]. These improvements do not come with a strong dependence on the choice and configuration of the MC event generator, and are confirmed by measured performance in recorded collisions. The innovative auxiliary training objectives bring excellent interpretability and opens up new avenues for future applications.

To facilitate future developments and strengthen the connections between collider experiments and the broader scientific research community, a subset of the training sample with all the required information to train GN2 can be acquired via the CERN Open Data Portal[32,33].

## Results
### Algorithm performance in simulation

The performance of a *b*-tagging algorithm is evaluated based on its ability to reject *c*-, *τ*- and light-jets while maintaining a desired *b*-jet tagging efficiency. Similarly, the *c*-tagging performance is assessed by its capability to distinguish *c*-jets from the other jet flavours. The data samples used for training and evaluation of the model must contain jets from all flavour classes. This is achieved using jets sampled from a mixture of simulated top quark pair ($t\bar{t}$) and $Z'$ events, where the latter sample considers a hypothetical heavy BSM particle, $Z'$[34], which can decay into pairs of *b*-quarks, *c*-quarks, *τ*-leptons or light quarks, to populate jets in the TeV regime. The samples are simulated with MC event generators at centre-of-mass energies of both $\sqrt{s} = 13$ TeV and $\sqrt{s} = 13.6$ TeV. All simulated events are processed through the ATLAS detector simulation[35] based on GEANT4[36–38]. Further details on the simulation samples and the jet flavour labelling are discussed in the 'Methods' section. A mixture of samples generated at $\sqrt{s} = 13.6$ TeV

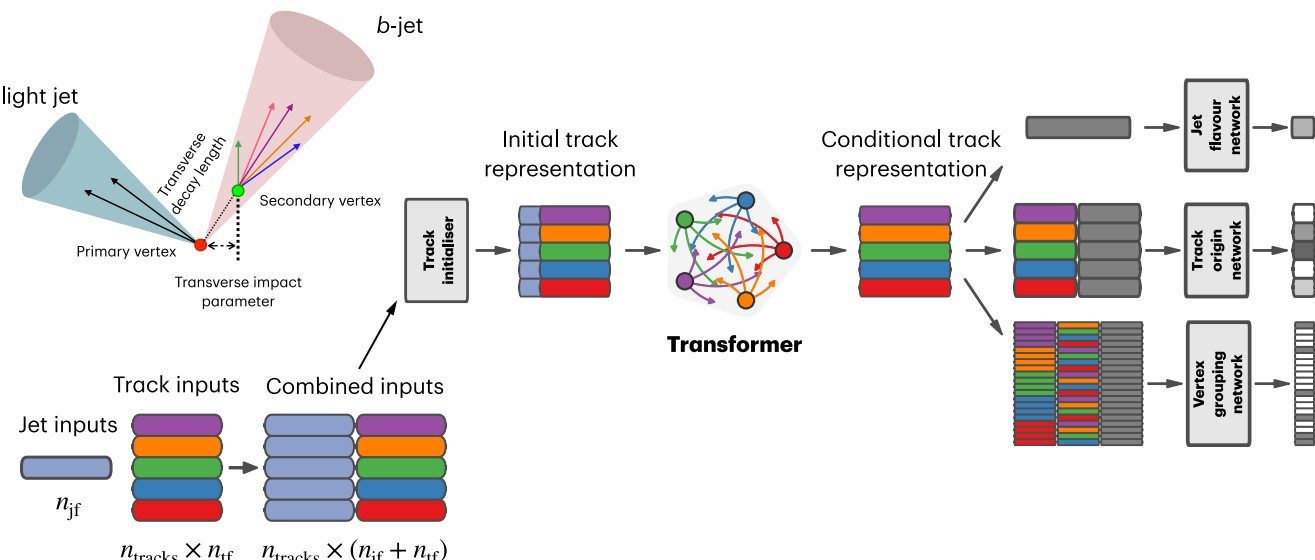

**Fig. 1 | Illustration of the GN2 algorithm with jet and track input variables, discriminating between jet flavours by exploiting secondary vertices and other properties stemming from the displaced decays of *b*-hadrons, in the transverse plane.** The jet features are copied for each track associated with the jet. The combined vectors are then fed into a per-track initialisation network, followed by a transformer encoder and a global representation of the jet. $n_{jf}$ ($n_{tf}$) corresponds to the number of jet (track) features. The pooled jet representation and output track embeddings are provided as inputs to the three task-specific networks. Details of the GN2 architecture are summarised in the 'Methods' section.

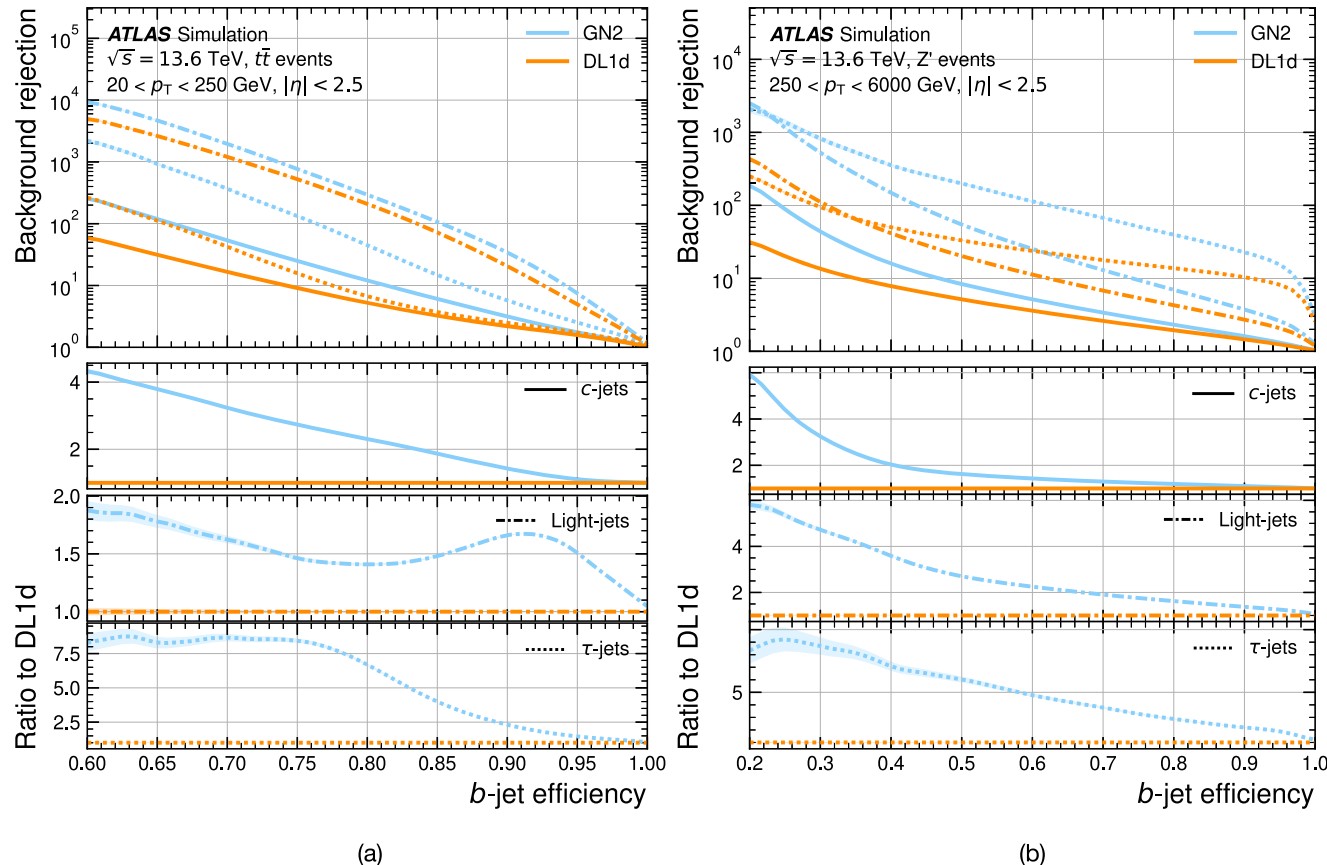

**Fig. 2 | $b$-tagging performance of GN2 and DL1d evaluated in MC simulations.** The $c$-jet (solid), light-jet (dotted-dashed), and $\tau$-jet (dashed) rejections as a function of the $b$-jet tagging efficiency for **a** jets in the $t\bar{t}$ sample with $20 < p_T < 250$ GeV and **b** jets in the $Z'$ sample with $250 < p_t < 6000$ GeV, for both GN2 (light blue) and DL1d (dark orange). The performance of GN2 with respect to DL1d is shown in the bottom panels. The 68% confidence intervals calculated assuming no correlations between the rejections are indicated by the shaded regions, and the uncertainty on each rejection is obtained according to a binomial distribution.

and $\sqrt{s} = 13$ TeV is used in the training, to achieve similar performance in both conditions. In this section, the performance evaluated with Run-3 samples at $\sqrt{s} = 13.6$ TeV is presented. Jets are classified for $b$-tagging using a single discriminant $D_b$, which combines the algorithm's jet flavour prediction output probabilities of a jet being a $b$-jet ($p_b$), a $c$-jet ($p_c$), a $\tau$-jet ($p_\tau$) or a light-jet ($p_u$) and is defined as:

$$D_b = \log\left(\frac{p_b}{f_c p_c + f_\tau p_\tau + (1 - f_c - f_\tau)p_u}\right).\quad (1)$$

A jet is considered $b$-tagged if it has a $D_b$ score larger than a given value. A selection on $D_b$ defines an operating point (OP) associated with a certain inclusive $b$-jet tagging efficiency, calculated as the fraction of $b$-jets that are $b$-tagged. The mis-tagging rate for $c$-, $\tau$- and light-jets is determined by the fraction of jets that are mistakenly $b$-tagged, for that given jet flavour, and the rejection is the reciprocal of the mis-tagging rate. The ATLAS Collaboration uses a sample of simulated $t\bar{t}$ events, where most jets have a $p_T$ below 250 GeV, to derive the OPs. The free parameters $f_{c(\tau)}$ determine the relative weighting between $p_{c(\tau)}$ and $p_u$ in the discriminant $D_b$. The specific value of $f_c$ is determined through an optimisation procedure aimed at obtaining a certain balance between rejections of $c$-jets and light-jets in simulated $t\bar{t}$ events. The value of $f_\tau$ is optimised to maximise the $\tau$-jet rejection, while ensuring a negligible impact upon the $c$-jet and light-jet rejection. In the case of GN2, $f_{c(\tau)}$ is set to 0.2 (0.05), while for DL1d, which does not have a $\tau$-jet output in the model, $f_c$ is set to 0.018. For GN2, $f_c$ is tuned to reach a much higher $c$-jet rejection, while still achieving a better light-jet rejection, compared with DL1d.

Figure 2 illustrates the tagger performance in terms of the $c$-jet, light-jet and $\tau$-jet rejection as a function of the $b$-jet tagging efficiency. In both the $t\bar{t}$ and $Z'$ samples, GN2 exhibits significantly better background rejection compared with DL1d across the entire range of $b$-jet tagging efficiencies. The degree of improvement depends on the $b$-jet tagging efficiency. In the $t\bar{t}$ sample, the $c$-jet (light-jet) rejection of GN2 improves by more than a factor of 3 (1.6), compared with DL1d, for the most commonly used 70% OP. The performance of both algorithms starts degrading once the jet $p_T$ reaches around 200 GeV, due to several confounding factors, including suboptimal tracking performance in dense environments where the spatial separation between tracks becomes smaller[39]. In the $Z'$ sample, applying the 70% OP selection on $D_b$ yields a $b$-jet tagging efficiency of 30%, and the $c$-jet (light-jet) rejection of GN2 improves by more than a factor of 3 (4), compared with DL1d. The inclusion of a $\tau$-jet output node in GN2 leads to an even greater enhancement in the $\tau$-jet rejection, by up to a factor of 8 (9) for jets in the $t\bar{t}$ ($Z'$) sample, without significantly degrading the $c$-jet and light-jet rejection.

The performance of a $c$-tagging algorithm is evaluated based on its ability to reject $b$-, $\tau$- and light-jets while maintaining a desired $c$-jet tagging efficiency. Due to the end-to-end architecture that does not rely on low-level tagger inputs, GN2 can seamlessly be adapted as a $c$-tagging algorithm without re-training any lower level algorithms. Similar to $b$-tagging, a discriminant, $D_c$, is constructed as:

$$D_c = \log\left(\frac{p_c}{f_b p_b + f_\tau p_\tau + (1 - f_b - f_\tau)p_u}\right),\quad (2)$$

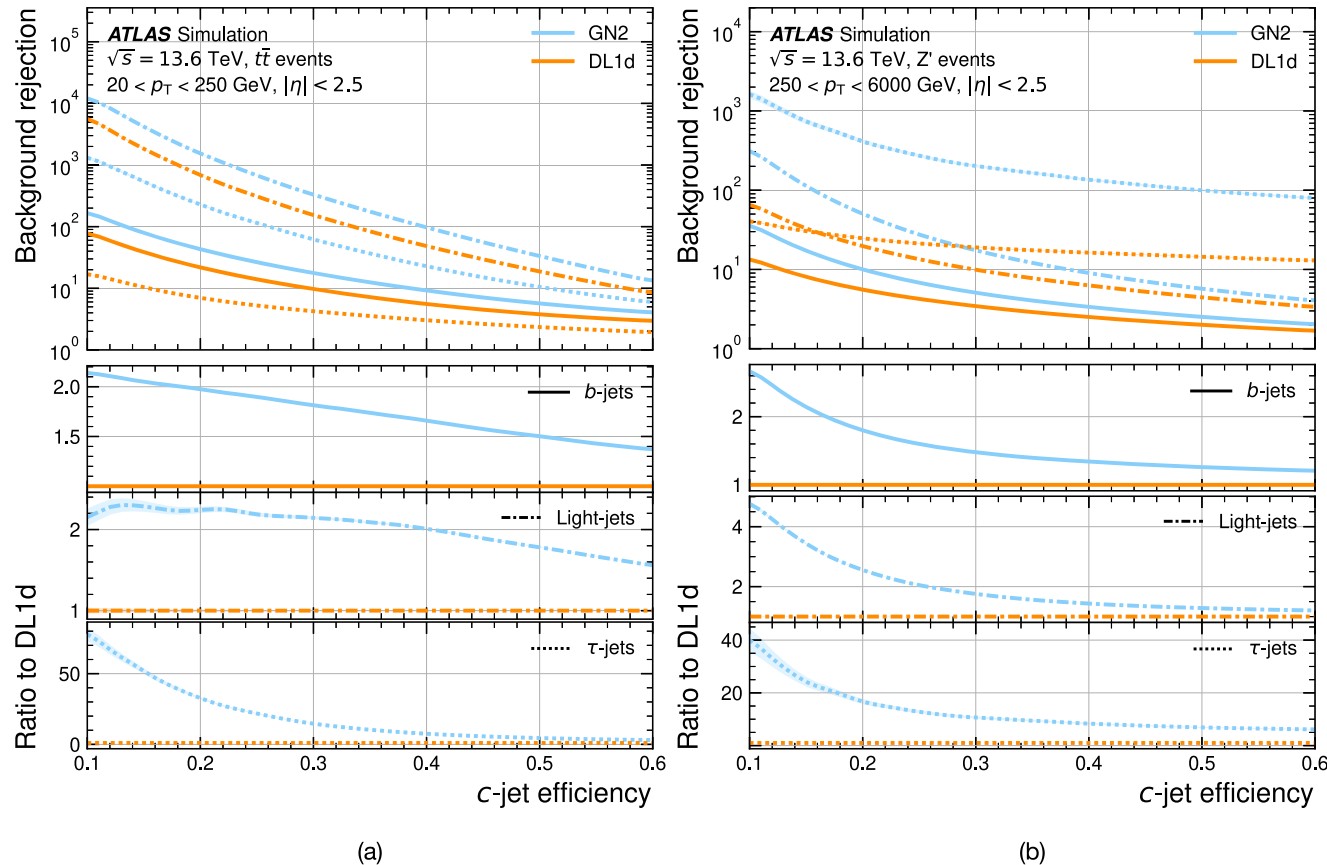

**Fig. 3 | c-tagging performance of GN2 and DL1d evaluated in MC simulations.** The $b$-jet (solid), light-jet (dotted-dashed), and $\tau$-jet (dashed) rejections as a function of the $c$-jet tagging efficiency for **a** jets in the $t\bar{t}$ sample with $20 < p_T < 250$ GeV and **b** jets in the $Z'$ sample with $250 < p_T < 6000$ GeV, for both GN2 (light blue) and DL1d (dark orange). The performance of GN2 relative to DL1d is shown in the bottom panels. The 68% confidence intervals calculated assuming no correlations between the rejections are indicated by the shaded regions, and the uncertainty on each rejection is obtained according to a binomial distribution.

where $f_{b(\tau)}$ is the free parameter that controls the flavour composition of the background in the background hypothesis. The value chosen for $f_{b(\tau)}$ is 0.3 (0.01) for GN2, while for DL1d, $f_b$ is 0.1, following a similar optimisation procedure as for $D_b$.

The $c$-tagging performance of DL1d and GN2 are compared in Fig. 3, which shows a significant improvement in performance across all $c$-jet tagging efficiencies. The $b$-jet (light-jet) rejection is enhanced by a factor of approximately 1.8 (2.2) in the $t\bar{t}$ sample at a 30% $c$-jet tagging efficiency, which is a typical choice in measurements of the $c$-quark Yukawa coupling[40]. The $b$-jet (light-jet) rejection is increased by a factor of approximately 2.7 (4.7) in the $Z'$ sample at a corresponding efficiency of 10%. The $\tau$-jet rejection is improved by a factor of approximately 15 (40) in the $t\bar{t}$ ($Z'$) sample.

**Algorithm performance in collision data**

Due to imperfections in the physics modelling of the MC generator and in the simulated detector response, the distribution of the input variables to the algorithms and their correlations differ between collision data and simulation, resulting in a performance difference. It is not practical to correct each individual mis-modelled variable, so dedicated calibration analyses are employed to measure the tagging efficiency of $b$-jets, $c$-jets and light-jets for pre-defined OPs directly[4,41,42]. In the case of the GN2 algorithm, five OPs are defined corresponding to inclusive $b$-jet tagging efficiencies of 65%, 70%, 77%, 85% and 90% while for DL1d four OPs are constructed corresponding to inclusive $b$-jet tagging efficiencies of 60%, 70%, 77% and 85%. The results presented in this paper are derived using $pp$ collision data recorded during Run 2 of the LHC at $\sqrt{s} = 13$ TeV, corresponding to an integrated luminosity of 140 fb$^{-1}$. The tagging performance in data for $b$-jets, $c$-jets, and light-jets is measured, in order to obtain jet-flavour-dependent simulation-to-data correction factors, binned in jet $p_T$. They are applied to MC-simulated jets to rescale their tagging efficiencies and mis-tagging rates to match those measured in collision data. The calibration of $b$-jets and $c$-jets is done with $t\bar{t}$ events[4,41], while the calibration of light-jets is performed using jets produced in association with a $Z$ boson[42]. Details of the calibration analyses are provided in the 'Methods' section.

Figure 4 presents the calibrated tagging efficiencies and rejections of GN2 and DL1d, along with their associated uncertainties, for each OP. The inclusive efficiencies and rejections are obtained by averaging over the events in a simulated $t\bar{t}$ sample after requiring the presence of one reconstructed electron or muon. The original efficiencies from the simulated sample are included as references, enabling a direct comparison that shows similar agreement between data and simulation for both GN2 and DL1d. The GN2 tagger demonstrates clear improvements over DL1d in collision data. For instance, the measured $c$-jet (light-jet) rejection in data is increased by a factor of 3.5 (1.8) for the 70% OP. The measurements in data provide conclusive evidence of the enhanced performance enabled by advanced machine-learning algorithms in identifying heavy-flavour jets at the LHC.

## Discussion

Key challenges with machine-learning algorithms based on low-level inputs, such as GN2, include the potential loss of interpretability and the need to ensure consistent performance across different MC simulation methods. Robustness against these potential shortcomings is critical to prevent the algorithm from relying on unphysical features

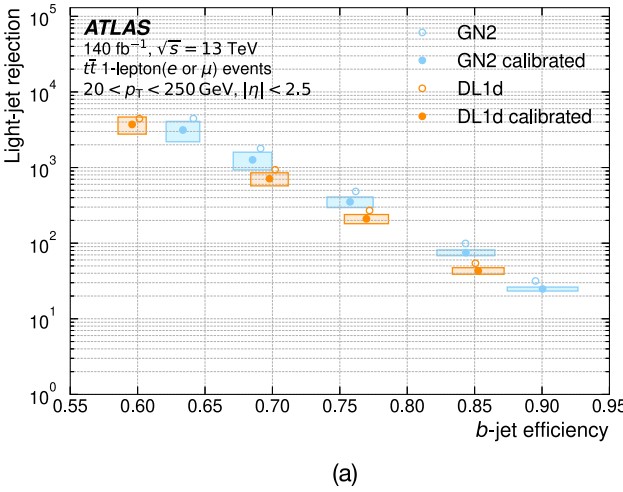

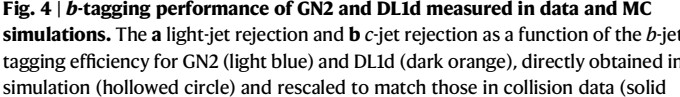

(a)                                                            (b)

**Fig. 4 | *b*-tagging performance of GN2 and DL1d measured in data and MC simulations.** The **a** light-jet rejection and **b** *c*-jet rejection as a function of the *b*-jet tagging efficiency for GN2 (light blue) and DL1d (dark orange), directly obtained in simulation (hollowed circle) and rescaled to match those in collision data (solid point). The horizontal error bands correspond to the uncertainties associated with the *b*-jet tagging efficiency measurement, while the vertical error bands indicate the uncertainties associated with the rejection measurements. A $t\bar{t}$ MC simulation sample with a reconstructed electron or muon is used to derive these results.

of the training sample. In this section, these aspects are discussed further.

## Physics inspiration and the auxiliary training objectives

A key strength of the GN2 model lies in its physics-inspired constraints, which aid the main task of jet classification while also improving the interpretability of the model. This is accomplished by incorporating two additional training objectives: predicting the origin of tracks associated with the jet and determining which tracks originate from common vertices. These objectives are not strictly necessary for the jet classification task and are therefore referred to as auxiliary training objectives. The technical implementation details are provided in the 'Methods' section.

The track classification auxiliary training objective aims to estimate the probability that a track originates from one of the following physical processes: a pile-up interaction[43]; the primary hard-scatter interaction; the decay of a *b*-hadron; the decay of a *c*-hadron produced by a *b*-hadron; the decay of a *c*-hadron; the decay of a $\tau$-lepton; or any other secondary source. Class-weighted losses are applied during training to mitigate the class imbalance, and tracks are classified by the highest-probability category during evaluation. The class weights are fixed and based on the inverse class frequencies in the training dataset.

The classification efficiency refers to the probability for the track's origin to be correctly predicted, in a group of tracks with certain target origins, while the purity corresponds to the fraction of correctly predicted tracks, within a group of tracks with specific predicted origins. When combining the two categories involving a *b*-hadron, GN2 achieves an efficiency (purity) of 84% (84%). For tracks that are not of heavy-flavour (HF) origin, the efficiency (purity) is 85% (96%). The above performance is evaluated in Run-3 samples at $\sqrt{s} = 13.6$ TeV.

The vertex finding auxiliary training objective aims to identify groups of tracks that originate from a common spatial point. Each pair of tracks in the jet is classified to determine whether they share the same vertex. Using these pair-wise compatibility scores, track groups (vertices) are formed via a union-find algorithm[44]. SV1, an existing secondary vertex reconstruction algorithm detailed in ref. 5, serves as a reference algorithm. SV1 reconstructs a single inclusive vertex, whereas GN2 can identify multiple vertices of various types within a jet. Therefore, an aggregation procedure is applied to the output of GN2 to enable a direct comparison with the single inclusive vertex produced by SV1. To study the vertex properties of *b*-jets, the identified vertex containing the most tracks that have a predicted primary origin is

removed, as this is likely to be the vertex associated with the primary hard-scatter interaction. Next, the remaining GN2 vertices that include at least one track predicted to have a HF origin are consolidated into a single inclusive vertex.

An inclusive reference vertex is constructed in simulated events, by combining all tracks from simulation-level secondary vertices within the jet that consist solely of HF tracks. A Billoir fit[45] is performed on the tracks selected by the GN2 and SV1 vertex finding algorithms to obtain the transverse displacement of the vertex, $L_{xy}$. Figure 5 presents the $L_{xy}$ distribution for vertices obtained with GN2 and DL1d in *b*-jets from a simulated $t\bar{t}$ sample, compared to the expected distribution derived from the inclusive reference vertex. GN2 consistently achieves higher vertex-finding efficiency than SV1 across the entire distribution of $L_{xy}$. The mass of the secondary vertex can also be calculated using the momenta of tracks selected by the vertex finding algorithms. The distribution of the secondary vertex mass normalised to unity is also shown in Fig. 5. Remarkably, the mass of the secondary vertices reconstructed by GN2 exhibits good agreement with the mass of the inclusive reference vertex, despite the vertex mass not being explicitly targeted during training. Unlike SV1, GN2 does not impose explicit selections on track properties such as impact parameters. This leads to a higher efficiency, albeit with a small contamination from non-HF tracks, which results in a slightly larger secondary vertex mass.

GN2 identifies all types of vertices including those from material interactions, photon conversions, and in-flight decays of light hadrons. Consequently, the rate of vertices reconstructed in light-jets, defined as the fraction of light-jets containing a GN2 inclusive vertex, is expected to be much higher compared to SV1 if no selections are applied in the aggregation procedure described above. Figure 6 confirms this with light-jets in the simulated $t\bar{t}$ sample and shows that once requiring the GN2 inclusive vertex to contain at least one track with predicted HF origin, the vertexing rate is dramatically reduced, down to the same level as SV1.

To test the impact of the auxiliary objectives on the performance of the main jet classification task, various GN2 configurations are trained and tested. The resulting *c*-jet and light-jet rejections are reduced by up to 30% in both the $t\bar{t}$ and $Z'$ samples, if both auxiliary objectives are disabled. Disabling only one of them is sufficient to recover most of the performance loss, indicating that the two tasks are highly correlated in their contributions to the main jet flavour tagging objective.

Although the outputs from the auxiliary tasks described above mainly serve as a way to improve HF jet identification, with future

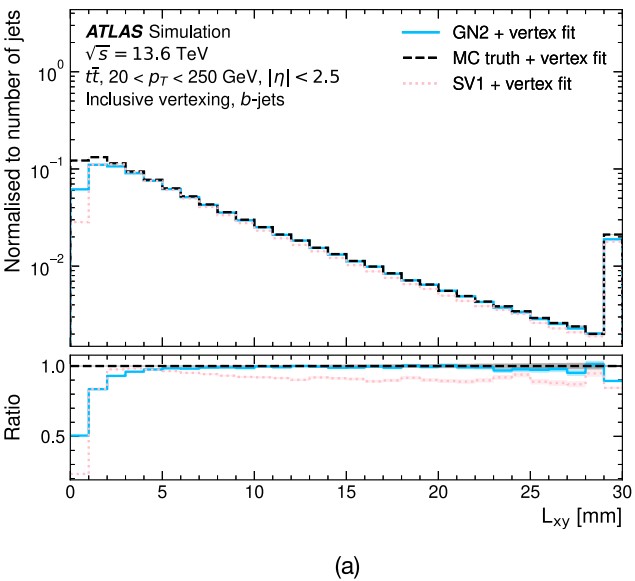
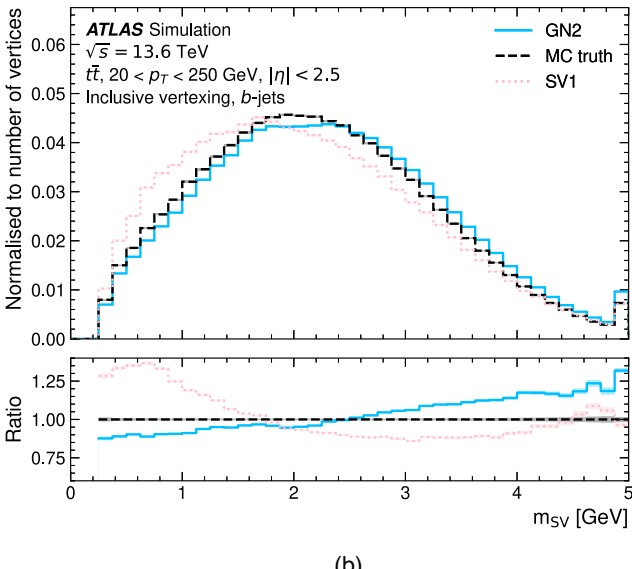

(a)  (b)

**Fig. 5 | Secondary vertex properties reconstructed using tracks grouped by the GN2 and SV1 algorithms.** The **a** transverse displacement and the **b** mass of the secondary vertex obtained by the GN2 (solid) and the SV1 (dotted) algorithms. While the transverse displacement is calculated via a Billoir fit performed on the tracks assigned to the vertex by the respective algorithm, the vertex mass is defined as the invariant mass of the same set of assigned tracks. MC truth (dashed) corresponds to an inclusive reference vertex derived from all tracks associated to simulation-level vertices containing only $b$-hadron tracks. The last bin in each plot includes overflow.

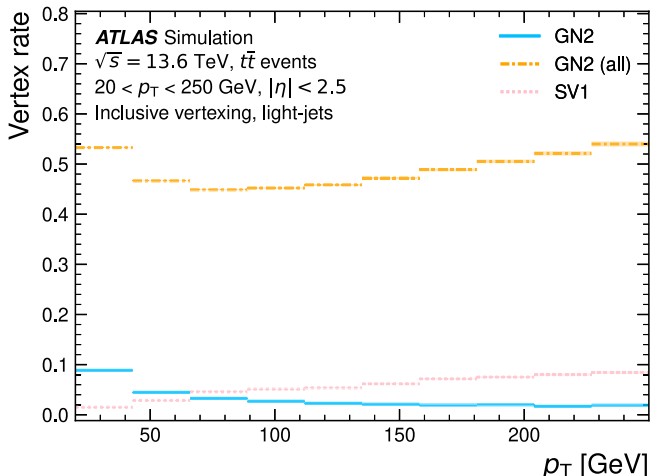

**Fig. 6 | The rate of inclusive vertices reconstructed by the GN2 algorithm in light-jets as a function of the jet $p_T$, without any selections (dotted-dashed) and with the requirement of the vertex containing at least one track with predicted HF origin (solid).** Results from the SV1 algorithm are added as a reference (dotted). The 68% confidence intervals calculated according to a binomial distribution are indicated by the shaded regions.

development, their direct usage in physics analyses remains a promising possibility.

## Robustness against generator modelling variations

Flavour-tagging algorithms are sensitive to the modelling of parton showering, hadronisation, the underlying event and the properties of heavy-hadron decays[46]. To evaluate the robustness of the algorithm against modelling variations, a comparative study of the GN2 performance in the nominal simulated $t\bar{t}$ sample used during training and samples produced with alternative generator settings, both with Run-2 conditions at $\sqrt{s} = 13$ TeV, is performed.

The event and showering generators adopted for the nominal sample are POWHEGBOX[47–50] and PYTHIA[51], respectively. The alternative samples include the use of a different showering generator (HERWIG[52–54]), whilst keeping the same event generator, and the use of SHERPA[55], which applies a different approach to all parts of the event generation model. The ratio between the efficiency obtained with an alternative generator setup and with the nominal setup is used to quantify the generator dependence of the algorithms.

Table 1 shows these ratios for $b$-jets, $c$-jets, and light-jets at the 70% and 85% OPs. Across the tested generators, the GN2 performance for $b$-jets agrees to within 1–2%, for $c$-jets the agreement is within 10%, and for light-jets the agreement is within 4%. Similar agreement is also observed for other OPs. The level of relative disagreement between DL1d and GN2 is close to unity, suggesting that despite the GN2 model being significantly more complex, it does not induce additional generator dependence.

## Methods

### The ATLAS detector

The ATLAS experiment[2,3] at the LHC is a multipurpose particle detector with a forward-backward symmetric cylindrical geometry and a solid-angle coverage of almost $4\pi$. It is used to record particles produced in $pp$ collisions at the LHC through a combination of particle position and energy measurements. It includes an inner-tracking detector (ID) surrounded by a thin superconducting solenoid providing a 2 T axial magnetic field, electromagnetic and hadronic calorimeters, and a muon spectrometer. The ID consists of silicon pixel, silicon microstrip, and transition radiation tracking detectors. The muon spectrometer surrounds the calorimeters and is based on three large superconducting air-core toroidal magnets with eight coils each providing a field integral of between 2 T m and 6 T m across the detector.

An extensive software suite[56] is used in data simulation, the reconstruction and analysis of real and simulated data, detector operations, and the trigger and data acquisition systems of the experiment.

### Monte Carlo simulation samples

The $t\bar{t}$ events at $\sqrt{s} = 13$ TeV are modelled using the POWHEGBOX[v2][47–50] event generator at next-to-leading-order (NLO) in the strong coupling

**Table 1 | Ratios of the efficiencies obtained with samples using alternative MC generators, relative to those in the nominal POWHEGBOX + PYTHIA sample used during training of the algorithm**

|  |  | 70% OP | | 85% OP | |
|---|---|---|---|---|---|
|  |  | DL1d | GN2 | DL1d | GN2 |
| POWHEGBOX + HERWIG | b-jets | 0.984 | 0.984 | 0.989 | 0.990 |
|  | c-jets | 0.951 | 0.904 | 0.977 | 0.983 |
|  | light-jets | 1.003 | 1.000 | 1.015 | 1.011 |
| SHERPA | b-jets | 0.996 | 0.995 | 0.992 | 0.992 |
|  | c -jets | 0.923 | 0.938 | 0.947 | 0.931 |
|  | light-jets | 1.039 | 1.013 | 1.077 | 1.042 |

Statistical uncertainties from evaluating the same algorithm on different samples are negligible and thus not shown.

constant $\alpha_s$ with the NNPDF3.0 NLO[57] parton distribution function (PDF) set and the first-gluon-emission cut-off scale parameter $h_{damp}$ set to $1.5 m_t$, with a top-quark mass of $m_t = 172.5$ GeV. Parton shower, hadronisation, and the underlying event are modelled by interfacing POWHEGBOX[v2] to PYTHIA 8.230[51], using the A14 set of tuned parameters[58] and the NNPDF2.3LO PDF set[59]. The decays of $b$- and $c$-hadrons are performed by EVTGEN 1.6.0[60].

The $Z'$ events at $\sqrt{s} = 13$ TeV used to enrich the dataset with high-$p_T$ jets are generated using PYTHIA 8.243 with the A14 set of tuned parameters for the underlying event and the leading-order (LO) NNPDF 2.3LO PDF set. A broad jet $p_T$ spectrum with an almost uniform distribution between 250 GeV and 1.5 TeV and a tail expanding to 6 TeV is obtained by applying a weighting factor that modifies the original cross-section of the $Z'$ resonance. The decays to $b\bar{b}$, $c\bar{c}$, and light-flavour quark pairs are set to have equal branching ratios, while the branching ratio to $\tau\bar{\tau}$ is set to 5%. The decays of $b$- and $c$-hadrons are performed by EVTGEN 1.7.0.

The $t\bar{t}$ and $Z'$ events at $\sqrt{s} = 13.6$ TeV are produced using the same setups, but with newer versions of PYTHIA (8.308) and EVTGEN (2.1.1).

The impact of using different generators and models for parton shower and hadronisation is studied with simulated $t\bar{t}$ events from alternative generator setups. Two scenarios are considered, where either only the showering algorithm is varied, or the entire chain is changed. The former is achieved by interfacing the POWHEGBOX[v2] generator with the HERWIG[7.2.1][52–54] showering algorithm using the HERWIG[7.1] default set of tuned parameters, with the NNPDF 3.0 NLO set of PDFs. The latter is realised with the SHERPA[2.2.12][55] generator, using NLO-accurate matrix elements for up to one additional parton, and LO-accurate matrix elements for up to four additional partons, calculated with the COMIX[61] and OPENLOOPS[62–64] libraries. The SHERPA parton shower[65,66] is applied using the MEPS@NLO prescription[67–70] and the set of tuned parameters developed by the SHERPA authors to match the NNPDF 3.0 NNLO set of PDFs.

## Objects for flavour tagging
ATLAS uses a right-handed coordinate system with its origin at the nominal interaction point in the centre of the detector and the $z$-axis along the beam pipe. The $x$-axis points from the nominal interaction point to the centre of the LHC ring, and the $y$-axis points upwards. Cylindrical coordinates $(r, \phi)$ are used in the transverse plane, $\phi$ being the azimuthal angle around the $z$-axis. The pseudorapidity is defined in terms of the polar angle $\theta$ as $\eta = -\ln\tan(\theta/2)$. Angular distance is measured in units of $\Delta R \equiv \sqrt{(\Delta\eta)^2 + (\Delta\phi)^2}$.

The fundamental objects for flavour tagging are jets, tracks, and vertices. A concise description of these objects is provided below, while a detailed description is available in ref. 5.

Tracks are reconstructed from ID information[39,71]. To be considered for jet flavour tagging they are required to be within $|\eta| < 2.5$, have $p_T > 0.5$ GeV and satisfy criteria designed to reject fake and poorly measured tracks[72].

Primary vertices (PVs) are reconstructed from tracks in the luminous region of the colliding LHC beams using an adaptive multi-vertex filter[73,74]. The PV with the highest sum of squared transverse momenta $p_T$ of contributing tracks is selected as the primary interaction point (IP) and provides the reference point in an event. The distance of closest approach of a track to the IP, the 'perigee', is indicated in the transverse plane by the transverse impact parameter $d_0$. The longitudinal separation between the IP and the point on the track where $d_0$ is measured, is indicated by the longitudinal impact parameter $z_0$. Tracks with large impact parameters can indicate the presence of displaced decays, providing vital information to the flavour tagging algorithms.

The DIPS and GN2 algorithms require tracks to be reconstructed from at least 8 hits in the silicon detector, at most one of which contributes to two tracks, at most two 'holes' in the silicon detector, and at most one hole in the pixel detector, where hole denotes a hit missing where one is expected from the track trajectory. Further, requirements on the track impact parameters, $|d_0| < 3.5$ mm and $|z_0 \sin\theta| < 5$ mm, retain charged particle tracks originating from HF hadron decays while suppressing tracks from other sources.

Jets are reconstructed using the anti-$k_t$ algorithm[75] with radius parameter $R = 0.4$ using the 'fastjet' package[76]. The input constituents are 'particle-flow' objects[77] which combine signals in the ATLAS calorimeters and ID to exploit precision tracking information for low-$p_T$ charged hadrons spatially matched with calorimeter energy deposits. The jet $p_T$ is corrected to the corresponding particle-level jet $p_T$ using calibration techniques described in ref. 78. The jets are required to have $p_T > 20$ GeV (to be within the valid calibration range) and $|\eta| < 2.5$ (to be within the tracking fiducial volume set by the ID acceptance) to be considered for flavour tagging. Additionally, jets from pile-up interactions are suppressed by the 'jet vertex tagger' (JVT) algorithm[79], which uses the ID tracks associated with the jet to form a multivariate discriminant. The JVT efficiency for jets originating from the IP is 92% in the simulation. The jet axis, derived from the sum of the momenta of the jet constituents, is used when associating tracks with the jet and when assigning a lifetime sign to the tracks' impact parameters. Tracks are associated with a given jet by setting a maximum allowed angular separation $\Delta R$ between the track momenta, defined at the perigee, and the jet axis. The $\Delta R$ requirement varies as a function of the jet $p_T$ to account for decay products from $b$-hadrons with larger $p_T$ being more collimated, ranging from 0.45 for jet $p_T = 20$ GeV to 0.26 for jet $p_T > 150$ GeV. If a track can be associated with multiple jets, it is assigned to the jet closest in $\Delta R$. The sign convention for the lifetime-signed impact parameters assigns a positive sign if the track intersects the jet axis in the transverse plane in front of the IP, and a negative sign if the intersection lies behind the IP[20]. The flavour labels of jets in simulation are assigned depending on the hadrons associated with the jet. The set of weakly decaying hadrons and hadronically decaying $\tau$-leptons with $p_T > 5$ GeV within a $\Delta R < 0.3$ cone around the jet axis determines the jet flavour following a sequential labelling decision tree. A jet is labelled a $b$-jet if it contains at least one $b$-hadron with $p_T > 5$ GeV, a $c$-jet ($\tau$-jet) if it contains at least one $c$-hadron (hadronic $\tau$-lepton decay) and no $b$-hadron, and otherwise it is called a light-jet, where the latter is an inclusive label for the jets originating from a light quark or gluon. These labels are used both for training the algorithms, and for evaluating their performance.

Targets for the auxiliary training objectives are obtained from the simulation-level event record. Tracks are matched with simulation-level particles using the approach in ref. 39. Track-origin labels are obtained by analysing the decay history of the matched particles, while track-pair-compatibility labels are obtained by considering the

production vertices of the matched particles. Production vertices within 0.1 mm in 3D space are merged to account for the finite resolution of the detector, and the matched track-pairs are assigned the same label.

## The algorithm architecture

The primary flavour tagging algorithm presented is GN2, which directly learns from the charged particle tracks via a transformer-based model. Another algorithm, DL1d, which follows previous approaches of combining inputs from several low-level taggers in a multivariate technique, is also discussed as a baseline reference.

Both algorithms are trained on a dataset created from combining the simulated $t\bar{t}$ and $Z'$ samples described earlier in this section. Jets with $20\,\text{GeV} < p_\text{T} < 250\,\text{GeV}$ are taken from the $t\bar{t}$ sample and those with $250\,\text{GeV} < p_\text{T} < 6\,\text{TeV}$ from the $Z'$ sample. The $b$-jets, light-jets and $\tau$-jets are re-sampled in $p_\text{T}$ and $\eta$ to match the corresponding $c$-jet distributions, thereby preventing the models from discriminating between jet flavours based on relative kinematic differences. All input variables to the algorithm training are normalised to have zero mean and unit variance. A coarse optimisation of hyperparameters, such as the number of layers, is carried out for both algorithms, and the AdamW[80] (Adam[81]) optimiser is used for training GN2 (DL1d) with the learning rate and optimisation schedule defined below.

The GN2 algorithm is an end-to-end architecture without any intermediate taggers involved, as illustrated in Fig. 1. It is based on the GN1[29] demonstrator version of the algorithm, replacing the Graph Attention Network[82] with a Transformer[30] along with other architecture optimisations. GN2 directly accepts information about the jet and associated tracks that are provided by the standard event reconstruction. This results in a simpler and more flexible algorithm which can be easily reoptimised for different physics objectives, such as the identification of highly energetic Higgs bosons decaying into $b$- or $c$-quark pairs[83], jet energy regression[84], exotic jet tagging[85], and jet flavour tagging in the ATLAS high-level trigger[86]. Additionally, when compared with DL1d, GN2 is trained to recognise an additional class of jets that originate from hadronic $\tau$-lepton decays.

First, the jet features are concatenated with a fixed-size array of 40 track feature vectors, with unused elements masked when fewer than 40 tracks are available, allowing it to handle variable track multiplicity without zero-padding. Tracks with smaller absolute track impact parameter significance[5] are dropped if there are more than 40 tracks. The same inputs as for GN1[29] are used, except the variables related to holes in the silicon tracker, which were found to have no impact on performance. A complementary interpretability analysis using integrated gradients shows that the impact parameter significances and angular variables emerge as particularly influential[87]. The combined vectors are then fed into a per-track initialisation network, which is composed of a single hidden layer and an output layer of size 256. Next, a four layer transformer encoder with eight attention heads is used to produce track representations that incorporate information from other tracks inside the jet. The transformer has an embedding size of 256 and a feed-forward dimension of 512, and uses pre-LayerNorm[88]. After the transformer encoder, the output track representations are projected down to dimension 128, and a global representation of the jet is produced using attention pooling[89]. The pooled jet representation and output track embeddings are provided as inputs to the three task-specific networks. The primary objective, jet classification, uses only the pooled jet representation and has an output layer of size 4, providing $p_b$, $p_c$, $p_u$ and $p_\tau$ for the final discriminant definition. The two auxiliary objectives introduced in the Discussion section take advantage of the track embeddings, in addition to the global jet representation. The track origin classification task uses individual track embeddings and has 7 output categories, while the track-pair compatibility task employs a binary output layer, using the embeddings of each pair of tracks. Each task-specific network consists of three hidden

layers with size 128, 64 and 32, respectively. ReLU activation[90] is used throughout the model. Cross-entropy loss is used by all three task-specific networks, which is combined with tunable weights to form the final loss function, enabling a simultaneous optimisation of the entire algorithm. GN2 applies the same auxiliary network structures and loss weights as GN1[29].

GN2 is trained using a 4-fold strategy to prevent memorisation of the training samples, given their possible use in ATLAS physics analyses. Jets are assigned to one of the four folds pseudo-randomly, with a number seeded by the event number and discrete jet properties. Four classifiers are then trained, each excluding one of the four folds from the training dataset. In physics analysis, each jet is tagged using the classifier it was excluded from during training. Each of the four networks has approximately 2.3M trainable parameters and is trained using approximately 45M (18M) $b$-jets, 45M (18M) $c$-jets, 90M (36M) light-jets and 6.25M (2.5M) $\tau$-jets from the $t\bar{t}$ ($Z'$) sample, simulated at both $\sqrt{s} = 13\,\text{TeV}$ and $\sqrt{s} = 13.6\,\text{TeV}$, with a mixing ratio of 2:1. A learning rate scheduler with cosine annealing[91] is used with the initial learning rate set to $1 \times 10^{-7}$, which is increased to $5 \times 10^{-4}$ after the first 1% of training steps have been completed. It reduces to $1 \times 10^{-5}$ over the remainder of the training run. A weight decay of $1 \times 10^{-5}$ is also added. A batch size of 12,000 is adopted. The different folds have compatible performance within statistical uncertainty. The training data is translated from a standard ATLAS format[56] to HDF5[92]. The network is trained with PYTORCH LIGHTNING[93–95], consuming roughly 300 GPU hours on an NVIDIA A100 card. It is deployed in ATLAS software with ONNXRUNTIME[96], adding negligible CPU time. With the updated architecture and training setup, the $c$-jet (light-jet) rejection is improved by a factor of 1.5 (1.7) for a 70% $b$-jet tagging efficiency, in the $t\bar{t}$ sample, and by a factor of 1.3 (1.4) for the corresponding 30% $b$-jet tagging efficiency, in the $Z'$ sample.

The DL1d algorithm inherits the architecture from its predecessor DL1r, described in Ref. 5, but processes track impact parameters with the DIPS algorithm based on DeepSets[21,22] instead of a recurrent neural network[97]. Overall, 44 input features are fed into DL1d, including the jet $p_\text{T}$ and $\eta$. The architecture of DL1d includes eight hidden layers of size 256, 128, 60, 48, 36, 24, 12, and 6, each followed with ReLU activation and batch normalisation. The training was performed with a learning rate of $1 \times 10^{-3}$ and training batch size of 15,000. The training data pipeline is similar to GN2, with the exception that training is done with KERAS and TENSORFLOW[98] via UMAMI, a dedicated Python toolkit[99], and deployed in the ATLAS software with LWTNN[100].

## Performance measurement strategies in collision data

The measurement of the $b$-jet tagging efficiency in collision data is carried out in a roughly 90% pure sample of $t\bar{t}$ events where both top quarks decay leptonically into a lepton, a neutrino and a $b$-quark. The events are required to contain exactly one electron and one muon of opposite charge, in addition to two jets. The invariant masses of the two lepton-jet pairs are used to define one region enriched in $b$-jets and three control regions (CRs). The $b$-jet-enriched region is determined by requiring that both lepton-jet pairs have invariant masses compatible with an on-shell top quark decay. The CRs are used in a likelihood fit to constrain the predicted jet flavour composition. They are constructed to have increased fractions of non-$b$-jets by requiring that at least one or both of the lepton-jet pairs do not originate from the same top-quark decay. The analysis employs a statistical model based on a likelihood function that extracts the efficiency in collision data binned in $p_\text{T}$ for all the $b$-jets in the sample. The dominant systematic uncertainty comes from the modelling of $t\bar{t}$ events. Additional details on the $b$-jet calibration procedure are available in ref. 4.

The calibration measurement of the $c$-jet mis-tagging rate is performed in $t\bar{t}$ events where one top quark decays leptonically while the other top quark decays hadronically. A sample of $c$-jets is obtained

through the $W^{\pm} \to c\bar{s}(\bar{c}s)$ decay from the hadronically decaying top quark. A likelihood-based kinematic reconstruction is employed to find, among the four jets in the event, two jets associated with the hadronically decaying $W$-boson and two jets stemming from the $b$-quarks produced in the top quark decays. The mis-tagging rate of $c$-jets is determined by minimising a $\chi^2$ function computed in bins of the jet $p_T$ of the two jets from the $W$-boson decay. Additional terms that correct for the potential mis-modelling of the total number of events in each jet $p_T$ bin are estimated simultaneously from the fit to collision data, while the contribution of background events, in which no $c$-jets are associated with the $W$-boson decay, is estimated from simulations. The mis-tagging rate of light-jets in this sample is corrected using the method described below. As with the $b$-jets calibration analysis, the leading source of systematic uncertainties is the modelling of $t\bar{t}$ events. The $c$-jet mis-tagging rate calibration procedure is detailed in ref. 41.

The mis-tagging rate for light-jets is determined using jets produced in association with a $Z$ boson, where the $Z$ boson decays into muon or electron pairs. The key challenge in this calibration is to develop a method capable of extracting a light-jet mis-tagging rate in data despite the high rejection of the taggers. The method used in this work involves exploiting transformed track variables in alternate taggers that provide reduced $b(c)$-jet tagging efficiency and almost unchanged light-jet rejection. The mis-tagging rate of this modified tagger is measured from a fit to the flavour-sensitive secondary vertex mass distribution in collision data, and dedicated uncertainties are introduced so that it can be extrapolated to that of the nominal tagger. These extrapolation uncertainties are a leading source of systematic uncertainty. A detailed description of the procedure is provided in ref. 42.

## Data availability
Raw data were generated by the ATLAS experiment. Derived data supporting the findings of this study are available from the ATLAS Collaboration upon request. A subset of the training sample and instructions to train GN2 can be acquired via the CERN Open Data Portal[32,33].

## Code availability
The ATLAS data reduction software is available at *Zenodo* (https://doi.org/10.5281/zenodo.4772550)[102]. The GN2 training software suite can be found at *JOSS* (https://joss.theoj.org/papers/10.21105/joss.07217)[95], and the DL1d training software stack is available at *JOSS* (https://joss.theoj.org/papers/10.21105/joss.05833)[99].

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

## Acknowledgements

The authors thank CERN for the very successful operation of the LHC and its injectors, as well as the support staff at CERN and at their institutions worldwide without whom ATLAS could not be operated efficiently. The crucial computing support from all WLCG partners is acknowledged gratefully, in particular from CERN, the ATLAS Tier-1 facilities at TRIUMF/SFU (Canada), NDGF (Denmark, Norway, Sweden), CC-IN2P3 (France), KIT/GridKA (Germany), INFN-CNAF (Italy), NL-T1 (Netherlands), PIC (Spain), RAL (UK) and BNL (USA), the Tier-2 facilities worldwide and large non-WLCG resource providers. Major contributors of computing resources are listed in ref. 101. The authors gratefully acknowledge the support of ANPCyT, Argentina; YerPhI, Armenia; ARC, Australia; BMWFW and FWF, Austria; ANAS, Azerbaijan; CNPq and FAPESP, Brazil; NSERC, NRC and CFI, Canada; CERN; ANID, Chile; CAS, MOST and NSFC, China; Minciencias, Colombia; MEYS CR, Czech Republic; DNRF and DNSRC, Denmark; IN2P3-CNRS and CEA-DRF/IRFU, France; SRNSFG, Georgia; BMFTR, HGF and MPG, Germany; GSRI, Greece; RGC and Hong Kong SAR, China; ICHEP and Academy of Sciences and Humanities, Israel; INFN, Italy; MEXT and JSPS, Japan; CNRST, Morocco; NWO, Netherlands; RCN, Norway; MNiSW, Poland; FCT, Portugal; MNE/IFA, Romania; MSTDI, Serbia; MSSR, Slovakia; ARIS and MVZI, Slovenia; DSI/NRF, South Africa; MICIU/AEI, Spain; SRC and Wallenberg Foundation, Sweden; SERI, SNSF and Cantons of Bern and Geneva, Switzerland; NSTC, Taipei; TENMAK, Türkiye; STFC/UKRI, United Kingdom; DOE and NSF, United States of America. Individual groups and members have received support from BCKDF, CANARIE, CRC and DRAC, Canada; CERN-CZ, FORTE and PRIMUS, Czech Republic; COST, ERC, ERDF, Horizon 2020, ICSC-NextGenerationEU and Marie Skłodowska-Curie Actions, European Union; Investissements d'Avenir Labex, Investissements d'Avenir Idex and ANR, France; DFG and AvH Foundation, Germany; Herakleitos, Thales and Aristeia programmes co-financed by EU-ESF and the Greek NSRF, Greece; BSF-NSF and MINERVA, Israel; NCN and NAWA, Poland; La Caixa Banking Foundation, CERCA Programme Generalitat de Catalunya and PROMETEO and GenT Programmes Generalitat Valenciana, Spain; Göran Gustafssons Stiftelse, Sweden; The Royal Society and Leverhulme Trust, United Kingdom. In addition, individual members wish to acknowledge support from Armenia: Yerevan Physics Institute (FAPERJ); CERN: European Organization for Nuclear Research (CERN DOCT); Chile: Agencia Nacional de Investigación y Desarrollo (FONDECYT 1230812, FONDECYT 1240864); China: Chinese Ministry of Science and Technology (MOST-2023YFA1605700, MOST-2023YFA1609300), National Natural Science Foundation of China (NSFC - 12175119, NSFC 12275265); Czech Republic: Czech Science Foundation (GACR - 24-11373S), Ministry of Education Youth and Sports (ERC-CZ-LL2327, FORTE CZ.02.01.01/00/22_008/0004632), PRIMUS Research Programme (PRIMUS/21/SCI/017); EU: H2020 European Research Council (ERC - 101002463); European Union: European Research Council (BARD No. 101116429, ERC - 948254, ERC 101089007), European Regional Development Fund (SMASH COFUND 101081355, SLO ERDF), Horizon 2020 Framework Programme (MUCCA - CHIST-ERA-19-XAI-00), European Union, Future Artificial Intelligence Research (FAIR-NextGenerationEU PE00000013), Italian Center for High Performance Computing, Big Data and Quantum Computing (ICSC, NextGenerationEU); France: Agence Nationale de la Recherche (ANR-21-CE31-0022, ANR-22-EDIR-0002); Germany: Baden-Württemberg Stiftung (BW Stiftung-Postdoc Eliteprogramme), Deutsche Forschungsgemeinschaft (DFG - 469666862, DFG - CR 312/5-2); China: Research Grants Council (GRF); Italy: Istituto Nazionale di Fisica Nucleare (ICSC, NextGenerationEU), Ministero dell'Università e della Ricerca (NextGenEU 153D23001490006 M4C2.1.1, NextGenEU I53D23000820006 M4C2.1.1, NextGenEU I53D23001490006 M4C2.1.1, SOE2024_0000023); Japan: Japan Society for the Promotion of Science (JSPS KAKENHI JP22H01227, JSPS KAKENHI JP22H04944, JSPS KAKENHI JP22KK0227, JSPS KAKENHI JP23KK0245, JSPS KAKENHI JP24K23939); Norway: Research Council of Norway (RCN-314472); Poland: Ministry of Science and Higher Education (IDUB AGH, POB8, D4 no 9722), Polish National Science Centre (NCN 2021/42/E/ST2/00350, NCN OPUS 2023/51/B/ST2/02507, NCN OPUS nr 2022/47/B/ST2/03059, NCN UMO-2019/34/E/ST2/00393, UMO-2022/47/O/ST2/00148, UMO-2023/49/B/ST2/04085, UMO-2023/51/B/ST2/00920, UMO-2024/53/N/ST2/00869); Portugal: Foundation for Science and Technology (FCT); Spain: Ministry of Science and Innovation (MCIN & NextGenEU PCI2022-135018-2, MICIN & FEDER PID2021-125273NB, RYC2019-028510-I, RYC2020-030254-I, RYC2021-031273-I, RYC2022-038164-I); Sweden: Carl Trygger Foundation (Carl Trygger Foundation CTS 22:2312), Swedish Research Council (Swedish Research Council 2023-04654, VR 2021-03651, VR 2022-03845, VR 2022-04683, VR 2023-03403, VR 2024-05451), Knut and Alice Wallenberg Foundation (KAW 2018.0458, KAW 2022.0358, KAW 2023.0366); Switzerland: Swiss National Science Foundation (SNSF - PCEFP2_194658); United Kingdom: Leverhulme Trust (Leverhulme Trust RPG-2020-004), Royal Society (NIF-R1-231091); United States of America: U.S. Department of Energy (ECA DE-AC02-76SF00515), Neubauer Family Foundation.

## Author contributions

All authors have contributed to the publication, being variously involved in the design and the construction of the detectors, in writing the software, calibrating subsystems, operating the detectors and acquiring data and finally analysing the processed data. The ATLAS

Collaboration members discussed and approved the scientific results. This Article was prepared by a subgroup of authors appointed by the ATLAS Collaboration and subjected to an internal collaboration-wide review process. All authors reviewed and approved the final version of the paper.

## Competing interests

The authors declare no competing interests.

## Additional information

## The ATLAS Collaboration

G. Aad [1], E. Aakvaag [2], B. Abbott [3], S. Abdelhameed [4], K. Abeling [5], N. J. Abicht [6], S. H. Abidi [7], M. Aboelela [8], A. Aboulhorma [9], H. Abramowicz [10], Y. Abulaiti [11], B. S. Acharya [12,13,242], A. Ackermann [14], C. Adam Bourdarios [15], L. Adamczyk [16], S. V. Addepalli [17], M. J. Addison [18], J. Adelman [19], A. Adiguzel [20], T. Adye [21], A. A. Affolder [22], Y. Afik [23], M. N. Agaras [24], A. Aggarwal [25], C. Agheorghiesei [26], F. Ahmadov [27,243], S. Ahuja [28], X. Ai [29], G. Aielli [30,31], A. Aikot [32], M. Ait Tamlihat [9], B. Aitbenchikh [33], M. Akbiyik [25], T. P. A. Åkesson [34], A. V. Akimov [35], D. Akiyama [36], N. N. Akolkar [37], S. Aktas [38], G. L. Alberghi [39], J. Albert [40], U. Alberti [41], P. Albicocco [42], G. L. Albouy [43], S. Alderweireldt [44], Z. L. Alegria [45], M. Aleksa [46], I. N. Aleksandrov [27], C. Alexa [47], T. Alexopoulos [48], F. Alfonsi [39], M. Algren [49], M. Alhroob [50], B. Ali [51], H. M. J. Ali [52,244], S. Ali [53], S. W. Alibocus [54], M. Aliev [55], G. Alimonti [56], W. Alkakhi [5], C. Allaire [57], B. M. M. Allbrooke [58], J. S. Allen [18], J. F. Allen [44], P. P. Allport [59], A. Aloisio [60,61], F. Alonso [62], C. Alpigiani [63], Z. M. K. Alsolami [52], A. Alvarez Fernandez [25], M. Alves Cardoso [49], M. G. Alviggi [60,61], M. Aly [18], Y. Amaral Coutinho [64], A. Ambler [65], C. Amelung [46], M. Amerl [18], C. G. Ames [66], T. Amezza [67], D. Amidei [68], B. Amini [69], K. Amirie [70], A. Amirkhanov [27], S. P. Amor Dos Santos [71], K. R. Amos [32], D. Amperiadou [72], S. An [73], C. Anastopoulos [74], T. Andeen [75], J. K. Anders [54], A. C. Anderson [76], A. Andreazza [56,77], S. Angelidakis [78], A. Angerami [79], A. V. Anisenkov [27], A. Annovi [80], C. Antel [46], E. Antipov [35], M. Antonelli [42], F. Anulli [81], M. Aoki [73], T. Aoki [82], M. A. Aparo [58], L. Aperio Bella [83], M. Apicella [84], C. Appelt [10], A. Apyan [85], M. Arampatzi [48], S. J. Arbiol Val [86], C. Arcangeletti [42], A. T. H. Arce [87], J-F. Arguin [88], S. Argyropoulos [72], J.-H. Arling [83], O. Arnaez [15], H. Arnold [35], G. Artoni [81,89], H. Asada [90], K. Asai [91], S. Asai [82], S. Asatryan [92], N. A. Asbah [46], R. A. Ashby Pickering [50], A. M. Aslam [28], K. Assamagan [7], R. Astalos [93], K. S. V. Astrand [34], S. Atashi [94], R. J. Atkin [95], H. Atmani[96], P. A. Atmasiddha [97], K. Augsten [51], A. D. Auriol [98], V. A. Austrup [18], G. Avolio [46], K. Axiotis [49], A. Azzam [24], D. Babal [99], H. Bachacou [100], K. Bachas [72,245], A. Bachiu [101], E. Bachmann [102], M. J. Backes [14], A. Badea [23], T. M. Baer [68], P. Bagnaia [81,89], M. Bahmani [103], D. Bahner [69], K. Bai [104], J. T. Baines [21], L. Baines [105], O. K. Baker [106], E. Bakos [107], D. Bakshi Gupta [108], L. E. Balabram Filho [64], V. Balakrishnan [3], R. Balasubramanian [15], E. M. Baldin [109], P. Balek [16], E. Ballabene [39,110], F. Balli [100], L. M. Baltes [14], W. K. Balunas [111], J. Balz [25], I. Bamwidhi [112], E. Banas [86], M. Bandieramonte [113], A. Bandyopadhyay [37], S. Bansal [37], L. Barak [10], M. Barakat [83], E. L. Barberio [114], D. Barberis [115], M. Barbero [1], M. Z. Barel [116], T. Barillari [117], M-S. Barisits [46], T. Barklow [17], P. Baron [118], D. A. Baron Moreno [18], A. Baroncelli [119], A. J. Barr [120], J. D. Barr [121], F. Barreiro [122], J. Barreiro Guimarães da Costa [123], M. G. Barros Teixeira [71], S. Barsov [109], F. Bartels [14], R. Bartoldus [17], A. E. Barton [52], P. Bartos [93], A. Basan [25], M. Baselga [6], S. Bashiri[86], A. Bassalat [57,246], M. J. Basso [124], S. Bataju [8], R. Bate [125], R. L. Bates [76], S. Batlamous[122], M. Battaglia [22], D. Battulga [103], M. Bauce [81,89], M. Bauer [126], P. Bauer [37], L. T. Bayer [83], L. T. Bazzano Hurrell [84], J. B. Beacham [117], T. Beau [67],

J. Y. Beaucamp [62], P. H. Beauchemin [127], P. Bechtle [37], H. P. Beck [41,247], K. Becker [50], A. J. Beddall [128], V. A. Bednyakov [27], C. P. Bee [35], L. J. Beemster [107], M. Begalli [129], M. Begel [7], J. K. Behr [83], J. F. Beirer [46], F. Beisiegel [37], M. Belfkir [112], G. Bella [10], L. Bellagamba [39], A. Bellerive [101], C. D. Bellgraph [130], P. Bellos [59], K. Beloborodov [109], D. Benchekroun [33], F. Bendebba [33], Y. Benhammou [10], K. C. Benkendorfer [131], L. Beresford [83], M. Beretta [42], E. Bergeaas Kuutmann [132], N. Berger [15], B. Bergmann [51], J. Beringer [133], G. Bernardi [134], C. Bernius [17], F. U. Bernlochner [37], F. Bernon [46], A. Berrocal Guardia [24], T. Berry [28], P. Berta [118], A. Berthold [102], A. Berti [71], R. Bertrand [1], S. Bethke [117], A. Betti [81,89], A. J. Bevan [105], L. Bezio [49], N. K. Bhalla [69], S. Bharthuar [117], S. Bhatta [35], P. Bhattarai [17], Z. M. Bhatti [11], K. D. Bhide [69], V. S. Bhopatkar [45], R. M. Bianchi [113], G. Bianco [39,110], O. Biebel [66], M. Biglietti [135], C. S. Billingsley [8], Y. Bimgdi [96], M. Bindi [5], A. Bingham [136], A. Bingul [137], C. Bini [81,89], G. A. Bird [111], M. Birman [138], M. Biros [118], S. Biryukov [58], T. Bisanz [6], E. Bisceglie [39,110], J. P. Biswal [21], D. Biswas [139], I. Bloch [83], A. Blue [76], U. Blumenschein [105], J. Blumenthal [25], V. S. Bobrovnikov [27], L. Boccardo [140,141], M. Boehler [69], B. Boehm [142], D. Bogavac [24], A. G. Bogdanchikov [109], L. S. Boggia [67], V. Boisvert [28], P. Bokan [46], T. Bold [16], M. Bomben [134], M. Bona [105], M. Boonekamp [100], A. G. Borbély [76], I. S. Bordulev [109], G. Borissov [52], D. Bortoletto [120], D. Boscherini [39], M. Bosman [24], K. Bouaouda [33], N. Bouchhar [32], L. Boudet [15], J. Boudreau [113], E. V. Bouhova-Thacker [52], D. Boumediene [98], R. Bouquet [140,141], A. Boveia [143], J. Boyd [46], D. Boye [7], I. R. Boyko [27], L. Bozianu [49], J. Bracinik [59], N. Brahimi [15], G. Brandt [136], O. Brandt [111], B. Brau [144], J. E. Brau [104], R. Brener [138], L. Brenner [116], R. Brenner [132], S. Bressler [138], G. Brianti [145,146], D. Britton [76], D. Britzger [117], I. Brock [37], R. Brock [147], G. Brooijmans [79], A. J. Brooks [130], E. M. Brooks [148], E. Brost [7], L. M. Brown [40,124], L. E. Bruce [131], T. L. Bruckler [120], P. A. Bruckman de Renstrom [86], B. Brüers [83], A. Bruni [39], G. Bruni [39], D. Brunner [149,150], M. Bruschi [39], N. Bruscino [81,89], T. Buanes [2], Q. Buat [63], D. Buchin [117], A. G. Buckley [76], O. Bulekov [128], B. A. Bullard [17], S. Burdin [54], C. D. Burgard [6], A. M. Burger [151], B. Burghgrave [108], O. Burlayenko [69], J. Burleson [152], J. C. Burzynski [153], E. L. Busch [79], V. Büscher [25], P. J. Bussey [76], J. M. Butler [154], C. M. Buttar [76], J. M. Butterworth [121], W. Buttinger [21], C. J. Buxo Vazquez [147], A. R. Buzykaev [27], S. Cabrera Urbán [32], L. Cadamuro [57], D. Caforio [155], H. Cai [113], Y. Cai [39,110,156], Y. Cai [157], V. M. M. Cairo [46], O. Cakir [158], N. Calace [46], P. Calafiura [133], G. Calderini [67], P. Calfayan [101], L. Calic [34], G. Callea [76], L. P. Caloba [64], D. Calvet [98], S. Calvet [98], R. Camacho Toro [67], S. Camarda [46], D. Camarero Munoz [85], P. Camarri [30,31], C. Camincher [40], M. Campanelli [121], A. Camplani [159], V. Canale [60,61], A. C. Canbay [158], E. Canonero [28], J. Cantero [32], Y. Cao [152], F. Capocasa [85], M. Capua [160,161], A. Carbone [56,77], R. Cardarelli [30], J. C. J. Cardenas [108], M. P. Cardiff [85], G. Carducci [160,161], T. Carli [46], G. Carlino [60], J. I. Carlotto [24], B. T. Carlson [113,248], E. M. Carlson [40], J. Carmignani [54], L. Carminati [56,77], A. Carnelli [15], M. Carnesale [46], S. Caron [162], E. Carquin [163], I. B. Carr [114], S. Carrá [164,165], G. Carratta [39,110], A. M. Carroll [104], M. P. Casado [24,249], P. Casolaro [60,61], M. Caspar [83], F. L. Castillo [15], L. Castillo Garcia [24], V. Castillo Gimenez [32], N. F. Castro [71,166], A. Catinaccio [46], J. R. Catmore [167], T. Cavaliere [15], V. Cavaliere [7], L. J. Caviedes Betancourt [168], E. Celebi [128], S. Cella [46], V. Cepaitis [49], K. Cerny [169], A. S. Cerqueira [170], A. Cerri [80,171,250], L. Cerrito [30,31], F. Cerutti [133], B. Cervato [56,77], A. Cervelli [39], G. Cesarini [42], S. A. Cetin [128], P. M. Chabrillat [67], R. Chakkappai [57], S. Chakraborty [50], J. Chan [133], W. Y. Chan [82], J. D. Chapman [111], E. Chapon [100], B. Chargeishvili [172], D. G. Charlton [59], C. Chauhan [118], Y. Che [157], S. Chekanov [173], S. V. Chekulaev [124], G. A. Chelkov [27,251], B. Chen [10], B. Chen [40], H. Chen [157], H. Chen [7], J. Chen [174], J. Chen [153], M. Chen [120], S. Chen [175], S. J. Chen [157], X. Chen [174], X. Chen [176,252], Z. Chen [119], C. L. Cheng [177], H. C. Cheng [178], S. Cheong [17], A. Cheplakov [27], E. Cherepanova [116], R. Cherkaoui El Moursli [9], E. Cheu [179], K. Cheung [180], L. Chevalier [100], V. Chiarella [42], G. Chiarelli [80], G. Chiodini [181], A. S. Chisholm [59], A. Chitan [47], M. Chitishvili [32], M. V. Chizhov [27,253], K. Choi [75], Y. Chou [63], E. Y. S. Chow [162], K. L. Chu [138], M. C. Chu [178], X. Chu [123,156], Z. Chubinidze [42], J. Chudoba [182], J. J. Chwastowski [86], D. Cieri [117], K. M. Ciesla [16], V. Cindro [183], A. Ciocio [133], F. Cirotto [60,61], Z. H. Citron [138], M. Citterio [56], D. A. Ciubotaru [47], A. Clark [49], P. J. Clark [44], N. Clarke Hall [121], C. Clarry [70], S. E. Clawson [83], C. Clement [149,150], Y. Coadou [1], M. Cobal [12,184], A. Coccaro [141], R. F. Coelho Barrue [71], R. Coelho Lopes De Sa [144], S. Coelli [56], L. S. Colangeli [70], B. Cole [79], P. Collado Soto [122], J. Collot [43], R. Coluccia [181,185], P. Conde Muiño [71,186], M. P. Connell [55], S. H. Connell [55], E. I. Conroy [120], M. Contreras Cossio [75], F. Conventi [60,254], A. M. Cooper-Sarkar [120], L. Corazzina [81,89], F. A. Corchia [39,110], A. Cordeiro Oudot Choi [63], L. D. Corpe [98], M. Corradi [81,89], F. Corriveau [65,255], A. Cortes-Gonzalez [82], M. J. Costa [32], F. Costanza [15], D. Costanzo [74], B. M. Cote [143], J. Couthures [15], G. Cowan [28], K. Cranmer [177], L. Cremer [6], D. Cremonini [39,110], S. Crépé-Renaudin [43], F. Crescioli [67], T. Cresta [164,165], M. Cristinziani [139], M. Cristoforetti [145,146], V. Croft [116], J. E. Crosby [45], G. Crosetti [160,161], A. Cueto [122], H. Cui [121], Z. Cui [179], B. M. Cunnett [58], W. R. Cunningham [76], F. Curcio [32], J. R. Curran [44], M. J. Da Cunha Sargedas De Sousa [140,141], J. V. Da Fonseca Pinto [64], C. Da Via [18], W. Dabrowski [16], T. Dado [46], S. Dahbi [187], T. Dai [68], D. Dal Santo [41], C. Dallapiccola [144], M. Dam [159], G. D'amen [7], V. D'Amico [66],

J. Damp [25], J. R. Dandoy [101], M. D'Andrea [140], D. Dannheim [46], G. D'anniballe [80,171], M. Danninger [153], V. Dao [35], G. Darbo [141], S. J. Das [7], F. Dattola [83], S. D'Auria [56,77], A. D'Avanzo [60,61], T. Davidek [118], J. Davidson [50], I. Dawson [105], K. De [108], C. De Almeida Rossi [70], R. De Asmundis [60], N. De Biase [83], S. De Castro [39,110], N. De Groot [162], P. de Jong [116], H. De la Torre [19], A. De Maria [157], A. De Salvo [81], U. De Sanctis [30,31], F. De Santis [181,185], A. De Santo [58], J. B. De Vivie De Regie [43], J. Debevc [183], D. V. Dedovich [27], J. Degens [54], A. M. Deiana [8], J. Del Peso [122], L. Delagrange [67], F. Deliot [100], C. M. Delitzsch [6], M. Della Pietra [60,61], D. Della Volpe [49], A. Dell'Acqua [46], L. Dell'Asta [56,77], M. Delmastro [15], C. C. Delogu [25], P. A. Delsart [43], S. Demers [106], M. Demichev [27], S. P. Denisov [109], H. Denizli [38,256], L. D'Eramo [98], D. Derendarz [86], F. Derue [67], P. Dervan [54], A. M. Desai [188], K. Desch [37], F. A. Di Bello [140,141], A. Di Ciaccio [30,31], L. Di Ciaccio [15], A. Di Domenico [81,89], C. Di Donato [60,61], A. Di Girolamo [46], G. Di Gregorio [46], A. Di Luca [145,146], B. Di Micco [135,189], R. Di Nardo [135,189], K. F. Di Petrillo [23], M. Diamantopoulou [101], F. A. Dias [116], M. A. Diaz [190,191], A. R. Didenko [27], M. Didenko [32], S. D. Diefenbacher [133], E. B. Diehl [68], S. Díez Cornell [83], C. Diez Pardos [139], C. Dimitriadi [192], A. Dimitrievska [59], A. Dimri [35], Y. Ding [119], J. Dingfelder [37], T. Dingley [120], I-M. Dinu [47], S. J. Dittmeier [193], F. Dittus [46], M. Divisek [118], B. Dixit [54], F. Djama [1], T. Djobava [172], C. Doglioni [18,34], A. Dohnalova [93], Z. Dolezal [118], K. Domijan [16], K. M. Dona [23], M. Donadelli [129], B. Dong [147], J. Donini [98], A. D'Onofrio [60,61], M. D'Onofrio [54], J. Dopke [21], A. Doria [60], N. Dos Santos Fernandes [71], I. A. Dos Santos Luz [194], P. Dougan [18], M. T. Dova [62], A. T. Doyle [76], M. A. Draguet [120], M. P. Drescher [5], E. Dreyer [138], I. Drivas-koulouris [48], M. Drnevich [11], M. Drozdova [49], D. Du [119], T. A. du Pree [116], Z. Duan [157], M. Dubau [15], F. Dubinin [27], M. Dubovsky [93], E. Duchovni [138], G. Duckeck [66], P. K. Duckett [121], O. A. Ducu [47], D. Duda [44], A. Dudarev [46], E. R. Duden [85], M. D'uffizi [18], L. Duflot [57], M. Dührssen [46], I. Duminica [195], A. E. Dumitriu [47], M. Dunford [14], S. Dungs [6], K. Dunne [149,150], A. Duperrin [1], H. Duran Yildiz [158], M. Düren [155], A. Durglishvili [172], D. Duvnjak [101], G. I. Dyckes [133], M. Dyndal [16], B. S. Dziedzic [46], Z. O. Earnshaw [58], G. H. Eberwein [120], B. Eckerova [93], S. Eggebrecht [5], E. Egidio Purcino De Souza [194], G. Eigen [2], K. Einsweiler [133], T. Ekelof [132], P. A. Ekman [34], S. El Farkh [196], Y. El Ghazali [119], H. El Jarrari [46], A. El Moussaouy [33], V. Ellajosyula [132], M. Ellert [132], F. Ellinghaus [136], T. A. Elliot [28], N. Ellis [46], J. Elmsheuser [7], M. Elsawy [4], M. Elsing [46], D. Emeliyanov [21], Y. Enari [73], I. Ene [133], S. Epari [88], D. Ernani Martins Neto [86], F. Ernst [46], M. Errenst [136], M. Escalier [57], C. Escobar [32], E. Etzion [10], G. Evans [71,197], H. Evans [130], L. S. Evans [28], A. Ezhilov [109], S. Ezzarqtouni [33], F. Fabbri [39,110], L. Fabbri [39,110], G. Facini [121], V. Fadeyev [22], R. M. Fakhrutdinov [109], D. Fakoudis [25], S. Falciano [81], L. F. Falda Ulhoa Coelho [71], F. Fallavollita [117], G. Falsetti [160,161], J. Faltova [118], C. Fan [152], K. Y. Fan [198], Y. Fan [123], Y. Fang [123,156], M. Fanti [56,77], M. Faraj [12,13], Z. Farazpay [199], A. Farbin [108], A. Farilla [135], K. Farman [187], T. Farooque [147], J. N. Farr [106], S. M. Farrington [21,44], F. Fassi [9], D. Fassouliotis [78], L. Fayard [57], P. Federic [118], P. Federicova [182], O. L. Fedin [109,251], M. Feickert [177], L. Feligioni [1], D. E. Fellers [133], C. Feng [200], Z. Feng [116], M. J. Fenton [94], L. Ferencz [83], B. Fernandez Barbadillo [52], P. Fernandez Martinez [201], M. J. V. Fernoux [1], J. Ferrando [52], A. Ferrari [132], P. Ferrari [116,162], R. Ferrari [164], D. Ferrere [49], C. Ferretti [68], M. P. Fewell [188], D. Fiacco [81,89], F. Fiedler [25], P. Fiedler [51], S. Filimonov [27], M. S. Filip [47,257], A. Filipčič [183], E. K. Filmer [124], F. Filthaut [162], M. C. N. Fiolhais [71,202,258], L. Fiorini [32], W. C. Fisher [147], T. Fitschen [18], P. M. Fitzhugh [100], I. Fleck [139], P. Fleischmann [68], T. Flick [136], M. Flores [203,259], L. R. Flores Castillo [178], L. Flores Sanz De Acedo [46], F. M. Follega [145,146], N. Fomin [111], J. H. Foo [70], A. Formica [100], A. C. Forti [18], E. Fortin [46], A. W. Fortman [133], L. Foster [133], L. Fountas [78,260], D. Fournier [57], H. Fox [52], P. Francavilla [80,171], S. Francescato [131], S. Franchellucci [49], M. Franchini [39,110], S. Franchino [14], D. Francis [46], L. Franco [162], V. Franco Lima [46], L. Franconi [83], M. Franklin [131], G. Frattari [85], Y. Y. Frid [10], J. Friend [76], N. Fritzsche [46], A. Froch [49], D. Froidevaux [46], J. A. Frost [120], Y. Fu [147], S. Fuenzalida Garrido [163], M. Fujimoto [1], K. Y. Fung [178], E. Furtado De Simas Filho [194], M. Furukawa [82], J. Fuster [32], A. Gaa [5], A. Gabrielli [39,110], A. Gabrielli [70], P. Gadow [46], G. Gagliardi [140,141], L. G. Gagnon [133], S. Gaid [204], S. Galantzan [10], J. Gallagher [188], E. J. Gallas [120], A. L. Gallen [132], B. J. Gallop [21], K. K. Gan [143], S. Ganguly [82], Y. Gao [44], A. Garabaglu [63], F. M. Garay Walls [190,191], C. García [32], A. Garcia Alonso [116], A. G. Garcia Caffaro [106], J. E. García Navarro [32], M. A. Garcia Ruiz [168], M. Garcia-Sciveres [133], G. L. Gardner [97], R. W. Gardner [23], N. Garelli [127], R. B. Garg [17], J. M. Gargan [44], C. A. Garner [70], C. M. Garvey [95], V. K. Gassmann [127], G. Gaudio [164], V. Gautam [24], P. Gauzzi [81,89], J. Gavranovic [183], I. L. Gavrilenko [71], A. Gavrilyuk [109], C. Gay [125], G. Gaycken [104], E. N. Gazis [48], A. Gekow [143], C. Gemme [141], M. H. Genest [43], A. D. Gentry [205], S. George [28], T. Geralis [206], A. A. Gerwin [3], P. Gessinger-Befurt [46], M. E. Geyik [136], M. Ghani [50], K. Ghorbanian [105], A. Ghosal [139], A. Ghosh [94], A. Ghosh [179], B. Giacobbe [39], S. Giagu [81,89], T. Giani [116], A. Giannini [119], S. M. Gibson [28], M. Gignac [22], D. T. Gil [207], A. K. Gilbert [16], B. J. Gilbert [79], D. Gillberg [101], G. Gilles [116], D. M. Gingrich [208,261], M. P. Giordani [12,184], P. F. Giraud [100], G. Giugliarelli [12,184], D. Giugni [56], F. Giuli [30,31], I. Gkialas [78,260], L. K. Gladilin [109], C. Glasman [122], M. Glazewska [41], R. M. Gleason [94], G. Glemža [83], M. Glisic [104], I. Gnesi [161], Y. Go [7], M. Goblirsch-Kolb [46],

B. Gocke [6], D. Godin[88], B. Gokturk [38], S. Goldfarb [114], T. Golling [49], M. G. D. Gololo [55], D. Golubkov [109], J. P. Gombas [147], A. Gomes [71,197], G. Gomes Da Silva [139], A. J. Gomez Delegido [32], R. Gonçalo [71], L. Gonella [59], A. Gongadze [209], F. Gonnella [59], J. L. Gonski [17], R. Y. González Andana [44], S. González de la Hoz [32], M. V. Gonzalez Rodrigues [83], R. Gonzalez Suarez [132], S. Gonzalez-Sevilla [49], L. Goossens [46], B. Gorini [46], E. Gorini [181,185], A. Gorišek [183], T. C. Gosart [97], A. T. Goshaw [87], M. I. Gostkin [27], S. Goswami [45], C. A. Gottardo [46], S. A. Gotz [66], M. Gouighri [196], A. G. Goussiou [63], N. Govender [55], R. P. Grabarczyk [120], I. Grabowska-Bold [16], K. Graham [101], E. Gramstad [167], S. Grancagnolo [181,185], C. M. Grant[188], P. M. Gravila [210], F. G. Gravili [181,185], H. M. Gray [133], M. Greco [117], M. J. Green [188], C. Grefe [37], A. S. Grefsrud [2], I. M. Gregor [83], K. T. Greif [94], P. Grenier [17], S. G. Grewe[117], A. A. Grillo [22], K. Grimm [53], S. Grinstein [24,262], J.-F. Grivaz [57], E. Gross [138], J. Grosse-Knetter [5], L. Guan [68], G. Guerrieri [46], D. Guest [103], R. Guevara [167], R. Gugel [25], J. A. M. Guhit [68], A. Guida [103], E. Guilloton [50], S. Guindon [46], F. Guo [123,156], J. Guo [174], L. Guo [83], L. Guo [211,263], Y. Guo [68], A. Gupta [6], R. Gupta [113], S. Gupta [85], S. Gurbuz [37], S. S. Gurdasani [83], G. Gustavino [81,89], P. Gutierrez [3], L. F. Gutierrez Zagazeta [97], M. Gutsche [102], C. Gutschow [121], C. Gwenlan [120], C. B. Gwilliam [54], E. S. Haaland [167], A. Haas [11], M. Habedank [76], C. Haber [133], H. K. Hadavand [108], A. Haddad [98], A. Hadef [102], A. I. Hagan [52], J. J. Hahn [139], E. H. Haines [121], M. Haleem [142], J. Haley [45], G. D. Hallewell [1], L. Halser [41], K. Hamano [40], H. Hamdaoui [132], M. Hamer [37], S. E. D. Hammoud [57], E. J. Hampshire [28], J. Han [200], L. Han [157], L. Han [119], S. Han [123], K. Hanagaki [73], M. Hance [22], D. A. Hangal [79], H. Hanif [153], M. D. Hank [97], J. B. Hansen [159], P. H. Hansen [159], D. Harada [49], T. Harenberg [136], S. Harkusha [92], M. L. Harris [144], Y. T. Harris [37], J. Harrison [24], N. M. Harrison [143], P. F. Harrison[50], M. L. E. Hart [121], N. M. Hartman [117], N. M. Hartmann [66], R. Z. Hasan [21,28], Y. Hasegawa [212], F. Haslbeck [120], S. Hassan [2], R. Hauser [147], M. Haviernik [118], C. M. Hawkes [59], R. J. Hawkings [46], Y. Hayashi [82], D. Hayden [147], C. Hayes [68], R. L. Hayes [116], C. P. Hays [120], J. M. Hays [105], H. S. Hayward [54], M. He [123,156], Y. He [83], Y. He [121], N. B. Heatley [105], V. Hedberg [34], C. Heidegger [69], K. K. Heidegger [69], J. Heilman [101], S. Heim [83], T. Heim [133], J. G. Heinlein [97], J. J. Heinrich [104], L. Heinrich [117], J. Hejbal [182], M. Helbig [102], A. Held [177], S. Hellesund [2], C. M. Helling [125], S. Hellman [149,150], A. M. Henriques Correia[46], H. Herde [34], Y. Hernández Jiménez [35], L. M. Herrmann [37], T. Herrmann [102], G. Herten [69], R. Hertenberger [66], L. Hervas [46], M. E. Hesping [25], N. P. Hessey [124], J. Hessler [117], M. Hidaoui [196], N. Hidic [118], E. Hill [70], T. S. Hillersoy [2], S. J. Hillier [59], J. R. Hinds [147], F. Hinterkeuser [37], M. Hirose [213], S. Hirose [214], D. Hirschbuehl [136], T. G. Hitchings [18], B. Hiti [183], J. Hobbs [35], R. Hobincu [215], N. Hod [138], A. M. Hodges [152], M. C. Hodgkinson [74], B. H. Hodkinson [120], A. Hoecker [46], D. D. Hofer [68], J. Hofer [32], M. Holzbock [46], L. B. A. H. Hommels [111], V. Homsak [120], B. P. Honan [18], J. J. Hong [130], T. M. Hong [113], B. H. Hooberman [152], W. H. Hopkins [173], M. C. Hoppesch [152], Y. Horii [90], M. E. Horstmann [117], S. Hou [187], M. R. Housenga [152], A. S. Howard [183], J. Howarth [76], J. Hoya [173], M. Hrabovsky [169], T. Hryn'ova [15], P. J. Hsu [180], S.-C. Hsu [63], T. Hsu [57], M. Hu [133], Q. Hu [119], S. Huang [111], X. Huang [123,156], Y. Huang [118], Y. Huang [211], Y. Huang [25], Y. Huang [123], Z. Huang [57], Z. Hubacek [51], M. Huebner [37], F. Huegging [37], T. B. Huffman [120], M. Hufnagel Maranha De Faria [170], C. A. Hugli [83], M. Huhtinen [46], S. K. Huiberts [2], R. Hulsken [65], C. E. Hultquist [133], D. L. Humphreys [144], N. Huseynov [216], J. Huston [147], J. Huth [131], R. Hyneman [179], G. Iacobucci [49], G. Iakovidis [7], L. Iconomidou-Fayard [57], J. P. Iddon [46], P. Iengo [60,61], R. Iguchi [82], Y. Iiyama [82], T. Iizawa [82], Y. Ikegami [73], D. Iliadis [72], N. Ilic [70], H. Imam [33], G. Inacio Goncalves [129], S. A. Infante Cabanas [217], T. Ingebretsen Carlson [149,150], J. M. Inglis [105], G. Introzzi [164,165], M. Iodice [135], V. Ippolito [81,89], R. K. Irwin [54], M. Ishino [82], W. Islam [177], C. Issever [103], S. Istin [38,264], K. Itabashi [73], H. Ito [36], R. Iuppa [145,146], A. Ivina [138], V. Izzo [60], P. Jacka [51], P. Jackson [188], P. Jain [83], K. Jakobs [69], T. Jakoubek [138], J. Jamieson [76], W. Jang [82], S. Jankovych [118], M. Javurkova [144], P. Jawahar [18], L. Jeanty [104], J. Jejelava [218,265], P. Jenni [69,266], C. E. Jessiman [101], C. Jia [200], H. Jia [125], J. Jia [35], X. Jia [123,156], Z. Jia [157], C. Jiang [44], Q. Jiang [198], S. Jiggins [83], M. Jimenez Ortega [32], J. Jimenez Pena [24], S. Jin [157], A. Jinaru [47], O. Jinnouchi [219], P. Johansson [74], K. A. Johns [179], J. W. Johnson [22], F. A. Jolly [83], D. M. Jones [58], E. Jones [83], K. S. Jones[108], P. Jones [111], R. W. L. Jones [52], T. J. Jones [54], H. L. Joos [5,46], R. Joshi [143], J. Jovicevic [107], X. Ju [133], J. J. Junggeburth [46], T. Junkermann [14], A. Juste Rozas [24,262], M. K. Juzek [86], S. Kabana [220], A. Kaczmarska [86], M. Kado [117], H. Kagan [143], M. Kagan [17], A. Kahn [97], C. Kahra [25], T. Kaji [82], E. Kajomovitz [221], N. Kakati [138], N. Kakoty [24], I. Kalaitzidou [69], S. Kandel [108], N. J. Kang [22], D. Kar [222], K. Karava [120], E. Karentzos [37], O. Karkout [116], S. N. Karpov [27], Z. M. Karpova [27], V. Kartvelishvili [52], A. N. Karyukhin [109], E. Kasimi [72], J. Katzy [83], S. Kaur [101], K. Kawade [212], M. P. Kawale [3], C. Kawamoto [175], T. Kawamoto [119], E. F. Kay [46], F. I. Kaya [127], S. Kazakos [147], V. F. Kazanin [109], J. M. Keaveney [95], R. Keeler [40], G. V. Kehris [131], J. S. Keller [101], J. M. Kelly [40], J. J. Kempster [58], O. Kepka [182], J. Kerr [148], B. P. Kerridge [21], B. P. Kerševan [183], L. Keszeghova [93], R. A. Khan [113], A. Khanov [45], A. G. Kharlamov [109], T. Kharlamova [109], E. E. Khoda [63], M. Kholodenko [71], T. J. Khoo [103], G. Khoriauli [142], Y. Khoulaki [33], J. Khubua [172,281], Y. A. R. Khwaira [67], B. Kibirige[222], D. Kim [173], D. W. Kim [149,150],

Y. K. Kim [23], N. Kimura [121], M. K. Kingston [5], A. Kirchhoff [5], C. Kirfel [37], F. Kirfel [37], J. Kirk [21], A. E. Kiryunin [117], S. Kita [214], O. Kivernyk [37], M. Klassen [127], C. Klein [101], L. Klein [142], M. H. Klein [8], S. B. Klein [49], U. Klein [54], A. Klimentov [7], T. Klioutchnikova [46], P. Kluit [116], S. Kluth [117], E. Kneringer [126], T. M. Knight [70], A. Knue [6], M. Kobel [102], D. Kobylianskii [138], S. F. Koch [120], M. Kocian [17], P. Kodyš [118], D. M. Koeck [104], T. Koffas [101], O. Kolay [102], I. Koletsou [15], T. Komarek [86], K. Köneke [5], A. X. Y. Kong [188], T. Kono [91], N. Konstantinidis [121], P. Kontaxakis [49], B. Konya [34], R. Kopeliansky [79], S. Koperny [16], K. Korcyl [86], K. Kordas [72,267], A. Korn [121], S. Korn [5], I. Korolkov [24], N. Korotkova [109], B. Kortman [116], O. Kortner [117], S. Kortner [117], W. H. Kostecka [19], M. Kostov [93], V. V. Kostyukhin [139], A. Kotsokechagia [46], A. Kotwal [87], A. Koulouris [46], A. Kourkoumeli-Charalampidi [164,165], C. Kourkoumelis [78], E. Kourlitis [117], O. Kovanda [104], R. Kowalewski [40], W. Kozanecki [104], A. S. Kozhin [109], V. A. Kramarenko [109], G. Kramberger [183], P. Kramer [37], M. W. Krasny [67], A. Krasznahorkay [144], A. C. Kraus [19], J. W. Kraus [136], J. A. Kremer [83], N. B. Krengel [139], T. Kresse [102], L. Kretschmann [136], J. Kretzschmar [54], P. Krieger [70], K. Krizka [59], K. Kroeninger [6], H. Kroha [117], J. Kroll [182], J. Kroll [97], K. S. Krowpman [147], U. Kruchonak [27], H. Krüger [37], N. Krumnack [223], M. C. Kruse [87], O. Kuchinskaia [27], S. Kuday [158], S. Kuehn [46], R. Kuesters [69], T. Kuhl [83], V. Kukhtin [27], Y. Kulchitsky [27], S. Kuleshov [191,224], J. Kull [188], E. V. Kumar [66], M. Kumar [222], N. Kumari [83], P. Kumari [148], A. Kupco [182], T. Kupfer [6], A. Kupich [109], O. Kuprash [69], H. Kurashige [225], L. L. Kurchaninov [124], O. Kurdysh [15], Y. A. Kurochkin [109], A. Kurova [109], M. Kuze [219], A. K. Kvam [144], J. Kvita [169], N. G. Kyriacou [68], C. Lacasta [32], F. Lacava [81,89], H. Lacker [103], D. Lacour [67], N. N. Lad [121], E. Ladygin [27], A. Lafarge [98], B. Laforge [67], T. Lagouri [106], F. Z. Lahbabi [33], S. Lai [5], W. S. Lai [121], J. E. Lambert [40], S. Lammers [130], W. Lampl [179], C. Lampoudis [72,267], G. Lamprinoudis [25], A. N. Lancaster [19], E. Lançon [7], U. Landgraf [69], M. P. J. Landon [105], V. S. Lang [69], O. K. B. Langrekken [167], A. J. Lankford [94], F. Lanni [46], K. Lantzsch [37], A. Lanza [164], M. Lanzac Berrocal [32], J. F. Laporte [100], T. Lari [56], D. Larsen [2], L. Larson [75], F. Lasagni Manghi [39], M. Lassnig [46], S. D. Lawlor [74], R. Lazaridou [94], A. Lazzaroni [56,77], H. D. M. Le [147], E. M. Le Boulicaut [106], L. T. Le Pottier [133], B. Leban [39,110], F. Ledroit-Guillon [43], T. F. Lee [148], L. L. Leeuw [55], M. Lefebvre [40], C. Leggett [133], G. Lehmann Miotto [46], M. Leigh [49], W. A. Leight [144], W. Leinonen [162], A. Leisos [72,268], M. A. L. Leite [226], C. E. Leitgeb [103], R. Leitner [118], K. J. C. Leney [8], T. Lenz [37], S. Leone [80], C. Leonidopoulos [44], A. Leopold [192], J. H. Lepage Bourbonnais [101], R. Les [147], C. G. Lester [111], M. Levchenko [109], J. Levêque [15], L. J. Levinson [138], G. Levrini [39,110], M. P. Lewicki [86], C. Lewis [63], D. J. Lewis [15], L. Lewitt [74], A. Li [7], B. Li [200], C. Li [68], C-Q. Li [117], H. Li [200], H. Li [18], H. Li [176], H. Li [119], H. Li [200], J. Li [174], K. Li [123], L. Li [174], R. Li [106], S. Li [123,156], S. Li [174,227], T. Li [134], X. Li [65], Z. Li [82], Z. Li [123,156], Z. Li [119], S. Liang [123,156], Z. Liang [123], M. Liberatore [100], B. Liberti [30], K. Lie [228], J. Lieber Marin [194], H. Lien [130], H. Lin [68], S. F. Lin [35], L. Linden [66], R. E. Lindley [179], J. H. Lindon [46], J. Ling [131], E. Lipeles [97], A. Lipniacka [2], A. Lister [125], J. D. Little [130], B. Liu [123], B. X. Liu [211], D. Liu [174,227], D. Liu [22], E. H. L. Liu [59], J. K. K. Liu [11], K. Liu [227], K. Liu [174,227], M. Liu [119], M. Y. Liu [119], P. Liu [123], Q. Liu [63,174,227], X. Liu [119], X. Liu [200], Y. Liu [156,211], Y. L. Liu [200], Y. W. Liu [119], Z. Liu [57,269], S. L. Lloyd [105], E. M. Lobodzinska [83], P. Loch [179], E. Lodhi [70], T. Lohse [103], K. Lohwasser [74], E. Loiacono [83], J. D. Lomas [59], J. D. Long [79], I. Longarini [94], R. Longo [152], A. Lopez Solis [24], N. A. Lopez-canelas [179], N. Lorenzo Martinez [15], A. M. Lory [66], M. Losada [4], G. Löschcke Centeno [58], X. Lou [149,150], X. Lou [123,156], A. Lounis [57], P. A. Love [52], M. Lu [57], S. Lu [97], Y. J. Lu [187], H. J. Lubatti [63], C. Luci [81,89], F. L. Lucio Alves [157], F. Luehring [130], B. S. Lunday [97], O. Lundberg [192], J. Lunde [46], N. A. Luongo [173], M. S. Lutz [46], A. B. Lux [154], D. Lynn [7], R. Lysak [182], V. Lysenko [51], E. Lytken [34], V. Lyubushkin [27], T. Lyubushkina [27], M. M. Lyukova [35], M. Firdaus M. Soberi [44], H. Ma [7], K. Ma [119], L. L. Ma [200], W. Ma [119], Y. Ma [45], J. C. MacDonald [25], P. C. Machado De Abreu Farias [194], R. Madar [98], T. Madula [121], J. Maeda [225], T. Maeno [7], P. T. Mafa [55,270], H. Maguire [74], M. Maheshwari [111], V. Maiboroda [57], A. Maio [71,197,229], K. Maj [16], O. Majersky [83], S. Majewski [104], R. Makhmanazarov [109], N. Makovec [57], V. Maksimovic [107], B. Malaescu [67], J. Malamant [167], Pa. Malecki [86], V. P. Maleev [109], F. Malek [43,271], M. Mali [183], D. Malito [28], U. Mallik [230,282], A. Maloizel [134], S. Maltezos [48], A. Malvezzi Lopes [129], S. Malyukov [27], J. Mamuzic [24], G. Mancini [42], M. N. Mancini [85], G. Manco [164,165], J. P. Mandalia [105], S. S. Mandarry [58], I. Mandić [183], L. Manhaes de Andrade Filho [170], I. M. Maniatis [138], J. Manjarres Ramos [151], D. C. Mankad [138], A. Mann [66], T. Manoussos [46], M. N. Mantinan [23], S. Manzoni [46], L. Mao [174], X. Mapekula [55], A. Marantis [72], R. R. Marcelo Gregorio [105], G. Marchiori [134], M. Marcisovsky [182], C. Marcon [56], E. Maricic [107], M. Marinescu [83], S. Marium [83], M. Marjanovic [3], A. Markhoos [69], M. Markovitch [57], M. K. Maroun [144], G. T. Marsden [18], E. J. Marshall [52], Z. Marshall [133], S. Marti-Garcia [32], J. Martin [121], T. A. Martin [21], V. J. Martin [44], B. Martin dit Latour [2], L. Martinelli [81,89], M. Martinez [24,262], P. Martinez Agullo [32], V. I. Martinez Outschoorn [144], P. Martinez Suarez [24], S. Martin-Haugh [21], G. Martinovicova [118], V. S. Martoiu [47], A. C. Martyniuk [121], A. Marzin [46], D. Mascione [145,146], L. Masetti [25], J. Masik [18], A. L. Maslennikov [27], S. L. Mason [79], P. Massarotti [60,61], P. Mastrandrea [80,171], A. Mastroberardino [160,161], T. Masubuchi [213],

T. T. Mathew [104], J. Matousek [118], D. M. Mattern [6], J. Maurer [47], T. Maurin [76], A. J. Maury [57], B. Maček [183], C. Mavungu Tsava [1], D. A. Maximov [109], A. E. May [18], E. Mayer [98], R. Mazini [222], I. Maznas [19], S. M. Mazza [22], E. Mazzeo [46], J. P. Mc Gowan [40], S. P. Mc Kee [68], C. A. Mc Lean [173], C. C. McCracken [125], E. F. McDonald [114], A. E. McDougall [116], L. F. Mcelhinney [52], J. A. Mcfayden [58], R. P. McGovern [97], R. P. Mckenzie [222], T. C. Mclachlan [83], D. J. Mclaughlin [121], S. J. McMahon [21], C. M. Mcpartland [54], R. A. McPherson [40,255], S. Mehlhase [66], A. Mehta [54], D. Melini [32], B. R. Mellado Garcia [222], A. H. Melo [5], F. Meloni [83], A. M. Mendes Jacques Da Costa [18], L. Meng [52], S. Menke [117], M. Mentink [46], E. Meoni [160,161], G. Mercado [19], S. Merianos [72], C. Merlassino [12,184], C. Meroni [56,77], J. Metcalfe [173], A. S. Mete [173], E. Meuser [25], C. Meyer [130], J-P. Meyer [100], Y. Miao [157], R. P. Middleton [21], M. Mihovilovic [57], L. Mijović [44], G. Mikenberg [138], M. Mikestikova [182], M. Mikuž [183], H. Mildner [25], A. Milic [46], D. W. Miller [23], E. H. Miller [17], L. S. Miller [101], A. Milov [138], D. A. Milstead [149,150], T. Min [157], A. A. Minaenko [109], I. A. Minashvili [172], A. I. Mincer [11], B. Mindur [16], M. Mineev [27], Y. Mino [175], L. M. Mir [24], M. Miralles Lopez [76], M. Mironova [133], M. C. Missio [162], A. Mitra [50], V. A. Mitsou [32], Y. Mitsumori [90], O. Miu [70], P. S. Miyagawa [105], T. Mkrtchyan [14], M. Mlinarevic [121], T. Mlinarevic [121], M. Mlynarikova [46], S. Mobius [41], M. H. Mohamed Farook [205], S. Mohapatra [79], S. Mohiuddin [45], G. Mokgatitswane [222], L. Moleri [138], U. Molinatti [120], L. G. Mollier [41], B. Mondal [182], S. Mondal [51], K. Mönig [83], E. Monnier [1], L. Monsonis Romero [32], J. Montejo Berlingen [24], A. Montella [149,150], M. Montella [143], F. Montereali [135,189], F. Monticelli [62], S. Monzani [12,184], A. Morancho Tarda [159], N. Morange [57], A. L. Moreira De Carvalho [83], M. Moreno Llácer [32], C. Moreno Martinez [49], J. M. Moreno Perez [168], P. Morettini [141], S. Morgenstern [46], M. Morii [131], M. Morinaga [82], M. Moritsu [231], F. Morodei [81,89], P. Moschovakos [46], B. Moser [69], M. Mosidze [172], T. Moskalets [8], P. Moskvitina [162], J. Moss [53], P. Moszkowicz [16], A. Moussa [232], Y. Moyal [138], H. Moyano Gomez [24], E. J. W. Moyse [144], T. G. Mroz [86], O. Mtintsilana [222], S. Muanza [1], M. Mucha [37], J. Mueller [113], R. Müller [46], G. A. Mullier [132], A. J. Mullin [111], J. J. Mullin [87], A. C. Mullins [8], A. E. Mulski [131], D. P. Mungo [70], D. Munoz Perez [32], F. J. Munoz Sanchez [18], W. J. Murray [21,50], M. Muškinja [183], C. Mwewa [83], A. G. Myagkov [109,251], A. J. Myers [108], G. Myers [68], M. Myska [51], B. P. Nachman [133], K. Nagai [120], K. Nagano [73], R. Nagasaka [82], J. L. Nagle [7,272], E. Nagy [1], A. M. Nairz [46], Y. Nakahama [73], K. Nakamura [73], K. Nakkalil [134], A. Nandi [193], H. Nanjo [213], E. A. Narayanan [8], Y. Narukawa [82], I. Naryshkin [109], L. Nasella [56,77], S. Nasri [112], C. Nass [37], G. Navarro [233], J. Navarro-Gonzalez [32], A. Nayaz [103], P. Y. Nechaeva [109], S. Nechaeva [39,110], F. Nechansky [182], L. Nedic [120], T. J. Neep [59], A. Negri [164,165], M. Negrini [39], C. Nellist [116], C. Nelson [65], K. Nelson [68], S. Nemecek [182], M. Nessi [46,273], M. S. Neubauer [152], J. Newell [54], P. R. Newman [59], Y. W. Y. Ng [152], B. Ngair [4], H. D. N. Nguyen [88], J. D. Nichols [3], R. B. Nickerson [120], R. Nicolaidou [100], J. Nielsen [22], M. Niemeyer [5], J. Niermann [46], N. Nikiforou [46], V. Nikolaenko [109,251], I. Nikolic-Audit [67], P. Nilsson [7], I. Ninca [83], G. Ninio [10], A. Nisati [81], R. Nisius [117], N. Nitika [12,184], J-E. Nitschke [102], E. K. Nkadimeng [234], T. Nobe [82], D. Noll [133], T. Nommensen [235], M. B. Norfolk [74], B. J. Norman [101], M. Noury [33], J. Novak [183], T. Novak [183], R. Novotny [51], L. Nozka [169], K. Ntekas [94], N. M. J. Nunes De Moura Junior [64], J. Ocariz [67], A. Ochi [225], I. Ochoa [71], S. Oerdek [83,274], J. T. Offermann [23], A. Ogrodnik [118], A. Oh [18], C. C. Ohm [192], H. Oide [73], M. L. Ojeda [46], Y. Okumura [82], L. F. Oleiro Seabra [71], I. Oleksiyuk [49], G. Oliveira Correa [24], D. Oliveira Damazio [7], J. L. Oliver [94], R. Omar [130], Ö. O. Öncel [69], A. P. O'Neill [41], A. Onofre [71,166,275], P. U. E. Onyisi [75], M. J. Oreglia [23], D. Orestano [135,189], R. Orlandini [135,189], R. S. Orr [70], L. M. Osojnak [97], Y. Osumi [90], G. Otero y Garzon [84], H. Otono [231], M. Ouchrif [232], F. Ould-Saada [167], T. Ovsiannikova [63], M. Owen [76], R. E. Owen [21], V. E. Ozcan [38], F. Ozturk [86], N. Ozturk [108], S. Ozturk [128], H. A. Pacey [120], K. Pachal [124], A. Pacheco Pages [24], C. Padilla Aranda [24], G. Padovano [81,89], S. Pagan Griso [133], G. Palacino [130], A. Palazzo [181,185], J. Pampel [37], J. Pan [106], T. Pan [178], D. K. Panchal [75], C. E. Pandini [43], J. G. Panduro Vazquez [21], H. D. Pandya [188], H. Pang [100], P. Pani [83], G. Panizzo [12,184], L. Panwar [67], L. Paolozzi [49], S. Parajuli [152], A. Paramonov [173], C. Paraskevopoulos [42], D. Paredes Hernandez [198], A. Pareti [164,165], K. R. Park [79], T. H. Park [117], F. Parodi [140,141], J. A. Parsons [79], U. Parzefall [69], B. Pascual Dias [98], L. Pascual Dominguez [122], E. Pasqualucci [81], S. Passaggio [141], F. Pastore [28], P. Patel [86], U. M. Patel [87], J. R. Pater [18], T. Pauly [46], F. Pauwels [118], C. I. Pazos [127], M. Pedersen [167], R. Pedro [71], S. V. Peleganchuk [109], O. Penc [182], E. A. Pender [44], S. Peng [176], G. D. Penn [106], K. E. Penski [66], M. Penzin [109], B. S. Peralva [129], A. P. Pereira Peixoto [63], L. Pereira Sanchez [17], D. V. Perepelitsa [7,272], G. Perera [144], E. Perez Codina [46], M. Perganti [48], H. Pernegger [46], S. Perrella [81,89], K. Peters [83], R. F. Y. Peters [18], B. A. Petersen [46], T. C. Petersen [159], E. Petit [1], V. Petousis [51], A. R. Petri [56,77], C. Petridou [72,267], T. Petru [118], A. Petrukhin [139], M. Pettee [133], A. Petukhov [128], K. Petukhova [46], R. Pezoa [163], L. Pezzotti [39,110], G. Pezzullo [106], L. Pfaffenbichler [46], A. J. Pfleger [46], T. M. Pham [177], T. Pham [114], P. W. Phillips [21], G. Piacquadio [35], E. Pianori [133], F. Piazza [104], R. Piegaia [84], D. Pietreanu [47], A. D. Pilkington [18], M. Pinamonti [12,184], J. L. Pinfold [208], B. C. Pinheiro Pereira [71], J. Pinol Bel [24], A. E. Pinto Pinoargote [67], L. Pintucci [12,184], K. M. Piper [58], A. Pirttikoski [49], D. A. Pizzi [101], L. Pizzimento [198], A. Plebani [111], M.-A. Pleier [7],

V. Pleskot [118], E. Plotnikova [27], G. Poddar [105], R. Poettgen [34], L. Poggioli [67], S. Polacek [118], G. Polesello [164], A. Poley [153], A. Polini [39], C. S. Pollard [50], Z. B. Pollock [143], E. Pompa Pacchi [3], N. I. Pond [121], D. Ponomarenko [130], L. Pontecorvo [46], S. Popa [236], G. A. Popeneciu [237], A. Poreba [46], D. M. Portillo Quintero [124], S. Pospisil [51], M. A. Postill [74], P. Postolache [26], K. Potamianos [50], P. A. Potepa [16], I. N. Potrap [27], C. J. Potter [111], H. Potti [235], J. Poveda [32], M. E. Pozo Astigarraga [46], R. Pozzi [46], A. Prades Ibanez [30,31], S. R. Pradhan [74], J. Pretel [40], D. Price [18], M. Primavera [181], L. Primomo [12,184], M. A. Principe Martin [122], R. Privara [169], T. Procter [207], M. L. Proffitt [63], N. Proklova [97], K. Prokofiev [228], G. Proto [117], J. Proudfoot [173], M. Przybycien [16], W. W. Przygoda [207], A. Psallidas [206], J. E. Puddefoot [74], D. Pudzha [42], H. I. Purnell [188], D. Pyatiizbyantseva [162], J. Qian [68], R. Qian [147], D. Qichen [18], Y. Qin [24], T. Qiu [44], A. Quadt [5], M. Queitsch-Maitland [18], G. Quetant [49], R. P. Quinn [125], G. Rabanal Bolanos [131], D. Rafanoharana [117], F. Raffaeli [30,31], F. Ragusa [56,77], J. L. Rainbolt [23], J. A. Raine [49], S. Rajagopalan [7], E. Ramakoti [27], L. Rambelli [140,141], I. A. Ramirez-Berend [101], K. Ran [83,156], D. S. Rankin [97], N. P. Rapheeha [222], H. Rasheed [47], D. F. Rassloff [14], A. Rastogi [133], S. Rave [25], S. Ravera [140,141], B. Ravina [46], I. Ravinovich [138], M. Raymond [46], A. L. Read [167], N. P. Readioff [74], D. M. Rebuzzi [164,165], A. S. Reed [76], K. Reeves [85], J. A. Reidelsturz [136], D. Reikher [104], A. Rej [6], C. Rembser [46], H. Ren [119], M. Renda [47], F. Renner [83], A. G. Rennie [76], A. L. Rescia [83], S. Resconi [56], M. Ressegotti [140,141], S. Rettie [46], W. F. Rettie [101], M. M. Revering [111], E. Reynolds [133], O. L. Rezanova [27], P. Reznicek [118], H. Riani [232], N. Ribaric [87], B. Ricci [12,184], E. Ricci [145,146], R. Richter [117], S. Richter [149,150], E. Richter-Was [207], S. Ridouani [232], P. Rieck [11], P. Riedler [46], E. M. Riefel [149,150], J. O. Rieger [116], M. Rijssenbeek [35], M. Rimoldi [46], L. Rinaldi [39,110], P. Rincke [5,132], G. Ripellino [132], I. Riu [24], J. C. Rivera Vergara [40], F. Rizatdinova [45], E. Rizvi [105], B. R. Roberts [133], S. S. Roberts [22], D. Robinson [111], M. Robles Manzano [25], A. Robson [76], A. Rocchi [30,31], C. Roda [80,171], S. Rodriguez Bosca [46], Y. Rodriguez Garcia [233], A. M. Rodríguez Vera [19], S. Roe [46], J. T. Roemer [46], O. Røhne [167], R. A. Rojas [46], C. P. A. Roland [67], A. Romaniouk [126], E. Romano [164,165], M. Romano [39], A. C. Romero Hernandez [152], N. Rompotis [54], L. Roos [67], S. Rosati [81], B. J. Rosser [23], E. Rossi [120], E. Rossi [60,61], L. P. Rossi [131], L. Rossini [69], R. Rosten [143], M. Rotaru [47], B. Rottler [69], D. Rousseau [57], D. Rousso [83], S. Roy-Garand [70], A. Rozanov [1], Z. M. A. Rozario [76], Y. Rozen [221], A. Rubio Jimenez [32], V. H. Ruelas Rivera [103], T. A. Ruggeri [188], A. Ruggiero [120], A. Ruiz-Martinez [32], A. Rummler [46], Z. Rurikova [69], N. A. Rusakovich [27], S. Ruscelli [6], H. L. Russell [40], G. Russo [81,89], J. P. Rutherfoord [179], S. Rutherford Colmenares [111], M. Rybar [118], P. Rybczynski [16], A. Ryzhov [8], J. A. Sabater Iglesias [49], H. F-W. Sadrozinski [22], F. Safai Tehrani [81], S. Saha [188], M. Sahinsoy [128], B. Sahoo [138], A. Saibel [32], B. T. Saifuddin [3], M. Saimpert [100], G. T. Saito [226], M. Saito [82], T. Saito [82], A. Sala [56,77], A. Salnikov [17], J. Salt [32], A. Salvador Salas [10], F. Salvatore [58], A. Salzburger [46], D. Sammel [69], E. Sampson [52], D. Sampsonidis [72,267], D. Sampsonidou [104], J. Sánchez [32], V. Sanchez Sebastian [32], H. Sandaker [167], C. O. Sander [83], J. A. Sandesara [177], M. Sandhoff [136], C. Sandoval [168], L. Sanfilippo [14], D. P. C. Sankey [21], T. Sano [175], A. Sansoni [42], M. Santana Queiroz [115], L. Santi [46], C. Santoni [98], H. Santos [71,197], A. Santra [138], E. Sanzani [39,110], K. A. Saoucha [204], J. G. Saraiva [71,229], J. Sardain [179], O. Sasaki [73], K. Sato [214], C. Sauer [46], E. Sauvan [15], P. Savard [70,261], R. Sawada [82], C. Sawyer [21], L. Sawyer [199], C. Sbarra [39], A. Sbrizzi [39,110], T. Scanlon [121], J. Schaarschmidt [63], U. Schäfer [25], A. C. Schaffer [8,57], D. Schaile [66], R. D. Schamberger [35], C. Scharf [103], M. M. Schefer [41], V. A. Schegelsky [109], D. Scheirich [118], M. Schernau [220], C. Scheulen [49], C. Schiavi [140,141], M. Schioppa [160,161], B. Schlag [17], S. Schlenker [46], J. Schmeing [136], E. Schmidt [117], M. A. Schmidt [136], K. Schmieden [25], C. Schmitt [25], N. Schmitt [25], S. Schmitt [83], N. A. Schneider [66], L. Schoeffel [100], A. Schoening [193], P. G. Scholer [101], E. Schopf [139], M. Schott [37], S. Schramm [49], T. Schroer [49], H-C. Schultz-Coulon [14], M. Schumacher [69], B. A. Schumm [22], Ph. Schune [100], H. R. Schwartz [22], A. Schwartzman [17], T. A. Schwarz [68], Ph. Schwemling [100], R. Schwienhorst [147], F. G. Sciacca [41], A. Sciandra [7], G. Sciolla [85], F. Scuri [80], C. D. Sebastiani [46], K. Sedlaczek [19], S. C. Seidel [205], A. Seiden [22], B. D. Seidlitz [79], C. Seitz [83], J. M. Seixas [64], G. Sekhniaidze [60], L. Selem [43], N. Semprini-Cesari [39,110], A. Semushin [92], D. Sengupta [49], V. Senthilkumar [32], L. Serin [57], M. Sessa [60,61], H. Severini [3], F. Sforza [140,141], A. Sfyrla [49], Q. Sha [123], E. Shabalina [5], H. Shaddix [19], A. H. Shah [111], R. Shaheen [192], J. D. Shahinian [97], M. Shamim [46], L. Y. Shan [123], M. Shapiro [133], A. Sharma [46], A. S. Sharma [125], P. Sharma [7], P. B. Shatalov [109], K. Shaw [58], S. M. Shaw [18], Q. Shen [174], D. J. Sheppard [153], P. Sherwood [121], L. Shi [121], X. Shi [123], S. Shimizu [73], C. O. Shimmin [106], I. P. J. Shipsey [120,283], S. Shirabe [231], M. Shiyakova [27,276], M. J. Shochet [23], D. R. Shope [167], B. Shrestha [3], S. Shrestha [143,277], I. Shreyber [27], M. J. Shroff [40], P. Sicho [182], A. M. Sickles [152], E. Sideras Haddad [222,238], A. C. Sidley [116], A. Sidoti [39], F. Siegert [102], Dj. Sijacki [107], F. Sili [62], J. M. Silva [44], I. Silva Ferreira [64], M. V. Silva Oliveira [7], S. B. Silverstein [149], S. Simion [57], R. Simoniello [46], E. L. Simpson [18], H. Simpson [58], L. R. Simpson [173], S. Simsek [128], S. Sindhu [5], P. Sinervo [70], S. N. Singh [85], S. Singh [7], S. Sinha [83], S. Sinha [18], M. Sioli [39,110], K. Sioulas [78], I. Siral [46], E. Sitnikova [83], J. Sjölin [149,150], A. Skaf [5], E. Skorda [59],

P. Skubic [3], M. Slawinska [86], I. Slazyk [2], I. Sliusar [167], V. Smakhtin [138], B. H. Smart [21], S. Yu. Smirnov [191], Y. Smirnov [128], L. N. Smirnova [109,251], O. Smirnova [34], A. C. Smith [79], D. R. Smith [94], J. L. Smith [18], M. B. Smith [101], R. Smith [17], H. Smitmanns [25], M. Smizanska [52], K. Smolek [51], P. Smolyanskiy [51], A. A. Snesarev [27], H. L. Snoek [116], S. Snyder [7], R. Sobie [40,255], A. Soffer [10], C. A. Solans Sanchez [46], E. Yu. Soldatov [27], U. Soldevila [32], A. A. Solodkov [222], S. Solomon [85], A. Soloshenko [27], K. Solovieva [69], O. V. Solovyanov [98], P. Sommer [102], A. Sonay [24], A. Sopczak [51], A. L. Sopio [44], F. Sopkova [99], J. D. Sorenson [205], I. R. Sotarriva Alvarez [219], V. Sothilingam [14], O. J. Soto Sandoval [191,217], S. Sottocornola [130], R. Soualah [239], Z. Soumaimi [9], D. South [83], N. Soybelman [138], S. Spagnolo [181,185], M. Spalla [117], D. Sperlich [69], B. Spisso [60,61], D. P. Spiteri [76], L. Splendori [1], M. Spousta [118], E. J. Staats [101], R. Stamen [14], E. Stanecka [86], W. Stanek-Maslouska [83], M. V. Stange [102], B. Stanislaus [133], M. M. Stanitzki [83], B. Stapf [83], E. A. Starchenko [109], G. H. Stark [22], J. Stark [151], P. Staroba [182], P. Starovoitov [204], R. Staszewski [86], C. Stauch [66], G. Stavropoulos [206], A. Stefl [46], A. Stein [25], P. Steinberg [7], B. Stelzer [124,153], H. J. Stelzer [113], O. Stelzer [124], H. Stenzel [155], T. J. Stevenson [58], G. A. Stewart [46], J. R. Stewart [45], M. C. Stockton [46], G. Stoicea [47], M. Stolarski [71], S. Stonjek [117], A. Straessner [102], J. Strandberg [192], S. Strandberg [149,150], M. Stratmann [136], M. Strauss [3], T. Strebler [1], P. Strizenec [99], R. Ströhmer [142], D. M. Strom [104], R. Stroynowski [8], A. Strubig [149,150], S. A. Stucci [7], B. Stugu [2], J. Stupak [3], N. A. Styles [83], D. Su [17], S. Su [119], X. Su [119], D. Suchy [93], K. Sugizaki [97], V. V. Sulin [109], M. J. Sullivan [54], D. M. S. Sultan [120], L. Sultanaliyeva [109], S. Sultansoy [240], S. Sun [177], W. Sun [123], O. Sunneborn Gudnadottir [132], N. Sur [34], M. R. Sutton [58], H. Suzuki [214], M. Svatos [182], P. N. Swallow [111], M. Swiatlowski [124], T. Swirski [142], A. Swoboda [46], I. Sykora [93], M. Sykora [118], T. Sykora [118], D. Ta [25], K. Tackmann [83,274], A. Taffard [94], R. Tafirout [124], Y. Takubo [73], M. Talby [1], A. A. Talyshev [109], K. C. Tam [198], N. M. Tamir [10], A. Tanaka [82], J. Tanaka [82], R. Tanaka [57], M. Tanasini [35], Z. Tao [125], S. Tapia Araya [163], S. Tapprogge [25], A. Tarek Abouelfadl Mohamed [147], S. Tarem [221], K. Tariq [123], G. Tarna [46], G. F. Tartarelli [56], M. J. Tartarin [151], P. Tas [118], M. Tasevsky [182], E. Tassi [160,161], A. C. Tate [152], G. Tateno [82], Y. Tayalati [9,278], G. N. Taylor [114], W. Taylor [148], A. S. Tegetmeier [151], P. Teixeira-Dias [28], J. J. Teoh [70], K. Terashi [82], J. Terron [122], S. Terzo [24], M. Testa [42], R. J. Teuscher [70,255], A. Thaler [126], O. Theiner [49], T. Theveneaux-Pelzer [1], D. W. Thomas [28], J. P. Thomas [59], E. A. Thompson [133], P. D. Thompson [59], E. Thomson [97], R. E. Thornberry [8], C. Tian [119], Y. Tian [49], V. Tikhomirov [128], Yu. A. Tikhonov [27], S. Timoshenko [109], D. Timoshyn [118], E. X. L. Ting [188], P. Tipton [106], A. Tishelman-Charny [7], K. Todome [219], S. Todorova-Nova [118], L. Toffolin [12,184], M. Togawa [73], J. Tojo [231], S. Tokár [93], O. Toldaiev [130], G. Tolkachev [1], M. Tomoto [73,90], L. Tompkins [17,279], E. Torrence [104], H. Torres [151], E. Torró Pastor [32], M. Toscani [84], C. Tosciri [23], M. Tost [75], D. R. Tovey [74], T. Trefzger [142], P. M. Tricarico [24], A. Tricoli [7], I. M. Trigger [124], S. Trincaz-Duvoid [67], D. A. Trischuk [85], A. Tropina [27], L. Truong [55], M. Trzebinski [86], A. Trzupek [86], F. Tsai [35], M. Tsai [68], A. Tsiamis [72], P. V. Tsiareshka [27], S. Tsigaridas [124], A. Tsirigotis [72,268], V. Tsiskaridze [218], E. G. Tskhadadze [218], M. Tsopoulou [72], Y. Tsujikawa [175], I. I. Tsukerman [109], V. Tsulaia [133], S. Tsuno [73], K. Tsuri [91], D. Tsybychev [35], Y. Tu [198], A. Tudorache [47], V. Tudorache [47], S. B. Tuncay [120], S. Turchikhin [140,141], I. Turk Cakir [158], R. Turra [56], T. Turtuvshin [27,280], P. M. Tuts [79], S. Tzamarias [72,267], E. Tzovara [25], Y. Uematsu [73], F. Ukegawa [214], P. A. Ulloa Poblete [191,217], E. N. Umaka [7], G. Unal [46], A. Undrus [7], G. Unel [94], J. Urban [99], P. Urrejola [190], G. Usai [108], R. Ushioda [241], M. Usman [88], F. Ustuner [44], Z. Uysal [128], V. Vacek [51], B. Vachon [65], T. Vafeiadis [46], A. Vaitkus [121], C. Valderanis [66], E. Valdes Santurio [149,150], M. Valente [46], S. Valentinetti [39,110], A. Valero [32], E. Valiente Moreno [32], A. Vallier [151], J. A. Valls Ferrer [32], D. R. Van Arneman [116], A. Van Der Graaf [6], H. Z. Van Der Schyf [222], P. Van Gemmeren [173], M. Van Rijnbach [46], S. Van Stroud [121], I. Van Vulpen [116], P. Vana [118], M. Vanadia [30,31], U. M. Vande Voorde [192], W. Vandelli [46], E. R. Vandewall [45], D. Vannicola [10], L. Vannoli [42], R. Vari [81], M. Varma [106], E. W. Varnes [179], C. Varni [19], D. Varouchas [57], L. Varriale [32], K. E. Varvell [235], M. E. Vasile [47], L. Vaslin [73], M. D. Vassilev [17], A. Vasyukov [27], L. M. Vaughan [45], R. Vavricka [118], T. Vazquez Schroeder [24], J. Veatch [53], V. Vecchio [18], M. J. Veen [144], I. Veliscek [7], I. Velkovska [183], L. M. Veloce [70], F. Veloso [71,202], S. Veneziano [81], A. Ventura [181,185], A. Verbytskyi [117], M. Verducci [80,171], C. Vergis [105], M. Verissimo De Araujo [64], W. Verkerke [116], J. C. Vermeulen [116], C. Vernieri [17], M. Vessella [94], M. C. Vetterli [153,261], A. Vgenopoulos [25], N. Viaux Maira [163], T. Vickey [74], O. E. Vickey Boeriu [74], G. H. A. Viehhauser [120], L. Vigani [193], M. Vigl [117], M. Villa [39,110], M. Villaplana Perez [32], E. M. Villhauer [23], E. Vilucchi [42], M. Vincent [32], M. G. Vincter [101], A. Visibile [116], C. Vittori [46], I. Vivarelli [39,110], E. Voevodina [117], F. Vogel [66], J. C. Voigt [102], P. Vokac [51], Yu. Volkotrub [207], L. Vomberg [37], E. Von Toerne [37], B. Vormwald [46], K. Vorobev [87], M. Vos [32], K. Voss [139], M. Vozak [46], L. Vozdecky [3], N. Vranjes [107], M. Vranjes Milosavljevic [107], M. Vreeswijk [116], N. K. Vu [174,227], R. Vuillermet [46], O. Vujinovic [25], I. Vukotic [23], I. K. Vyas [101], J. F. Wack [111], S. Wada [214], C. Wagner [17], J. M. Wagner [133], W. Wagner [136], S. Wahdan [136], H. Wahlberg [62], C. H. Waits [3], J. Walder [21], R. Walker [66], K. Walkingshaw Pass [76], W. Walkowiak [139], A. Wall [97], E. J. Wallin [34], T. Wamorkar [133],

K. Wandall-Christensen [32], A. Wang [119], A. Z. Wang [22], C. Wang [25], C. Wang [75], H. Wang [133], J. Wang [228], P. Wang [18], P. Wang [121], R. Wang [131], R. Wang [173], S. M. Wang [187], S. Wang [123], T. Wang [119], T. Wang [119], W. T. Wang [230], W. Wang [123], X. Wang [152], X. Wang [174], X. Wang [83], Y. Wang [157], Y. Wang [119], Z. Wang [68], Z. Wang [227], Z. Wang [68], C. Wanotayaroj [73], A. Warburton [65], A. L. Warnerbring [139], N. Warrack [76], S. Waterhouse [28], A. T. Watson [59], H. Watson [44], M. F. Watson [59], E. Watton [76], G. Watts [63], B. M. Waugh [121], J. M. Webb [69], C. Weber [7], H. A. Weber [103], M. S. Weber [41], S. M. Weber [14], C. Wei [119], Y. Wei [69], A. R. Weidberg [120], E. J. Weik [11], J. Weingarten [6], C. Weiser [69], C. J. Wells [83], T. Wenaus [7], B. Wendland [6], T. Wengler [46], N. S. Wenke [117], N. Wermes [37], M. Wessels [14], A. M. Wharton [52], A. S. White [131], A. White [108], M. J. White [188], D. Whiteson [94], L. Wickremasinghe [213], W. Wiedenmann [177], M. Wielers [21], R. Wierda [192], C. Wiglesworth [159], H. G. Wilkens [46], J. J. H. Wilkinson [111], D. M. Williams [79], H. H. Williams [97], S. Williams [111], S. Willocq [144], B. J. Wilson [18], D. J. Wilson [18], P. J. Windischhofer [23], F. I. Winkel [84], F. Winklmeier [104], B. T. Winter [69], M. Wittgen [17], M. Wobisch [199], T. Wojtkowski [43], Z. Wolffs [116], J. Wollrath [46], M. W. Wolter [86], H. Wolters [71,202], M. C. Wong [22], E. L. Woodward [79], S. D. Worm [83], B. K. Wosiek [86], K. W. Woźniak [86], S. Wozniewski [5], K. Wraight [76], C. Wu [70], C. Wu [59], J. Wu [82], M. Wu [211], M. Wu [162], S. L. Wu [177], S. Wu [123], X. Wu [119], Y. Wu [119], Z. Wu [15], Z. Wu [157], J. Wuerzinger [117], T. R. Wyatt [18], B. M. Wynne [44], S. Xella [159], L. Xia [157], M. Xia [176], M. Xie [119], A. Xiong [104], J. Xiong [133], D. Xu [123], H. Xu [119], L. Xu [119], R. Xu [97], T. Xu [68], Y. Xu [63], Z. Xu [44], R. Xue [113], B. Yabsley [235], S. Yacoob [95], Y. Yamaguchi [73], E. Yamashita [82], H. Yamauchi [214], T. Yamazaki [133], Y. Yamazaki [225], S. Yan [76], Z. Yan [144], H. J. Yang [174,227], H. T. Yang [119], S. Yang [119], T. Yang [228], X. Yang [46], X. Yang [123], Y. Yang [82], Y. Yang [119], W-M. Yao [133], C. L. Yardley [58], J. Ye [123], S. Ye [7], X. Ye [119], Y. Yeh [121], I. Yeletskikh [27], B. Yeo [115], M. R. Yexley [121], T. P. Yildirim [120], K. Yorita [36], C. J. S. Young [46], C. Young [17], N. D. Young [104], Y. Yu [119], J. Yuan [123,156], M. Yuan [68], R. Yuan [174,227], L. Yue [121], M. Zaazoua [119], B. Zabinski [86], I. Zahir [33], A. Zaio [140,141], Z. K. Zak [86], T. Zakareishvili [32], S. Zambito [49], J. A. Zamora Saa [224], J. Zang [82], R. Zanzottera [56,77], O. Zaplatilek [51], C. Zeitnitz [136], H. Zeng [123], J. C. Zeng [152], D. T. Zenger Jr [85], O. Zenin [109], T. Ženiš [93], S. Zenz [105], D. Zerwas [57], M. Zhai [123,156], D. F. Zhang [74], G. Zhang [123], J. Zhang [200], J. Zhang [173], K. Zhang [123,156], L. Zhang [119], L. Zhang [157], P. Zhang [123,156], R. Zhang [157], S. Zhang [151], T. Zhang [82], Y. Zhang [63], Y. Zhang [121], Y. Zhang [119], Y. Zhang [157], Z. Zhang [200], Z. Zhang [57], H. Zhao [63], T. Zhao [200], Y. Zhao [101], Z. Zhao [119], Z. Zhao [119], A. Zhemchugov [27], J. Zheng [157], K. Zheng [152], X. Zheng [119], Z. Zheng [17], D. Zhong [152], B. Zhou [68], H. Zhou [179], N. Zhou [174], Y. Zhou [176], Y. Zhou [157], Y. Zhou [179], C. G. Zhu [200], J. Zhu [68], X. Zhu [227], Y. Zhu [174], Y. Zhu [119], X. Zhuang [123], K. Zhukov [130], N. I. Zimine [27], J. Zinsser [193], M. Ziolkowski [139], L. Živković [107], A. Zoccoli [39,110], K. Zoch [131], A. Zografos [46], T. G. Zorbas [74], O. Zormpa [206] & L. Zwalinski [46]

[1]CPPM, Aix-Marseille Université, CNRS/IN2P3, Marseille, France. [2]Department for Physics and Technology, University of Bergen, Bergen, Norway. [3]Homer L. Dodge Department of Physics and Astronomy, University of Oklahoma, Norman, OK, USA. [4]New York University Abu Dhabi, Abu Dhabi, United Arab Emirates. [5]II. Physikalisches Institut, Georg-August-Universität Göttingen, Göttingen, Germany. [6]Fakultät Physik, Technische Universität Dortmund, Dortmund, Germany. [7]Physics Department, Brookhaven National Laboratory, Upton, NY, USA. [8]Physics Department, Southern Methodist University, Dallas, TX, USA. [9]Faculté des sciences, Université Mohammed V, Rabat, Morocco. [10]Raymond and Beverly Sackler School of Physics and Astronomy, Tel Aviv University, Tel Aviv, Israel. [11]Department of Physics, New York University, New York, NY, USA. [12]INFN Gruppo Collegato di Udine, Sezione di Trieste, Udine, Italy. [13]ICTP, Trieste, Italy. [14]Kirchhoff-Institut für Physik, Ruprecht-Karls-Universität Heidelberg, Heidelberg, Germany. [15]LAPP, Université Savoie Mont Blanc, CNRS/IN2P3, Annecy, France. [16]Faculty of Physics and Applied Computer Science, AGH University of Krakow, Krakow, Poland. [17]SLAC National Accelerator Laboratory, Stanford, CA, USA. [18]School of Physics and Astronomy, University of Manchester, Manchester, UK. [19]Department of Physics, Northern Illinois University, DeKalb, IL, USA. [20]Department of Physics, Istanbul University, Istanbul, Türkiye. [21]Particle Physics Department, Rutherford Appleton Laboratory, Didcot, UK. [22]Santa Cruz Institute for Particle Physics, University of California Santa Cruz, Santa Cruz, CA, USA. [23]Enrico Fermi Institute, University of Chicago, Chicago, IL, USA. [24]Institut de Física d'Altes Energies (IFAE), Barcelona Institute of Science and Technology, Barcelona, Spain. [25]Institut für Physik, Universität Mainz, Mainz, Germany. [26]Department of Physics, Alexandru Ioan Cuza University of Iasi, Iasi, Romania. [27]Affiliated with an international laboratory covered by a cooperation agreement with CERN, Geneva, Switzerland. [28]Department of Physics, Royal Holloway University of London, Egham, UK. [29]School of Physics, Zhengzhou University, Zhengzhou, China. [30]INFN Sezione di Roma Tor Vergata, Rome, Italy. [31]Dipartimento di Fisica, Università di Roma Tor Vergata, Rome, Italy. [32]Instituto de Física Corpuscular (IFIC), Centro Mixto Universidad de Valencia - CSIC, Valencia, Spain. [33]Faculté des Sciences Ain Chock, Université Hassan II de Casablanca, Casablanca, Morocco. [34]Fysiska institutionen, Lunds universitet, Lund, Sweden. [35]Departments of Physics and Astronomy, Stony Brook University, Stony Brook, NY, USA. [36]Waseda University, Tokyo, Japan. [37]Physikalisches Institut, Universität Bonn, Bonn, Germany. [38]Department of Physics, Bogazici University, Istanbul, Türkiye. [39]INFN Sezione di Bologna, Bologna, Italy. [40]Department of Physics and Astronomy, University of Victoria, Victoria, BC, Canada. [41]Albert Einstein Center for Fundamental Physics and Laboratory for High Energy Physics, University of Bern, Bern, Switzerland. [42]INFN e Laboratori Nazionali di Frascati, Frascati, Italy. [43]LPSC, Université Grenoble Alpes, CNRS/IN2P3, Grenoble INP, Grenoble, France. [44]SUPA - School of Physics and Astronomy, University of Edinburgh, Edinburgh, UK. [45]Department of Physics, Oklahoma State University, Stillwater, OK, USA. [46]CERN, Geneva, Switzerland. [47]Horia Hulubei National Institute of Physics and Nuclear Engineering, Bucharest, Romania. [48]Physics Department, National Technical University of Athens, Zografou, Greece. [49]Département de Physique Nucléaire et Corpusculaire, Université de Genève, Geneva, Switzerland. [50]Department of Physics, University of Warwick, Coventry, UK. [51]Czech Technical University in Prague, Prague, Czech Republic. [52]Physics Department, Lancaster University, Lancaster, UK. [53]California State University, Fresno, CA, USA. [54]Oliver Lodge Laboratory, University of Liverpool, Liverpool, UK. [55]Department of Mechanical Engineering Science, University of Johannesburg, Johannesburg, South Africa. [56]INFN Sezione di Milano, Milan, Italy. [57]IJCLab, Université Paris-Saclay, CNRS/IN2P3, Orsay, France. [58]Department of Physics and Astronomy, University of Sussex, Brighton, UK. [59]School of Physics and

Astronomy, University of Birmingham, Birmingham, UK. [60]INFN Sezione di Napoli, Naples, Italy. [61]Dipartimento di Fisica, Università di Napoli, Naples, Italy. [62]Instituto de Física La Plata, Universidad Nacional de La Plata and CONICET, La Plata, Argentina. [63]Department of Physics, University of Washington, Seattle, WA, USA. [64]Universidade Federal do Rio De Janeiro COPPE/EE/IF, Rio de Janeiro, Brazil. [65]Department of Physics, McGill University, Montreal, QC, Canada. [66]Fakultät für Physik, Ludwig-Maximilians-Universität München, München, Germany. [67]LPNHE, Sorbonne Université, Université Paris Cité, CNRS/IN2P3, Paris, France. [68]Department of Physics, University of Michigan, Ann Arbor, MI, USA. [69]Physikalisches Institut, Albert-Ludwigs-Universität Freiburg, Freiburg, Germany. [70]Department of Physics, University of Toronto, Toronto, ON, Canada. [71]Laboratório de Instrumentação e Física Experimental de Partículas - LIP, Lisbon, Portugal. [72]Department of Physics, Aristotle University of Thessaloniki, Thessaloniki, Greece. [73]KEK, High Energy Accelerator Research Organization, Tsukuba, Japan. [74]Department of Physics and Astronomy, University of Sheffield, Sheffield, UK. [75]Department of Physics, University of Texas at Austin, Austin, TX, USA. [76]SUPA - School of Physics and Astronomy, University of Glasgow, Glasgow, UK. [77]Dipartimento di Fisica, Università di Milano, Milan, Italy. [78]Physics Department, National and Kapodistrian University of Athens, Athens, Greece. [79]Nevis Laboratory, Columbia University, Irvington, NY, USA. [80]INFN Sezione di Pisa, Pisa, Italy. [81]INFN Sezione di Roma, Rome, Italy. [82]International Center for Elementary Particle Physics and Department of Physics, University of Tokyo, Tokyo, Japan. [83]Deutsches Elektronen-Synchrotron DESY, Hamburg and Zeuthen, Germany. [84]Universidad de Buenos Aires, Facultad de Ciencias Exactas y Naturales, Departamento de Física, y CONICET, Instituto de Física de Buenos Aires (IFIBA), Buenos Aires, Argentina. [85]Department of Physics, Brandeis University, Waltham, MA, USA. [86]Institute of Nuclear Physics Polish Academy of Sciences, Krakow, Poland. [87]Department of Physics, Duke University, Durham, NC, USA. [88]Group of Particle Physics, University of Montreal, Montreal, QC, Canada. [89]Dipartimento di Fisica, Sapienza Università di Roma, Rome, Italy. [90]Graduate School of Science and Kobayashi-Maskawa Institute, Nagoya University, Nagoya, Japan. [91]Ochanomizu University, Bunkyo-ku, Tokyo, Japan. [92]Yerevan Physics Institute, Yerevan, Armenia. [93]Faculty of Mathematics, Physics and Informatics, Comenius University, Bratislava, Slovakia. [94]Department of Physics and Astronomy, University of California Irvine, Irvine, CA, USA. [95]Department of Physics, University of Cape Town, Cape Town, South Africa. [96]Institute of Applied Physics, Mohammed VI Polytechnic University, Ben Guerir, Morocco. [97]Department of Physics, University of Pennsylvania, Philadelphia, PA, USA. [98]LPC, Université Clermont Auvergne, CNRS/IN2P3, Clermont-Ferrand, France. [99]Department of Subnuclear Physics, Institute of Experimental Physics of the Slovak Academy of Sciences, Kosice, Slovak Republic. [100]IRFU, CEA, Université Paris-Saclay, Gif-sur-Yvette, France. [101]Department of Physics, Carleton University, Ottawa, ON, Canada. [102]Institut für Kern- und Teilchenphysik, Technische Universität Dresden, Dresden, Germany. [103]Institut für Physik, Humboldt Universität zu Berlin, Berlin, Germany. [104]Institute for Fundamental Science, University of Oregon, Eugene, OR, USA. [105]Department of Physics and Astronomy, Queen Mary University of London, London, UK. [106]Department of Physics, Yale University, New Haven, CT, USA. [107]Institute of Physics, University of Belgrade, Belgrade, Serbia. [108]Department of Physics, University of Texas at Arlington, Arlington, TX, USA. [109]Affiliated with an institute formerly covered by a cooperation agreement with CERN, Geneva, Switzerland. [110]Dipartimento di Fisica e Astronomia A. Righi, Università di Bologna, Bologna, Italy. [111]Cavendish Laboratory, University of Cambridge, Cambridge, UK. [112]United Arab Emirates University, Al Ain, United Arab Emirates. [113]Department of Physics and Astronomy, University of Pittsburgh, Pittsburgh, PA, USA. [114]School of Physics, University of Melbourne, Melbourne, VIC, Australia. [115]University of California, Berkeley, CA, USA. [116]Nikhef National Institute for Subatomic Physics and University of Amsterdam, Amsterdam, The Netherlands. [117]Max-Planck-Institut für Physik (Werner-Heisenberg-Institut), München, Germany. [118]Charles University, Faculty of Mathematics and Physics, Prague, Czech Republic. [119]Department of Modern Physics and State Key Laboratory of Particle Detection and Electronics, University of Science and Technology of China, Hefei, China. [120]Department of Physics, Oxford University, Oxford, UK. [121]Department of Physics and Astronomy, University College London, London, UK. [122]Departamento de Física Teorica C-15 and CIAFF, Universidad Autónoma de Madrid, Madrid, Spain. [123]Institute of High Energy Physics, Chinese Academy of Sciences, Beijing, China. [124]TRIUMF, Vancouver, BC, Canada. [125]Department of Physics, University of British Columbia, Vancouver, BC, Canada. [126]Universität Innsbruck, Department of Astro and Particle Physics, Innsbruck, Austria. [127]Department of Physics and Astronomy, Tufts University, Medford, MA, USA. [128]Istinye University, Sariyer, Istanbul, Türkiye. [129]Rio de Janeiro State University, Rio de Janeiro, Brazil. [130]Department of Physics, Indiana University, Bloomington, IN, USA. [131]Laboratory for Particle Physics and Cosmology, Harvard University, Cambridge, MA, USA. [132]Department of Physics and Astronomy, University of Uppsala, Uppsala, Sweden. [133]Physics Division, Lawrence Berkeley National Laboratory, Berkeley, CA, USA. [134]APC, Université Paris Cité, CNRS/IN2P3, Paris, France. [135]INFN Sezione di Roma Tre, Rome, Italy. [136]Fakultät für Mathematik und Naturwissenschaften, Fachgruppe Physik, Bergische Universität Wuppertal, Wuppertal, Germany. [137]Department of Physics Engineering, Gaziantep University, Gaziantep, Türkiye. [138]Department of Particle Physics and Astrophysics, Weizmann Institute of Science, Rehovot, Israel. [139]Department Physik, Universität Siegen, Siegen, Germany. [140]Dipartimento di Fisica, Università di Genova, Genoa, Italy. [141]INFN Sezione di Genova, Genoa, Italy. [142]Fakultät für Physik und Astronomie, Julius-Maximilians-Universität Würzburg, Würzburg, Germany. [143]Ohio State University, Columbus, OH, USA. [144]Department of Physics, University of Massachusetts, Amherst, MA, USA. [145]INFN-TIFPA, Trento, Italy. [146]Università degli Studi di Trento, Trento, Italy. [147]Department of Physics and Astronomy, Michigan State University, East Lansing, MI, USA. [148]Department of Physics and Astronomy, York University, Toronto, ON, Canada. [149]Department of Physics, Stockholm University, Stockholm, Sweden. [150]Oskar Klein Centre, Stockholm, Sweden. [151]L2IT, Université de Toulouse, CNRS/IN2P3, UPS, Toulouse, France. [152]Department of Physics, University of Illinois, Urbana, IL, USA. [153]Department of Physics, Simon Fraser University, Burnaby, BC, Canada. [154]Department of Physics, Boston University, Boston, MA, USA. [155]II. Physikalisches Institut, Justus-Liebig-Universität Giessen, Giessen, Germany. [156]University of Chinese Academy of Science (UCAS), Beijing, China. [157]Department of Physics, Nanjing University, Nanjing, China. [158]Department of Physics, Ankara University, Ankara, Türkiye. [159]Niels Bohr Institute, University of Copenhagen, Copenhagen, Denmark. [160]Dipartimento di Fisica, Università della Calabria, Rende, Italy. [161]INFN Gruppo Collegato di Cosenza, Laboratori Nazionali di Frascati, Frascati, Italy. [162]Institute for Mathematics, Astrophysics and Particle Physics, Radboud University/Nikhef, Nijmegen, The Netherlands. [163]Departamento de Física, Universidad Técnica Federico Santa María, Valparaíso, Chile. [164]INFN Sezione di Pavia, Pavia, Italy. [165]Dipartimento di Fisica, Università di Pavia, Pavia, Italy. [166]Departamento de Física, Escola de Ciências, Universidade do Minho, Braga, Portugal. [167]Department of Physics, University of Oslo, Oslo, Norway. [168]Departamento de Física, Universidad Nacional de Colombia, Bogotá, Colombia. [169]Joint Laboratory of Optics, Palacký University, Olomouc, Czech Republic. [170]Departamento de Engenharia Elétrica, Universidade Federal de Juiz de Fora (UFJF), Juiz de Fora, Brazil. [171]Dipartimento di Fisica E. Fermi, Università di Pisa, Pisa, Italy. [172]High Energy Physics Institute, Tbilisi State University, Tbilisi, Georgia. [173]High Energy Physics Division, Argonne National Laboratory, Lemont, IL, USA. [174]State Key Laboratory of Dark Matter Physics, School of Physics and Astronomy, Shanghai Jiao Tong University, Key Laboratory for Particle Astrophysics and Cosmology (MOE), SKLPPC, Shanghai, China. [175]Faculty of Science, Kyoto University, Kyoto, Japan. [176]Physics Department, Tsinghua University, Beijing, China. [177]Department of Physics, University of Wisconsin, Madison, WI, USA. [178]Department of Physics, Chinese University of Hong Kong, Shatin, N.T., Hong Kong, China. [179]Department of Physics, University of Arizona, Tucson, AZ, USA. [180]Department of Physics, National Tsing Hua University, Hsinchu, Taiwan. [181]INFN Sezione di Lecce, Lecce, Italy. [182]Institute of Physics of the Czech Academy of Sciences, Prague, Czech Republic. [183]Department of Experimental Particle Physics, Jožef Stefan Institute and Department of Physics, University of Ljubljana, Ljubljana, Slovenia. [184]Dipartimento Politecnico di Ingegneria e Architettura, Università di Udine, Udine, Italy. [185]Dipartimento di Matematica e Fisica, Università del Salento, Lecce, Italy. [186]Departamento de Fisica, Instituto Superior Técnico, Universidade de Lisboa, Lisbon, Portugal. [187]Institute of Physics, Academia Sinica, Taipei, Taiwan. [188]Department of Physics, University of Adelaide, Adelaide, SA, Australia. [189]Dipartimento di Matematica e Fisica, Università Roma Tre,

Rome, Italy. [190]Departamento de Física, Pontificia Universidad Católica de Chile, Santiago, Chile. [191]Millennium Institute for Subatomic physics at high energy frontier (SAPHIR), Santiago, Chile. [192]Department of Physics, Royal Institute of Technology, Stockholm, Sweden. [193]Physikalisches Institut, Ruprecht-Karls-Universität Heidelberg, Heidelberg, Germany. [194]Federal University of Bahia, Salvador, Bahia, Brazil. [195]Faculty of Physics, University of Bucharest, Bucharest, Romania. [196]Faculté des Sciences, Université Ibn-Tofail, Kénitra, Morocco. [197]Departamento de Física, Faculdade de Ciências, Universidade de Lisboa, Lisbon, Portugal. [198]Department of Physics, University of Hong Kong, Hong Kong, China. [199]Louisiana Tech University, Ruston, LA, USA. [200]Institute of Frontier and Interdisciplinary Science and Key Laboratory of Particle Physics and Particle Irradiation (MOE), Shandong University, Qingdao, China. [201]Centro Nacional de Microelectrónica (IMB-CNM-CSIC), Barcelona, Spain. [202]Departamento de Física, Universidade de Coimbra, Coimbra, Portugal. [203]National Institute of Physics, University of the Philippines Diliman (Philippines), Quezon City, Philippines. [204]University of Sharjah, Sharjah, United Arab Emirates. [205]Department of Physics and Astronomy, University of New Mexico, Albuquerque, NM, USA. [206]National Centre for Scientific Research "Demokritos", Agia Paraskevi, Greece. [207]Marian Smoluchowski Institute of Physics, Jagiellonian University, Krakow, Poland. [208]Department of Physics, University of Alberta, Edmonton, AB, Canada. [209]University of Georgia, Tbilisi, Georgia. [210]West University in Timisoara, Timisoara, Romania. [211]School of Science, Shenzhen Campus of Sun Yat-sen University, Guangzhou, China. [212]Department of Physics, Shinshu University, Nagano, Japan. [213]Graduate School of Science, University of Osaka, Osaka, Japan. [214]Division of Physics and Tomonaga Center for the History of the Universe, Faculty of Pure and Applied Sciences, University of Tsukuba, Tsukuba, Japan. [215]National University of Science and Technology Politechnica, Bucharest, Romania. [216]Institute of Physics, Azerbaijan Academy of Sciences, Baku, Azerbaijan. [217]Instituto de Investigación Multidisciplinario en Ciencia y Tecnología, y Departamento de Física, Universidad de La Serena, La Serena, Chile. [218]E. Andronikashvili Institute of Physics, Iv. Javakhishvili Tbilisi State University, Tbilisi, Georgia. [219]Department of Physics, Institute of Science, Tokyo, Japan. [220]Instituto de Alta Investigación, Universidad de Tarapacá, Arica, Chile. [221]Department of Physics, Technion, Israel Institute of Technology, Haifa, Israel. [222]School of Physics, University of the Witwatersrand, Johannesburg, South Africa. [223]Department of Physics and Astronomy, Iowa State University, Ames, IA, USA. [224]Department of Physics, Universidad Andres Bello, Santiago, Chile. [225]Graduate School of Science, Kobe University, Kobe, Japan. [226]Instituto de Física, Universidade de São Paulo, São Paulo, Brazil. [227]State Key Laboratory of Dark Matter Physics, Tsung-Dao Lee Institute, Shanghai Jiao Tong University, Shanghai, China. [228]Department of Physics and Institute for Advanced Study, Hong Kong University of Science and Technology, Clear Water Bay, Kowloon, Hong Kong, China. [229]Centro de Física Nuclear da Universidade de Lisboa, Lisbon, Portugal. [230]University of Iowa, Iowa City, IA, USA. [231]Research Center for Advanced Particle Physics and Department of Physics, Kyushu University, Fukuoka, Japan. [232]LPMR, Faculté des Sciences, Université Mohamed Premier, Oujda, Morocco. [233]Facultad de Ciencias y Centro de Investigaciónes, Universidad Antonio Nariño, Bogotá, Colombia. [234]iThemba Labs, Cape Town, Western Cape, South Africa. [235]School of Physics, University of Sydney, Sydney, NSW, Australia. [236]Transilvania University of Brasov, Brasov, Romania. [237]Physics Department, National Institute for Research and Development of Isotopic and Molecular Technologies, Cluj-Napoca, Romania. [238]University of West Attica, Athens, Greece. [239]Khalifa University of Science and Technology, Abu Dhabi, United Arab Emirates. [240]Division of Physics, TOBB University of Economics and Technology, Ankara, Türkiye. [241]Graduate School of Science and Technology, Tokyo Metropolitan University, Tokyo, Japan. [242]Department of Physics, King's College London, London, UK. [243]Institute of Physics, Azerbaijan Academy of Sciences, Baku, Azerbaijan. [244]Imam Mohammad Ibn Saud Islamic University, Riyadh, Saudi Arabia. [245]Department of Physics, University of Thessaly, Volos, Greece. [246]An-Najah National University, Nablus, Palestine. [247]Department of Physics, University of Fribourg, Fribourg, Switzerland. [248]Department of Physics, Westmont College, Santa Barbara, CA, USA. [249]Departament de Fisica de la Universitat Autonoma de Barcelona, Barcelona, Spain. [250]University of Siena, Siena, Italy. [251]Affiliated with an institute formerly covered by a cooperation agreement with CERN, Geneva, Switzerland. [252]The Collaborative Innovation Center of Quantum Matter (CICQM), Beijing, China. [253]Faculty of Physics, Sofia University 'St. Kliment Ohridski', Sofia, Bulgaria. [254]Università di Napoli Parthenope, Naples, Italy. [255]Institute of Particle Physics (IPP), Ottawa, ON, Canada. [256]Department of Physics, Bolu Abant Izzet Baysal University, Bolu, Türkiye. [257]Faculty of Physics, University of Bucharest, Bucharest, Romania. [258]Borough of Manhattan Community College, City University of New York, New York, NY, USA. [259]National Institute of Physics, University of the Philippines Diliman (Philippines), Quezon City, Philippines. [260]Department of Financial and Management Engineering, University of the Aegean, Chios, Greece. [261]TRIUMF, Vancouver, BC, Canada. [262]Institucio Catalana de Recerca i Estudis Avancats, ICREA, Barcelona, Spain. [263]Henan University, Kaifeng, China. [264]Physics Department, Yeditepe University, Istanbul, Türkiye. [265]Institute of Theoretical Physics, Ilia State University, Tbilisi, Georgia. [266]CERN, Geneva, Switzerland. [267]Center for Interdisciplinary Research and Innovation (CIRI-AUTH), Thessaloniki, Greece. [268]Hellenic Open University, Patras, Greece. [269]Department of Modern Physics and State Key Laboratory of Particle Detection and Electronics, University of Science and Technology of China, Hefei, China. [270]Department of Mathematical Sciences, University of South Africa, Johannesburg, South Africa. [271]Department of Physics, Stellenbosch University, Stellenbosch, South Africa. [272]Department of Physics, University of Colorado Boulder, Boulder, CO, USA. [273]Département de Physique Nucléaire et Corpusculaire, Université de Genève, Geneva, Switzerland. [274]Institut für Experimentalphysik, Universität Hamburg, Hamburg, Germany. [275]Centre of Physics of the Universities of Minho and Porto (CF-UM-UP), Porto, Portugal. [276]Institute for Nuclear Research and Nuclear Energy (INRNE) of the Bulgarian Academy of Sciences, Sofia, Bulgaria. [277]Washington College, Chestertown, MD, USA. [278]Institute of Applied Physics, Mohammed VI Polytechnic University, Ben Guerir, Morocco. [279]Department of Physics, Stanford University, Stanford, CA, USA. [280]Institute of Physics and Technology, Mongolian Academy of Sciences, Ulaanbaatar, Mongolia. [281]Deceased: J. Khubua. [282]Deceased: U. Mallik. [283]Deceased: I. P. J. Shipsey. [284]University of South Africa, Department of Physics, Pretoria, South Africa. [285]University of Zululand, KwaDlangezwa, South Africa. [286]Faculté des Sciences Semlalia, Université Cadi Ayyad, LPHEA-Marrakech, Morocco. [287]Departamento de Física Téorica y del Cosmos, Universidad de Granada, Granada, Spain. [288]Universidad San Sebastian, Recoleta, Chile.
✉e-mail: atlas.publications@cern.ch

