## [Transparent Peer Review file · Nature Communications]

Transforming jet flavour tagging at ATLAS

Corresponding Author: Dr ATLAS Collaboration

Version 0:

Reviewer comments:

Reviewer #1

(Remarks to the Author)

Report on “Transforming jet flavour tagging at ATLAS” by the ATLAS Collaboration.

This manuscript describes GN2, a new jet flavour tagging algorithm, designed to be deployed in the analysis chain of the ATLAS experiment at CERN. It is based on a transformer-based neural network architecture and trained to tag $b/c/\tau$ light jets. In contrast to previous approaches (which were receiving track-level information from a separate algorithmic step as input), the model works on low-level input information and processes them in a single step. Additional outputs of the model predict the track origin and vertex grouping. These auxiliary tasks do not only provide useful information for the subsequent analysis, but also help during the training phase. The authors train their model with simulated events and evaluate on both simulated and real data. The algorithm shows a significant improvement of a factor of a few over the existing model DL1d, which is very impressive!

In total, the manuscript presents very solid results with far reaching implications to the entire ATLAS physics program. As such, they should be published in Nature Communications, as they clearly “will be important to the field and advance understanding in a way that will move the field forward.” (taken from the [criteria for publication] (<https://www.nature.com/ncomms/for-reviewers/writing-your-report>) for Nature Communications).

In addition, the inclusion of the two auxiliary training objectives—track origin classification and vertex finding—represents a significant innovation. These tasks are not only physically motivated and interpretable, but are also tightly integrated into the architecture in a way that strengthens the overall performance. The successful implementation and validation of these auxiliary tasks demonstrate a deep understanding of both machine learning and collider physics and create new possibilities for future applications.

Before publication, however, I would like to see a few points regarding the description of the method and the broader context addressed:

- The way the results are presented is very centered on comparisons to ATLAS benchmarks. As someone from outside of ATLAS, I was wondering how GN2 connects to other available models for jet flavour tagging. For example the ParticleTransformer of 2202.03772. How different / similar is the architecture? I know the input data of GN2 and ParT is different, so a direct comparison of these is probably impossible, but nevertheless putting GN2 more into context would be very beneficial. Along these lines, I was also wondering how much GN2 improves over GN1, which was not transformer based. Since GN2 shares the same input features (except the variables related to holes in the silicon tracker) and an overall similar architecture with GN1—replacing the Graph Attention Network with a Transformer—it would be helpful to explicitly report the baseline performance of GN1. This would allow readers to better quantify the improvement achieved by the architectural update. Can the improvement from GN1 to GN2 be summarized in the text or shown in figures 2 and 3?

- I really like that you added the paragraph:

“To facilitate future developments and strengthen the connections between collider experiments and the broader scientific research community, a simulation sample with all the required information to train GN2 can be acquired via the CERN Open Data Portal [21].” However, to really facilitate further developments:

- it should be added that this data is only part of the training data. Would it be possible to publish the Z' samples too?

- the link to the actual code should be added as well. I think the repo with the configuration files is <https://gitlab.cern.ch/atlas/open-data/transforming-jet-flavor> while the repo with the actual code is <https://gitlab.cern.ch/atlas-flavor-tagging-tools/algorithms/salt/> . Is that correct? If yes, please add those references as well, maybe with a commit hash or via zenodo snapshot such that it is clear which version of the code was used for the publication here.

- In general, please follow the guidelines of <https://www.nature.com/nature-portfolio/editorial-policies/reporting-standards>

Minor comments:

- In fig. 1, top left: Is the beamline along the horizontal axis? Does the “impact parameter $\langle - \rightarrow$ ” correspond to z_0 of the text (p 11)? Does the vertical dotted line correspond to d_0 of the text (p 11)?

- When referring to GEANT4 (currently [24]), the authors should refer to the 3 publications that are given at the bottom of <https://geant4.web.cern.ch/> to give credit to the GEANT4 collaboration.

- In figs. 2 and 3, would it be possible to indicate the OPs with vertical dashed lines?

- Figure 4 presents the calibrated tagging efficiencies and rejections of GN2 and DL1d, along with their associated uncertainties, based on Run 2 data. Are any results available for Run 3 data using GN2? Given that the LHC is now operating in Run 3, it would be valuable to assess the calibration stability and potential performance shifts under the updated detector and data-taking conditions. Even a qualitative discussion would help understand the robustness of the method.

- The manuscript notes that class-weighted losses are applied during training to mitigate the class imbalance in the track origin classification task. Could the authors clarify what specific class weights were used, and whether they were fixed or dynamically adjusted during training? Providing this information would improve the transparency and reproducibility of the training procedure, especially given the strong class imbalance among the track origin categories.

- In the Discussion section, the manuscript reports that “GN2 achieves an efficiency (purity) of 84% (84%). For tracks that are not of heavy-flavour origin, the efficiency (purity) is 85% (96%).” However, it is not clear whether these results are derived from Run 2 or Run 3 data (Does the statement from page 4, first paragraph, apply here, too, even though it is in a different section?). Similarly, Table 1 presents ratios of the efficiencies obtained with samples using alternative MC generators, relative to the nominal Powheg Box + Pythia sample, but does not specify which data-taking period these results correspond to. It would be helpful for assessing the potential dependence on detector conditions if the authors could clarify whether these results are based on Run 2 or Run 3 data.

- More of a curiosity: Figure 5b seems to indicate a linear shift between GN2 and the MC truth. Is there an easy explanation for that, or did that happen by chance?

- To better quantify the contributions of each auxiliary objective, it would be valuable if the authors could provide an ablation table showing the individual impact of disabling the track-origin classification and vertex-grouping tasks. This would help clarify how much each auxiliary task contributes to the overall jet classification performance. In addition, I think it would be better for the flow of reading the manuscript when it would be mentioned that these auxiliary tasks also help to improve the jet tagging performance. Right now, these auxiliary tasks are mentioned a few times, but their impact on jet tagging is only mentioned at the very end, making the reader wonder if there was a benefit from including them.

- In table 1, what are the not-shown statistical uncertainties? Are these from retraining the networks with different random seeds or from different MC samples? If the quoted uncertainties do not include the effect from retraining the network, please add this information.

- How sensitive is the GN2 performance on the choice of jet clustering algorithm or radius parameters? How relevant are other jet clustering algorithms for GN2 and subsequent analyses?

- The manuscript states that “the jet features are concatenated with the feature vectors of up to 40 leading tracks associated with the jet, ranked by the absolute track impact parameter significance.” It would be helpful if the authors could clarify whether zero-padding is applied when fewer than 40 tracks are available, and also specify the batch size used during training. This information is important for understanding the model’s input handling and memory efficiency.

- The manuscript provides only a brief architectural description of the two auxiliary objectives. For example, it does not describe how the pairwise track embeddings are constructed for the track-pair compatibility task. Since this part is identical to the architecture used in GN1, it would be helpful to include a sentence such as “Please refer to the GN1 paper for details.”

- What is the form of the global loss function? What are the “tunable weights” to combine the 3 objectives? How are they tuned?

- The authors should clarify that the “4-fold strategy” refers to training four independent models with the same architecture on mutually exclusive subsets of the training data, rather than performing standard 4-fold cross-validation. If the outputs of the four models are combined during inference (e.g., by averaging predictions), this should also be explicitly stated.

- Given that each GN2 fold has approximately 2.3 million trainable parameters and operates on large low-level input data, it would be valuable if the authors could provide an estimate of the computational cost during training and inference, at least in the context of simulated events. If available, an estimate of the inference latency in the ATLAS offline or trigger system would also be highly informative, especially for assessing the model's deployability in practical data-taking scenarios.

- Could the authors please comment on the IRC safety of the algorithm?

Reviewer #2

(Remarks to the Author)

Reviewer #3

(Remarks to the Author)

The ATLAS Collaboration presents a new transformer-based algorithm for flavour tagging of jets at the LHC and demonstrates a significant improvement with respect to previous algorithms used in ATLAS. The novelty of this article lies in the use of auxiliary prediction tasks, which affect the main task of flavour tagging, and reveal new avenues for future physics applications. The article also provides details about the methods employed in this process and proves the robustness of the model with performance measurements in collision data. Additionally, the authors provide publicly available simulation that can potentially act as a benchmark dataset for future Machine Learning developments and marks a step in the right direction. I would like to congratulate the authors on these significant achievements.

However, there are certain claims in the article that can be considered inflated, especially due to the lack of a chronological review and references to related developments. I would recommend publication in Nature Communications after the following comments are addressed.

Major comments (comments regarding scientific validity):

1. The article claims a “paradigm shift from previous approaches” in the abstract and throughout the text (page 2, 12). This statement, as it is, is too strong to be universally true. It needs to be clarified that the paradigm shift being referred to is specific only within the ATLAS experiment, where an end-to-end algorithm is being used for the first time starting with the “GN” series. This fact is reflected in the title of the article, but not elsewhere.

2. A brief paragraph with a chronological review of past/parallel algorithm developments, both in phenomenological studies and in experiments like CMS, is imperative even if the statement of a general “paradigm shift” is dropped. In fact, the present article from ATLAS builds on several concepts that were introduced earlier/in parallel (such as end-to-end tagging and the use of transformers; I would label the inclusion of auxiliary targets as the main novelty of the present article). This should be discussed somewhere, e.g. in a dedicated section before the Results section. Following is a non-exhaustive list of references relevant to LHC experiments:

- JINST 15 (2020) P12012 (end-to-end tagging)
- JINST 13 (2018) P05011 (end-to-end tagging)
- Phys. Rev. D 101, 056019 (2020) (end-to-end tagging)
- CMS-PAS-BTV-22-001 (end-to-end bb-tagging)
- ATL-PHYS-PUB-2022-027 (followed up in major comment #3)
- PMLR 162:18281-18292, 2022 (end-to-end tagging + transformer)

I did not find transformer-based peer-reviewed publications from CMS, but review articles may be cited for further reading on work parallel to GN1/GN2.

3. The GN1 is described as a “demonstrator version” in the present article with a non-peer-reviewed reference from three years ago. This information does not appear until page 13. At this point, it directly contradicts the claim that “GN2 presents a novel approach which breaks the design paradigm of the two-staged algorithms”.

I fail to see why the information about the development of a first version was not included right before the “new paradigm” discussion in the introductory section. This would:

- (a) help the reader understand the motivation behind the nomenclature of “GN2”,
- (b) establish the “GN” series (not GN2, in particular) as the new paradigm in ATLAS, which is factually correct,
- (c) enrich the article's ability to convey the step-by-step developments that large ML models typically go through, and,
- (d) along the lines of major comment #2, the article can be considered a “novel” development only if the “GN” series, as a whole, is established as work done in parallel to other related works, and not a mere application of concepts previously described in other articles.

Therefore, I strongly recommend highlighting GN1, in addition to GN2, in the introductory sections, along with a review of previous/parallel methods.

Further comments to improve the article based on Nature Communications standards:

1. The abstract mentions “substantial benefits for physics analyses” with specific examples, but the improvements, e.g. in sensitivity, are not explored in the article, but only referenced. I recommend removing this information from the abstract and replacing it with a quantification of the improvement in the per-jet background rejection/tagging efficiency, which is the primary message of the article (please also see Further comment #8).
 2. Page 2, para 3: It would be interesting to know what fraction (quantitatively) of b-jets contain a tertiary vertex in simulation. How does this additional vertex, when present, affect the Billoir fit, L_{xy} , m_{SV} (page 8 and Fig 5)? Naively, it would appear that the presence of a tertiary vertex would skew the quantities under the one-vertex-per-jet hypothesis, which is apparently adopted (“...performed on the tracks selected by the GN2 and SV1 vertex finding algorithms”).
 3. Page 2, para 4: “relative to the algorithm used in most”: do you mean “algorithms” or is there one algorithm you are referring to? For the latter (former), a quantification of the (ballpark of) improvement would be useful.
 4. Page 2, para 5: “Underlying physics process” may be misleading here (though it is clarified later on page 7, last para). Parentheses with “primary interaction”, “hadrons”, “pileup” may help clarify what “physics process” means in this context.
 5. Page 4, para 1: “similar results are obtained with Run-2 samples”: Two comments here:
 - Please clarify that Run-2 samples are with $\sqrt{s}=13$ TeV at the end of the para; this is mentioned before, but mentioning 13.6 for one makes it look like the same energy is used for the other in this text.
 - This is an interesting observation. How “similar” are they? Is this a consequence of having heterogenous \sqrt{s} samples in the training? Or is it because flavour-tagging information leveraged by the network is independent of \sqrt{s} ? What gain (if any) would we expect from evaluating performance on 13 TeV samples using a standalone 13 TeV training, and vice versa? A clarification here would help motivate the use of different Run samples in the training.
 6. Page 4, para 4: “Applying the 70% OP ... efficiency of 30%”. This indicates a strong dependence on jet $p_{\{T\}}$. While the reasons are qualitatively discussed here, the discussion would benefit from a $p_{\{T\}}$ -dependent plot of the tagging efficiencies at a fixed mistag rate (or vice versa). A less-ideal option is to at least mention the median $p_{\{T\}}$ of the two categories discussed here and shown in Fig 2.
 7. Fig 2 caption: “binomial distribution”: How are correlations treated while taking the ratio, since I understand that the same jets are used in GN2 and DL1d curves?
 8. Page 7, para 1: “clear improvements over DL1d”: This is one of the main results of the article. Can this be quantified, e.g. with “x-times higher c-jet rejection at b-jet efficiency of y in collision data”, including a rough uncertainty stemming from the calibrations? This information should also be added to the abstract.
 9. Page 8, para 3 and Fig 5: SV1 seems to overall underestimate the SV mass, whereas GN2 consistently overestimates. Is the reason understood? The text mentions “good agreement”, but this statement should be improved with a short discussion.
 10. Table 1: The c-jet efficiency is altered by 10% when choosing Herwig over Pythia. This is not insignificant. This is also counterintuitive as I would expect mismodeling in b-jets to be an exaggeration of the mismodeling in c-jets, unless, e.g., D-hadron simulation is particularly different between Pythia and Herwig.
 - (a) The text (page 10, para 1) can be improved with a discussion and a summary of key differences between Pythia and Herwig which may affect D/c hadron simulation.
 - (b) This observation also invites investigating the efficiency ratios corresponding to at least one c-tagging OP (e.g. 30% c-eff).The same comment somewhat applies to Sherpa vs. Powheg+Pythia as well, where differences in c and light efficiencies are up to 7.7%, but negligible for b.
 11. Page 12: A coarse hyperparameter optimization is mentioned.
 - (a) It may help to elaborate on which hyperparameters were altered. Was the model size changed? Is it known if a larger model would lead to overfitting?
 - (b) It would also be good to document if any specialized computing methods were used for hyperparameter optimization in the GN series (e.g. Grid GPU architectures [*]), as this is a challenge in modern large model trainings and is of interest to the HEP community as well as the broader ML community.
- [*] <https://atlas.web.cern.ch/Atlas/GROUPS/PHYSICS/PLOTS/FTAG-2019-001/>
12. Page 13, para 2: “ranked by absolute track impact parameter significance”, is the ranking/sorting of any significance? For a transformer architecture, I would expect “ordering” to have no physical significance. A comment on whether the ordering matters will be helpful for the article, especially since “DIPS” emphasizes on the lack of ordering.

13. Page 13, para 3: 4-fold strategy: What is the aggregation strategy and plans for usage in analyses? Does the proposal involve averaging over the predictions of 4 different models in physics analyses? Are the results shown in the article the average of 4 models? A comment here would help clarify.

Textual comments:

Page 2, para 2: neither of the above -> none of the above

Page 2, para 2: The state-of-the-art algorithm -> The state-of-the-art algorithms (unless you mean one single algorithm, in which case it should be named and referenced)

Page 2, para 2: should end with ", respectively"

Page 2 and onward: please use "Run-2" or "Run 2", "Run-3" or "Run 3" consistently.

"Transformer" vs. "transformer": Is there a reason for different types of capitalizations?

Figure 1: The text, especially the ones with subscripts, will be more readable (e.g. on an A4 print) with larger font sizes.

Reference 3: "The" in "ATLAS Collaboration" is an outlier.

References 2, 3, 42, and potentially others are missing preprint links.

Reference 73: "A." -> "ATLAS"

Reviewer #4

(Remarks to the Author)

The paper presents an improved jet flavor tagger for the ATLAS experiment. It exploits modern AI methods to include low-level data and shows a significantly better performance than previous algorithms. The paper is well written and the results look convincing.

Having the Methods section at the end may make it difficult for non-experts to understand some earlier parts, but that is a consequence of the given structure. I only have some minor comments.

- Is GN2 an abbreviation? If yes I suggest mentioning it in the paper.
- It may be worth adding a sentence in the introduction that gives the non-expert readers an idea of what a hadronic jet is as this is the main object of study in this paper.
- The third paragraph nicely explains the physics behind the signature of b-jets. As the presented tagger also identifies c-jets and tau decays, it would be good to also explain their signatures.
- It is very good that the data of this study is publicly available.
- Are the relative weighting factors f_b , f_c and f_{τ} dependent on the specific selection of an analysis? If yes, does this mean that they should be determined for each analysis? I suggest clarifying this in the paper.
- Regarding the above mentioned fractions, why are they different for the different taggers? Shouldn't the optimal values be those of the true fractions (which are known for MC)?
- Figure 5: Does the difference between distributions come from different efficiencies or are there also different resolutions of the reconstructed vertices?
- Table 1 and related text. The main argument is based on the relative disagreement. Therefore I suggest adding these values to the table.
- Methods section: I suggest "pp collisions at the LHC" instead of "LHC collisions".

Version 1:

Reviewer comments:

Reviewer #1

(Remarks to the Author)

Report on the revised manuscript "Transforming jet flavour tagging at ATLAS" by the ATLAS Collaboration.

The authors have addressed all my previous comments to my satisfaction and improved their manuscript considerably. I'm happy to recommend it for publication in Nature Communications and I'm looking forward to many nice physics results with the improved jet flavour tagging.

There are just 2 minor things I spotted which the authors can consider before sending the paper for proofs:

- On page 4, in the newly introduced sentence before the equation, they say "with Run-3 samples at $\sqrt{s}=13$ TeV", which contradicts itself. Please correct.
- On page 8, when discussing the track classification performance, the authors promised in their reply to add the information that the results are based on Run 3, but I cannot find this information in the manuscript.

Reviewer #2

(Remarks to the Author)

Reviewer #3

(Remarks to the Author)

Dear authors,

Thank you very much for implementing the requested changes. I am satisfied with the updated draft and recommend publication.

I also lately noticed that the author formatting, "P. D. Group et al", in Ref [18] may be unintentional.

Congratulations again on the very nice results!

Reviewer #4

(Remarks to the Author)

My comments have been addressed and I recommend publishing the paper.

Dear Referee,

We thank you for the careful reading and the very constructive suggestions. Your comments and suggestions have been addressed in the new revision. We have also prepared a pdf-diff to aid the review. Please see our detailed replies to your comments below:

=====
Comment: The way the results are presented is very centered on comparisons to ATLAS benchmarks. As someone from outside of ATLAS, I was wondering how GN2 connects to other available models for jet flavour tagging. For example the ParticleTransformer of 2202.03772. How different / similar is the architecture? I know the input data of GN2 and PartT is different, so a direct comparison of these is probably impossible, but nevertheless putting GN2 more into context would be very beneficial. Along these lines, I was also wondering how much GN2 improves over GN1, which was not transformer based. Since GN2 shares the same input features (except the variables related to holes in the silicon tracker) and an overall similar architecture with GN1—replacing the Graph Attention Network with a Transformer—it would be helpful to explicitly report the baseline performance of GN1. This would allow readers to better quantify the improvement achieved by the architectural update. Can the improvement from GN1 to GN2 be summarized in the text or shown in figures 2 and 3?

Reply: Thank you for the comment and the questions. GN1 is the demonstrator version, and it has not been calibrated. Therefore, we prefer not showcasing this improvement in the main Results section. We added a line indicating the improvement from GN1 to GN2 in MC simulations, after the architecture is introduced in the Methods section:

With the updated architecture and training setup, the $\text{cjet}(\text{ljet})$ rejection is improved by a factor of 1.5 (1.7) for a 70% bjet tagging efficiency, in the ttbar sample, and by a factor of 1.3 (1.4) for the corresponding 30% bjet tagging efficiency, in the Zprime sample.

We prefer keeping Figure 2 and 3 as they are. Adding a new GN1 line would make those plots too crowded.

Regarding the comparison with PartT, as you mention, directly comparing GN2 to PartT is difficult. However, we added a paragraph giving a chronological overview in the introduction, including the work done by CMS. So it is clear now in the paper that CMS has its own transformer-based algorithm being developed. We hope by releasing this public dataset (hopefully CMS would do the same), both collaborations can systematically compare the tagger performance and advance together.

Comment: I really like that you added the paragraph:
To facilitate future developments and strengthen the connections between collider experiments and the broader scientific research community, a

simulation sample with all the required information to train GN2 can be acquired via the CERN Open Data Portal [21].â€ However, to really facilitate further developments:

- it should be added that this data is only part of the training data. Would it be possible to publish the Zâ€™ samples too?

Reply: We added a remark saying it is a partial training dataset.

Currently, we prefer to publish only the ttbar sample as both CMS and ATLAS use ttbar events in their training, providing a good benchmark to start comparing algorithms.

Additionally, ttbar modelling has been studied extensively and there has been quite many exchanges on ttbar modelling between collaborations. We believe releasing the ttbar training sample aligns well with this general effort, and the community can benefit a lot from it. Both collaborations (and researchers in other fields) can study flavour tagging using this shared benchmark sample, and provide valuable feedback to the modelling/simulation communities.

In terms of high pT jets, there is still a lot to consolidate. ATLAS uses a modified, non-physical Zâ€™ BSM model, while CMS uses a multijet sample. It is a less commonly studied regime, both the simulation and reconstruction, compared to the ttbar process. We think it would be a better time to release those samples when we have discussed the high pT physics processes (jets) more extensively between collaborations.

To summarise, we want the released public sample to be a good benchmark that allows the community to study jet flavour tagging purely algorithmically factoring out the physics modelling\reconstruction aspects.

Comment: the link to the actual code should be added as well. I think the repo with the configuration files is [<https://gitlab.cern.ch/atlas/open-data/transforming-jet-flavor>] (<https://gitlab.cern.ch/atlas/open-data/transforming-jet-flavor>) while the repo with the actual code is [<https://gitlab.cern.ch/atlas-flavor-tagging-tools/algorithms/salt/>] (<https://gitlab.cern.ch/atlas-flavor-tagging-tools/algorithms/salt/>) . Is that correct? If yes, please add those references as well, maybe with a commit hash or via zenodo snapshot such that it is clear which version of the code was used for the publication here.

- In general, please follow the guidelines of [<https://www.nature.com/nature-portfolio/editorial-policies/reporting-standards>] (<https://www.nature.com/nature-portfolio/editorial-policies/reporting-standards>)

Reply: Thanks a lot for pointing this out. We should have included this vital info. We have added a new repo documenting how to use the open dataset, and information such as software tags are also included
<https://gitlab.cern.ch/atlas/open-data/transforming-jet-flavor>

It is added as a reference in the paper when the open dataset is introduced in the introduction.

Minor comments:

Comment: In fig. 1, top left: Is the beamline along the horizontal axis? Does the "impact parameter \rightarrow " correspond to z_0 of the text (p 11)? Does the vertical dotted line correspond to d_0 of the text (p 11)?

Reply: It is the transverse plane. We added "in the transverse plane" in the caption, and changed "Decay length" to "Transverse decay length", "Impact parameter" to "Transverse impact parameter" (d_0).

Comment: When referring to GEANT4 (currently [24]), the authors should refer to the 3 publications that are given at the bottom of [<https://geant4.web.cern.ch/>] (<https://geant4.web.cern.ch/>) to give credit to the GEANT4 collaboration.

Reply: Added.

Comment: In figs. 2 and 3, would it be possible to indicate the OPs with vertical dashed lines?

Reply: There is already a grid to help the readers locate the OPs. In particular, the 60%, 65%, 70%, 85% and 90% are indicated by the grid. We think it is OK to leave as it is.

Comment: Figure 4 presents the calibrated tagging efficiencies and rejections of GN2 and DL1d, along with their associated uncertainties, based on Run 2 data. Are any results available for Run 3 data using GN2? Given that the LHC is now operating in Run 3, it would be valuable to assess the calibration stability and potential performance shifts under the updated detector and data-taking conditions. Even a qualitative discussion would help understand the robustness of the method.

Reply: We choose Run 2 data to show the performance in data as the simulation/data is well understood. The samples have been reprocessed with our best understanding of the Run 2 data-taking conditions. Although we have collected a large amount of Run 3 data, the collaboration is still in the process of understanding the data. We have preliminary GN2/DL1d

calibration results using partial Run 3 data. A recent Run2+3 Di-higgs analysis has used those preliminary results: <https://arxiv.org/abs/2507.03495>. The sensitivity is improved by 20% due to GN2 (the second last line on page 13). The overall tagger performance is slightly worse compared to Run 2, given the fact that the calibration/reconstruction is still being optimised to the Run 3 conditions. However, the relative improvement from DL1d to GN2 is at the same level as the Run 2 performance. We would prefer not to add such a statement on the Run 3 calibration result that will only be valid for a limited time. We will have publications to document the Run 3 calibrations in the near future, given Run 3 is coming to an end soon.

Comment: The manuscript notes that class-weighted losses are applied during training to mitigate the class imbalance in the track origin classification task. Could the authors clarify what specific class weights were used, and whether they were fixed or dynamically adjusted during training? Providing this information would improve the transparency and reproducibility of the training procedure, especially given the strong class imbalance among the track origin categories.

Reply: Thanks for the question. Track origin class weights are fixed and based on the inverse class frequencies in a representative sample from the training dataset. This is now clarified in the text.

â€œThe class weights are fixed and based on the inverse class frequencies in the training datasetâ€

Comment: In the Discussion section, the manuscript reports that â€œGN2 achieves an efficiency (purity) of 84% (84%). For tracks that are not of heavy-flavour origin, the efficiency (purity) is 85% (96%).â€ However, it is not clear whether these results are derived from Run 2 or Run 3 data (Does the statement from page 4, first paragraph, apply here, too, even though it is in a different section?). Similarly, Table 1 presents ratios of the efficiencies obtained with samples using alternative MC generators, relative to the nominal Powheg Box + Pythia sample, but does not specify which data-taking period these results correspond to. It would be helpful for assessing the potential dependence on detector conditions if the authors could clarify whether these results are based on Run 2 or Run 3 data.

Reply: Thanks for the question, the table shown is from Run 2. Similar results are visible in Run 3. The track eff./purity is from Run 3 samples. This is now clarified in the text.

Comment: More of a curiosity: Figure 5b seems to indicate a linear shift between GN2 and the MC truth. Is there an easy explanation for that, or did that happen by chance?

Reply: Thanks for pointing this out, we added an explanation in the text. It is related to the efficiency/purity of the vertex.

“Unlike SV1, GN2 does not impose explicit selections on track properties such as impact parameters. This leads to higher efficiency, albeit with a small contamination from non-HF tracks, which results in a slightly larger secondary vertex mass”

Comment: To better quantify the contributions of each auxiliary objective, it would be valuable if the authors could provide an ablation table showing the individual impact of disabling the track-origin classification and vertex-grouping tasks. This would help clarify how much each auxiliary task contributes to the overall jet classification performance. In addition, I think it would be better for the flow of reading the manuscript when it would be mentioned that these auxiliary tasks also help to improve the jet tagging performance. Right now, these auxiliary tasks are mentioned a few times, but their impact on jet tagging is only mentioned at the very end, making the reader wonder if there was a benefit from including them.

Reply: Yes, we did perform individual ablation studies. Given the length requirement of the paper, we prefer modifying texts to provide this information. Our studies show that turning off both aux decreases the performance by 30% but turning on one of them can recover most of the performance loss. This indicates the correlations between their contributions to the main tagging task. It is now added in the text:

“Disabling only one of them is sufficient to recover most of the performance loss, indicating that the two tasks are highly correlated in their contributions to the main jet flavour tagging objective.”

Comment: In table 1, what are the not-shown statistical uncertainties? Are these from retraining the networks with different random seeds or from different MC samples? If the quoted uncertainties do not include the effect from retraining the network, please add this information.

Reply: The network is not retrained. The uncertainty comes from evaluating the network with different samples that are finite. Since the sample sizes are large, the uncertainty can be ignored. Updated the last line in the caption:

“Statistical uncertainties from evaluating the same algorithm on different samples are negligible and thus not shown.”

Comment: How sensitive is the GN2 performance on the choice of jet clustering algorithm or radius parameters? How relevant are other jet clustering algorithms for GN2 and subsequent analyses?

Reply: GN2 has a flexible architecture, and it is adapted to achieve different goals, such as $x \rightarrow b\bar{b}$ tagging (clustering radius 1.0) and exotic

jet tagging (re-clustered jet with $R = 1.0$ using $R_{0.4}$ jets). When the underlying physics processes are vastly different, the jet clustering algorithm plays a critical role. However, we have not checked the performance of GN2 on the choice of jet clustering algorithm (or radius parameter) for a given physics process. It is because ATLAS uses mainly the anti-kT $R_{0.4}$ and $R_{1.0}$ jets.

Comment: The manuscript states that “the jet features are concatenated with the feature vectors of up to 40 leading tracks associated with the jet, ranked by the absolute track impact parameter significance.” It would be helpful if the authors could clarify whether zero-padding is applied when fewer than 40 tracks are available, and also specify the batch size used during training. This information is important for understanding the model’s input handling and memory efficiency.

Reply: Thanks for this comment. You are right that it was not clear. In short: formally there is a concatenation, but then there is a masking in the code so that only the tracks present are actually used, so there is no padding. Practically, what this does is only to have an upper cut on the total number of tracks considered, which is 40. We added:

“First, the jet features are concatenated with a fixed-size array of 40 track feature vectors, with unused elements masked when fewer than 40 tracks are available, allowing it to handle variable track multiplicity without zero-padding. Tracks with smaller absolute track impact parameter significance~\cite{FTAG-2019-07} are dropped if there are more than 40 tracks.”

The batch size is 12000 which is now specified in the draft.

“A batch size of 12000 is adopted.”

Comment: The manuscript provides only a brief architectural description of the two auxiliary objectives. For example, it does not describe how the pairwise track embeddings are constructed for the track-pair compatibility task. Since this part is identical to the architecture used in GN1, it would be helpful to include a sentence such as “Please refer to the GN1 paper for details.”

Reply: Update the sentence to be more inclusive:

“GN2 applies the same auxiliary network structures and loss weights as GN1~\cite{ATL-PHYS-PUB-2022-027}.”

Comment: What is the form of the global loss function? What are the “tunable weights” to combine the 3 objectives? How are they tuned?

Reply: They are the same as what was used for GN1. They have not been returned. In the text, it is written

“GN2 applies the same auxiliary network structures and loss weights as GN1~\cite{ATL-PHYS-PUB-2022-027}.”

Comment: The authors should clarify that the “4-fold strategy” refers to training four independent models with the same architecture on mutually exclusive subsets of the training data, rather than performing standard 4-fold cross-validation. If the outputs of the four models are combined during inference (e.g., by averaging predictions), this should also be explicitly stated.

Reply: Indeed, it is a bit complex. Now this is clarified in the text.

“GN2 is trained using a 4-fold strategy to prevent memorisation of the training samples, given their possible use in ATLAS physics analyses. Jets are assigned to one of the four folds pseudo-randomly, with a number seeded by the event number and discrete jet properties. Four classifiers are then trained, each excluding one of the four folds from the training dataset. In physics analysis, each jet is tagged using the classifier it was excluded from during training.”

Comment: Given that each GN2 fold has approximately 2.3 million trainable parameters and operates on large low-level input data, it would be valuable if the authors could provide an estimate of the computational cost during training and inference, at least in the context of simulated events. If available, an estimate of the inference latency in the ATLAS offline or trigger system would also be highly informative, especially for assessing the model's deployability in practical data-taking scenarios.

Reply: We added the estimated training time:

“The network is trained with `\textsc{PyTorchLightning}`~\cite{pytorch,lightning,salt}, consuming roughly 300 GPU hours on an NVIDIA A100 card.”

The inference time is very minimal and negligible compared to other main consumers such as tracking.

“It is deployed in ATLAS software with `\textsc{OnnxRuntime}`~\cite{onnxruntime}, adding negligible CPU time.”

Comment: Could the authors please comment on the IRC safety of the algorithm?

Reply: The previous versions of flavour tagging algorithms used in ATLAS were not IRC safe. GN2 was not designed to cope with IRC safety neither. The design goal of flavour tagging algorithm is mainly to achieve better discriminating power using the reconstructed objects to support the broad physics programmes.

=====

Thanks again for the very constructive comments.

Best regards.

The author

Dear Referee,

We thank you for the careful reading and the very constructive suggestions. Your comments and suggestions have been addressed in the new revision. We have also prepared a pdf-diff to aid the review. Please see our detailed replies to your comments below:

=====

Major comments (comments regarding scientific validity):

Reply: Since all these points are connected, we have prepared one single reply below.

1. The article claims a "paradigm shift from previous approaches" in the abstract and throughout the text (page 2, 12). This statement, as it is, is too strong to be universally true. It needs to be clarified that the paradigm shift being referred to is specific only within the ATLAS experiment, where an end-to-end algorithm is being used for the first time starting with the "GN" series. This fact is reflected in the title of the article, but not elsewhere.

2. A brief paragraph with a chronological review of past/parallel algorithm developments, both in phenomenological studies and in experiments like CMS, is imperative even if the statement of a general "paradigm shift" is dropped. In fact, the present article from ATLAS builds on several concepts that were introduced earlier/in parallel (such as end-to-end tagging and the use of transformers; I would label the inclusion of auxiliary targets as the main novelty of the present article). This should be discussed somewhere, e.g. in a dedicated section before the Results section. Following is a non-exhaustive list of references relevant to LHC experiments:

- JINST 15 (2020) P12012 (end-to-end tagging)
- JINST 13 (2018) P05011 (end-to-end tagging)
- Phys. Rev. D 101, 056019 (2020) (end-to-end tagging)
- CMS-PAS-BTV-22-001 (end-to-end bb-tagging)
- ATL-PHYS-PUB-2022-027 (followed up in major comment #3)
- PMLR 162:18281-18292, 2022 (end-to-end tagging + transformer)

I did not find transformer-based peer-reviewed publications from CMS, but review articles may be cited for further reading on work parallel to GN1/GN2.

3. The GN1 is described as a "demonstrator version" in the present article with a non-peer-reviewed reference from three years ago. This information does not appear until page 13. At this point, it directly contradicts the claim that "GN2 presents a novel approach which breaks the design paradigm of the two-staged algorithms". I fail to see why the information about the development of a first version was not included right before the "new paradigm" discussion in the introductory section. This would:

- (a) help the reader understand the motivation behind the nomenclature of "GN2",
- (b) establish the "GN" series (not GN2, in particular) as the new paradigm in ATLAS, which is factually correct,
- (c) enrich the article's ability to convey the step-by-step developments that large ML models typically go through, and,
- (d) along the lines of major comment #2, the article can be considered a "novel" development only if the "GN" series, as a whole, is established as work done in parallel to other related works, and not a mere application of concepts previously described in other articles.

Therefore, I strongly recommend highlighting GN1, in addition to GN2, in the introductory sections, along with a review of previous/parallel methods.

Reply:

Thank you for these useful comments and the additional references. We have rephrased the introduction accordingly (see the quoted paragraph below), and have added the suggested references. However, we think JINST 15 (2020) P12012 and JINST 13 (2018) P05011 are not strictly "end-to-end" approaches, as they rely on secondary vertex inputs (lower-level algorithms).

We agree with the referee and now explicitly state that the paradigm shift is within ATLAS. We also agree that the aux. tasks are the main novelty, algorithmically. However, we do think that the integration into the ATLAS reconstruction framework, the calibration on real collision data, and the performance robustness across different generators are all valuable components of this draft.

We appreciate the referee's suggestion to have a chronological review of the development so that the readers can get a more complete picture. The updated introduction reflects the suggestions:

The introduction of graph neural networks for object reconstruction in particle physics experiments [23] prompted a shift in the design strategy of the ATLAS Collaboration. This led to the development of the General Network (GN) series of flavour-tagging algorithms, which directly process track and jet information and are trained using target labels extracted from Monte Carlo (MC) simulation. In parallel, the CMS Collaboration followed a similar trajectory, evolving from two-stage approaches [24, 25] to unified, end-to-end network architectures [26–28].

The ATLAS GN taggers use jet flavour prediction as its primary training target and introduces auxiliary training objectives to reconstruct the internal structure of a jet by grouping tracks originating from a common vertex and by predicting the underlying physics process from which each track originated. Such physics domain knowledge is embedded in a combined loss function that enables a simultaneous optimisation, instead of relying on manually optimised low-level algorithms. This flexible structure allows the swift re-tuning of the algorithms to suit alternative experimental conditions or physics goals. A demonstrator version, GN1, achieves the above design goals using a graph-neural-network [29], while the deployment version, GN2, applies a single Transformer model [30], illustrated in Figure 1.

Further comments to improve the article based on Nature Communications standards:

1. The abstract mentions "substantial benefits for physics analyses" with specific examples, but the improvements, e.g. in sensitivity, are not explored in the article, but only referenced. I recommend removing this information from the abstract and replacing it with a quantification of the improvement in the per-jet background rejection/tagging efficiency, which is the primary message of the article (please also see Further comment #8).

Reply: Thank you for this comment. We added the improvements seen in the data measurements.

The GN2 tagger demonstrates clear improvements over DL1d in collision data. For instance, the measured $\langle \text{cjet} \rangle$ rejection in data is increased by a factor of 3.5 (1.8) for the 70% OP.

I see that you have added this info to page 7 (in reaction to a comment further down). I was under the impression that the referee wanted this info added to the abstract too, but I cannot see that the abstract has changed in the diff.

2. Page 2, para 3: It would be interesting to know what fraction (quantitatively) of b-jets contain a tertiary vertex in simulation. How does this additional vertex, when present, affect the Billoir fit, L_{xy} , m_{SV} (page 8 and Fig 5)? Naively, it would appear that the presence of a tertiary vertex would skew the quantities under the one-vertex-per-jet hypothesis, which is apparently adopted ("...performed on the tracks selected by the GN2 and SV1 vertex finding algorithms").

Reply: In general, both GN2 and SV1 include tracks from the tertiary vertex, while GN2 has more flexibility for them. Most of the time the distance between the b-hadron and the b-to-c tertiary vertex is within the spatial resolution of the tracker and hence can be merged into a single vertex. Indeed, one typically needs additional information to try to recover the tertiary such as assuming the b-hadron decay axis as the jet axis. More information is available here: <https://cds.cern.ch/record/2645405>

We agree with the referee generally that these are interesting studies, but we would like to avoid going into details for this paper. The possibility to further study and develop the auxiliary tasks is mentioned in the paper:

Although the outputs from the auxiliary tasks described above mainly serve as a way to improve heavy-flavour jet identification, with future development, their direct usage in physics analyses remains a promising possibility.

3. Page 2, para 4: “relative to the algorithm used in most”: do you mean “algorithms” or is there one algorithm you are referring to? For the latter (former), a quantification of the (ballpark of) improvement would be useful.

Reply: Thank you. Indeed the reference discussed three taggers, so this needs to be clarified that we are comparing DL1d to the best one among the three: DL1r. Changed this line to:

“DL1d has already improved the performance by a factor of 1.3 relative to the most advanced algorithm used in published Run-2 physics analyses~\cite{FTAG-2019-07}.”

4. Page 2, para 5: “Underlying physics process” may be misleading here (though it is clarified later on page 7, last para). Parentheses with “primary interaction”, “hadrons”, “pileup” may help clarify what “physics process” means in this context.

Reply: We would like to keep the sentence as is. The introduction briefly tells the readers what each auxiliary training object tries to achieve, and the Discussion section discusses it in detail.

5. Page 4, para 1: “similar results are obtained with Run-2 samples”: Two comments here:

- Please clarify that Run-2 samples are with $\sqrt{s}=13$ TeV at the end of the para; this is mentioned before, but mentioning 13.6 for one makes it look like the same energy is used for the other in this text.
- This is an interesting observation. How “similar” are they? Is this a consequence of having heterogeneous \sqrt{s} samples in the training? Or is it because flavour-tagging information leveraged by the network is independent of \sqrt{s} ? What gain (if any) would we expect from evaluating performance on 13 TeV samples using a standalone 13 TeV training, and vice versa? A clarification here would help motivate the use of different Run samples in the training.

Reply: We added “at $\sqrt{s} = 13$ TeV” at the end of this paragraph.

The difference in CME (0.6 TeV) has minimal impact on the tagger performance, as it only affects the p_T distributions of the jets mildly. The major difference between Run 2 and Run 3 MC samples comes from the simulation conditions. The Run 2 data and MC have been understood really well. The 13 TeV training sample uses the latest-and-best track and jet reconstruction made available for Run 2. On the contrary, we are still in the process of tuning the Run 3 MC samples. The 13.6 TeV training sample uses preliminary track and jet reconstruction that will be improved. The generators are also subject to further tuning.

Training with the 13 TeV (13.6 TeV) samples only makes the performance in the 13 TeV (13.6 TeV) samples slightly better (~5%), compared to training with mixed 13 TeV and 13.6 TeV samples. However, mixing the 13 TeV and 13.6 TeV samples allows us to train an algorithm that is not biased by the above difference significantly, as the referee hypothesized. It is a practical decision, as ATLAS cannot afford to maintain two separate versions of the tagger simultaneously, given the amount of resources needed to perform validation and calibration.

We added this line in the draft to better motivate this decision:

“A mixture of samples generated at $\sqrt{s}=\sqrt{13.6}$ TeV and $\sqrt{s}=\sqrt{13}$ TeV is used in the training, to achieve similar performance in both conditions. In this section, the performance evaluated with Run-3 samples at $\sqrt{s}=\sqrt{13}$ TeV is presented.”

6. Page 4, para 4: “Applying the 70% OP efficiency of 30%”. This indicates a strong dependence on jet p_T . While the reasons are qualitatively discussed here, the discussion would benefit from a p_T -dependent plot of the tagging efficiencies at a fixed mistag rate (or vice versa). A less-ideal option is to at least mention the median p_T of the two categories discussed here and shown in Fig 2.

Reply: the plots are added as Aux Material here:
<https://atlas.web.cern.ch/Atlas/GROUPS/PHYSICS/PAPERS/FTAG-2023-05/>

These are not part of the paper to avoid having it overcrowded with plots. The pT range mentioned in the texts and plot captions can already give the reader an idea about the pT. Therefore, we do not think adding the median of the pT would add more value.

7. Fig 2 caption: "binomial distribution": How are correlations treated while taking the ratio, since I understand that the same jets are used in GN2 and DL1d curves?

Reply: The correlations are not taken into account when calculating the ratio (standard error propagation assuming independent variables is used). Thanks for pointing it out. The errors on the individual rejections are binomial errors, but not the errors on the ratio. Changed it to:

"The 68% confidence intervals calculated assuming no correlations between the rejections are indicated by the shaded regions, and the uncertainty on each rejection is obtained according to a binomial distribution."

It does lead to an overestimate of the ratio errors, but this should not change any conclusions.

8. Page 7, para 1: "clear improvements over DL1d": This is one of the main results of the article. Can this be quantified, e.g. with "x-times higher c-jet rejection at b-jet efficiency of y in collision data", including a rough uncertainty stemming from the calibrations? This information should also be added to the abstract.

Reply: Changed the texts to:

"The GN2 tagger demonstrates clear improvements over DL1d in collision data. For instance, the measured c_{jet} (l_{jet}) rejection in data is increased by a factor of 3.5 (1.8) for the 70% OP. The measurements in data provides conclusive evidence of the enhanced performance enabled by advanced machine-learning algorithms in identifying heavy-flavour jets at the LHC."

9. Page 8, para 3 and Fig 5: SV1 seems to overall underestimate the SV mass, whereas GN2 consistently overestimates. Is the reason understood? The text mentions "good agreement", but this statement should be improved with a short discussion.

Reply: SV1 is a traditional vertexing approach that relies on selecting tracks with a number of hard cuts such as cuts on the impact parameters significance. This means SV1 filters out a fraction of tracks coming from the b-hadron, resulting in overall degradation of the overall efficiency (see Aux plot here for efficiency vs purity comparison: <https://atlas.web.cern.ch/Atlas/GROUPS/PHYSICS/PAPERS/FTAG-2023-05/>). On the contrary, GN2 does not place explicit selections on tracks and we apply a rather loose selection by requiring only one track with predicted HF origin when producing Fig 5. There are tracks not from the b-hadron decays grouped by GN2, resulting in overestimating the mass.

Added this line in the draft:

"Unlike SV1, GN2 does not impose explicit selections on track properties such as impact parameters. This leads to higher efficiency, albeit with a small contamination from non-HF tracks, which results in a slightly larger secondary vertex mass."

10. Table 1: The c-jet efficiency is altered by 10% when choosing Herwig over Pythia. This is not insignificant. This is also counterintuitive as I would expect mismodeling in b-jets to be an exaggeration of the midmodeling in c-jets, unless, e.g., D-hadron simulation is particularly different between Pythia and Herwig.

(a) The text (page 10, para 1) can be improved with a discussion and a summary of key differences between Pythia and Herwig which may affect D/c hadron simulation.

(b) This observation also invites investigating the efficiency ratios corresponding to at least one c-tagging OP (e.g. 30% c-eff). The same comment somewhat applies to Sherpa vs. Powheg+Pythia as well, where differences in c and light efficiencies are up to 7.7%, but negligible for b.

Reply: We thank the referee for this insightful comment. The modelling of

charm hadrons including aspects such as fragmentation and hadronisation is indeed a particularly challenging area. Compared to bottom hadrons, the experimental constraints from LEP and other measurements are less precise, which results in greater variability across generator tunes when it comes to charm jets. This naturally leads to more significant discrepancies between generators like Pythia and Herwig in c-jet observables, as also seen in the c-tagging efficiency. Some comparative studies can be found here: <https://cds.cern.ch/record/1709132/files/ATL-COM-PHYS-2014-288.pdf> and <https://cds.cern.ch/record/2816367/files/ATL-PHYS-PUB-2022-035.pdf> while a concise review on c-hadron modelling is here: https://indico.cern.ch/event/1387465/contributions/6019731/attachments/2925372/5135321/miham_9_11_2024_FTAG_Genova.pdf.

Regarding point b), this is systematically studied by ATLAS. We attach here an example of c-tagging MCMC ratio.

We certainly agree that this is an interesting and important area of study, and indeed internal investigations are ongoing in this direction. However, we believe that a detailed discussion on how these differences impact FTAG performance would go beyond the scope of this paper. Here, our primary focus is to show that GN2 does not exacerbate the existing discrepancies between generators.

11. Page 12: A coarse hyperparameter optimization is mentioned.
(a) It may help to elaborate on which hyperparameters were altered. Was the model size changed? Is it known if a larger model would lead to overfitting?
(b) It would also be good to document if any specialized computing methods were used for hyperparameter optimization in the GN series (e.g. Grid GPU architectures [*]), as this is a challenge in modern large model trainings and is of interest to the HEP community as well as the broader ML community.
[*] <https://atlas.web.cern.ch/Atlas/GROUPS/PHYSICS/PLOTS/FTAG-2019-001/>

Reply: Good point. We added such as the number of layers, so the readers know the model size is changed during the hyperparameter optimisation. Larger models do not necessarily lead to overfitting. The large training dataset and the 4-fold strategy ensures GN2 is not overtrained.

We did not use any Grid GPU functionalities in the GN2 training. It is a method currently being explored. We added the following line to give the readers an idea of the resource cost:

The network is trained with `torch.nn.LazyTensor` (PyTorch Lightning)~\cite{pytorch,lightning,salt}, consuming roughly 300 GPU hours on an NVIDIA A100 card.

12. Page 13, para 2: ranked by absolute track impact parameter significance, is the ranking/sorting of any significance? For a transformer architecture, I would expect ordering to have no physical significance. A comment on whether the ordering matters will be helpful for the article, especially since DIPS emphasizes on the lack of ordering.

Reply: Here the ranking refers to the truncation of tracks where there are more than 40 tracks available in a given jet, not the ordering of the inputs. This part has been rewritten according to another referee's suggestions, which avoids such confusions:

First, the jet features are concatenated with a fixed-size array of 40 track feature vectors, with unused elements masked when fewer than 40 tracks are available, allowing it to handle variable track multiplicity without zero-padding. Tracks with smaller absolute track impact parameter significance~\cite{FTAG-2019-07} are dropped if there are more than 40 tracks.

13. Page 13, para 3: 4-fold strategy: What is the aggregation strategy and

plans for usage in analyses? Does the proposal involve averaging over the predictions of 4 different models in physics analyses? Are the results shown in the article the average of 4 models? A comment here would help clarify.

Reply: Indeed, it is a bit complex. Now this is clarified in the texts.

â€œGN2 is trained using a 4-fold strategy to prevent memorisation of the training samples, given their possible use in ATLAS physics analyses. Jets are assigned to one of the four folds pseudo-randomly, with a number seeded by the event number and discrete jet properties. Four classifiers are then trained, each excluding one of the four folds from the training dataset. In physics analysis, each jet is tagged using the classifier it was excluded from during training.â€

14. Page 2, para 2: neither of the above -> none of the above

Reply: Fixed.

Page 2, para 2: The state-of-the-art algorithm -> The state-of-the-art algorithms (unless you mean one single algorithm, in which case it should be named and referenced)

Reply: Fixed.

Page 2, para 2: should end with â€œ, respectivelyâ€

Reply: Fixed.

Page 2 and onward: please use â€œRun-2â€ or â€œRun 2â€, â€œRun-3â€ or â€œRun 3â€ consistently. â€œTransformerâ€ vs. â€œtransformerâ€: Is there a reason for different types of capitalizations?

Reply: Changed to â€œtransformerâ€ consistently. Run 2 (3) refers to the data taking periods themselves, while Run-2 (3) are used as adjectives.

Figure 1: The text, especially the ones with subscripts, will be more readable (e.g. on an A4 print) with larger font sizes.

Reply: Enlarged fonts.

Reference 3: â€œTheâ€ in â€œATLAS Collaborationâ€ is an outlier. References 2, 3, 42, and potentially others are missing preprint links. Reference 73: â€œA.â€ -> â€œATLASâ€

Reply: Fixed.

=====

Thanks again for the very constructive comments.

Best regards.

The author

Dear Referee,

We thank you for the careful reading and the very constructive suggestions. Your comments and suggestions have been addressed in the new revision. We have also prepared a pdf-diff to aid the review. Please see our detailed replies to your comments below:

=====
Comments: Is GN2 an abbreviation? If yes I suggest mentioning it in the paper.

Reply: Yes, it is short for "general network". We added a more detailed overview of the development history, as requested by another referee, and now it is spelled out.

"This led to the development of the General Network (GN) series of flavour-tagging algorithms, which directly process track and jet information and are trained using target labels extracted from Monte Carlo (MC) simulation."

Comments: It may be worth adding a sentence in the introduction that gives the non-expert readers an idea of what a hadronic jet is as this is the main object of study in this paper.

Reply: Added a short phrase: Hadronic jets, collimated streams of particles initialised by quarks or gluons,....

Comments: The third paragraph nicely explains the physics behind the signature of b-jets. As the presented tagger also identifies c-jets and tau decays, it would be good to also explain their signatures.

Reply: Mentioned the lifetime of c-hadrons and tau-leptons, and the low decay multiplicity of tau-leptons.

"Displaced vertices can also be produced by χ hadrons, which have lifetimes of ~ 0.2 - 1.0 ps, depending on the species~\cite{Belle-II:2021cxx,Belle-II:2022ggx,10.1093/ptep/ptaa104}, and τ leptons, which have a lifetime of ~ 0.29 ps but a much lower decay multiplicity~\cite{Belle:2013teo,10.1093/ptep/ptaa104}. The majority of b jets also contain a tertiary vertex from the decay of the χ hadron produced in the b hadron decay."

Comments: It is very good that the data of this study is publicly available.

Reply: Thank you!

Comments: Are the relative weighting factors f_b , f_c and f_{τ} dependent on the specific selection of an analysis? If yes, does this mean that they should be determined for each analysis? I suggest clarifying this in the paper.

Reply: f_b (c , τ , u) is not determined for each analysis but set by the flavour tagging working group to harmonize the calibration efforts. The choice was historically driven by the $VH \rightarrow bb/cc$ analyses. Although they are not optimised individually, we offer several operating points for the analyses to choose. We would prefer not to go in the details of this in the paper, as it is largely an organisational effort.

Comments: Regarding the above mentioned fractions, why are they different for the different taggers? Shouldn't the optimal values be those of the true fractions (which are known for MC)?

Reply: The optimal values depend on both the flavour fractions AND the $p_b(u, c, \tau)$ distributions. Since the latter vary across different taggers, the optimal f_i is not the same given a specific MC sample.

Comments: Figure 5: Does the difference between distributions come from different efficiencies, or are there also different resolutions of the reconstructed vertices?

Reply: The difference mainly comes from the efficiencies, as seen in Fig 5 (a). At low L_{xy} , the efficiency of SV1 is much lower. It results in differences in the vertex mass. A short discussion is added:

“Unlike SV1, GN2 does not impose explicit selections on track properties such as impact parameters. This leads to higher efficiency, albeit with a small contamination from non-HF tracks, which results in a slightly larger secondary vertex mass”

Comments: Table 1 and related text. The main argument is based on the relative disagreement. Therefore I suggest adding these values to the table.

Reply: Indeed the main argument is based on the relative disagreement, however we would like to keep the table as is, to avoid complicating it with errors on the relative disagreement, which are negligible anyway. We believe the message comes across sufficiently well with the present form of

the table.

Comments: Methods section: I suggest "pp collisions at the LHC" instead of "LHC collisions".

Reply: Done.

=====

Thanks again for the very constructive comments.

Best regards.

The author